# Feature Bagging Provides Stability

**Yuheng Ma** [1]    **Qiang Sun** [2][3]

## Abstract

We study feature bagging through the lens of algorithmic stability. Feature bagging is an ensemble strategy that aggregates base learners trained on randomly subsampled feature subsets, possibly in a data-dependent manner. We introduce feature instability (FI), the feature-axis analogue of instance instability (II), which measures sensitivity to removing a single feature. Smaller values of II or FI correspond to stronger stability, and our experiments show that FI captures generalization-relevant information complementary to II. Within this framework, we analyze feature bagging in both a parametric linear model and a model-free setting inspired by recursive feature subsampling in random forests. In both settings, we establish formal guarantees showing that feature bagging improves the relevant stability relative to its non-bagged counterpart, with larger improvements under more aggressive subsampling. We further show that a modest number of bagging rounds is sufficient to approach the infinite-bagging stability level.

## 1. Introduction

Stability is a fundamental prerequisite for trustworthy machine learning (Murdoch et al., 2019; Yu & Kumbier, 2020; Xing et al., 2021; Zhou et al., 2023), and is closely connected to generalization (Bousquet & Elisseeff, 2002), uncertainty quantification (Wang et al., 2023), and model selection (Meinshausen & Bühlmann, 2010; Nogueira et al., 2018). For a predictor $f$ trained on $n$ samples in dimension $d$, an informal way to capture instance-wise instability is to measure how much its prediction changes when one training

[1]KLATASDS-MOE, School of Statistics, East China Normal University [2]University of Toronto [3]MBZUAI. Correspondence to: Qiang Sun <qsunstats@gmail.com>.

*Proceedings of the 43rd International Conference on Machine Learning*, Seoul, South Korea. PMLR 306, 2026. Copyright 2026 by the author(s).

instance is removed:

$$\frac{1}{n}\sum_{i=1}^{n}\left(f(\boldsymbol{x}) - f^{-i,:}(\boldsymbol{x})\right)^2 \tag{1}$$

where $f^{-i,:}$ denotes the predictor obtained after removing the $i$-th training instance. Smaller values indicate that the algorithm is more stable to single-instance perturbations. We use this quantity only as motivation; Definition 2.1 formalizes $\phi$-instance stability and names the associated instance-instability measure. Regularization (Bousquet & Elisseeff, 2002) and bagging (Soloff et al., 2024a) are two standard approaches for improving instance stability.

Feature bagging has been recognized as a potential driver of ensemble success (Breiman, 2001; Sutton et al., 2005; LeJeune et al., 2020), but the mechanism behind its empirical effectiveness remains debated. One promising explanation is that feature bagging acts as a form of regularization (Mentch & Zhou, 2020; Curth et al., 2024). This viewpoint makes stability a natural lens for studying feature bagging. Instance stability is a natural starting point because it measures sensitivity to changes in the training instances. However, it does not directly capture sensitivity to changes in the feature set, precisely the axis on which feature bagging operates; see Section 2.2 for empirical evidence that feature-axis perturbations carry generalization-relevant information that is complementary to instance stability. These considerations motivate our central question:

> *Does feature bagging improve stability, and if so, how?*

Answering this question requires a feature-axis analogue of classical instance instability. We therefore introduce *feature instability* (FI), the leave-one-feature-out analogue of (1):

$$\frac{1}{d}\sum_{j=1}^{d}\left(f(\boldsymbol{x}) - f^{:,-j}(\boldsymbol{x})\right)^2, \tag{2}$$

where $f^{:,-j}$ denotes the predictor obtained after removing the $j$-th feature. This informal quantity is the feature-axis analogue of instance instability; the formal definitions of feature-instability and $\phi$-stability are given later in Definition 2.2.

We answer this question through the following contributions:

1. We introduce FI as the feature-axis analogue of classical II. This gives a two-axis view of stability tailored to feature bagging: II measures sensitivity to removing training instances, whereas FI measures sensitivity to removing features. We further show empirically that FI carries generalization-relevant information that is complementary to II, motivating FI as a distinct object of study.

2. In linear regression, we give an exact asymptotic characterization of how bagging affects II and FI. The analysis explains how ensemble averaging improves stability and how the stability gains depend on the sampling of both instances and features. It also reveals that feature-level perturbations have their own scaling behavior, governed by the ambient dimension, which is invisible from II alone.

3. Beyond parametric models, we develop a model-free theory for FI under feature bagging, with particular emphasis on recursive feature subsampling. This theory applies to algorithms such as random forward selection and random forests, where features are resampled repeatedly and data-dependently. It shows that more aggressive feature subsampling yields stronger feature-stability guarantees, identifies regimes in which feature bagging is provably more stable than its non-bagged counterpart, and establishes that a number of bagging rounds proportional to the dimension is enough to approach the infinite-bagging stability level.

The rest of the paper proceeds as follows. Section 2 formalizes II, FI, and bagging. Section 3 analyzes the linear-model setting, while Section 4 develops the model-free theory for recursive feature subsampling. Proofs, derivations, and additional experiments are deferred to the appendix.

## 2. Basic Concepts

### 2.1. Stability

We begin by fixing notation for the two perturbation axes considered in this paper. Let $\mathcal{D} = (\boldsymbol{X}, \boldsymbol{y})$ be a dataset with $n$ instances and $d$ features. The data matrix $\boldsymbol{X} \in \mathbb{R}^{n \times d}$ has rows $\boldsymbol{x}_i \in \mathcal{X} = \mathbb{R}^d$, and the labels are collected in $\boldsymbol{y} \in \mathcal{Y}^n$. For integer $K$, let $[K] = \{1, \dots, K\}$. For $i \in [n]$, denote by $\mathcal{D}^{(i),:}$ and $\mathcal{D}^{-i,:}$ the datasets obtained by replacing or removing the $i$-th sample from $\mathcal{D}$, with corresponding data matrices $\boldsymbol{X}^{(i),:} \in \mathbb{R}^{n \times d}$ and $\boldsymbol{X}^{-i,:} \in \mathbb{R}^{(n-1) \times d}$. Similarly, for $j \in [d]$, let $\boldsymbol{X}^{:,(j)}$ and $\boldsymbol{X}^{:,-j}$ denote the data matrices with the $j$-th feature replaced or removed, and write $\mathcal{D}^{:,(j)}$ and $\mathcal{D}^{:,-j}$ for the associated datasets. Throughout

the paper, our primary convention is feature or instance removal. Replacement-based variants are also common in the stability literature, and we discuss their relationship to our removal-based convention in Appendix A.

Following the standard algorithmic-stability setup, we let a learning algorithm $\mathcal{A}$ map a dataset and a random seed to a predictor $f = \mathcal{A}(\mathcal{D}, \xi) : \mathcal{X} \to \mathcal{Y}$, where $\xi$ collects all sources of algorithmic randomness. Conditional on $\xi$, the algorithm is deterministic; for example, $\xi$ may encode random initialization, data shuffling, or subsampling. Classical instance stability quantifies the sensitivity of $\mathcal{A}$ to instance-wise perturbations of $\mathcal{D}$. We recall this notion first, and defer a discussion of related formulations to Appendix A.

**Definition 2.1.** An algorithm $\mathcal{A}$ is $\phi$-instance stable if

$$\mathrm{II} := \frac{1}{n} \sum_{i=1}^{n} \mathbb{E}\left[\left(f(\boldsymbol{x}) - f^{-i,:}(\boldsymbol{x})\right)^2\right] \leq \phi^2, \qquad (3)$$

where $f = \mathcal{A}(\mathcal{D}; \xi)$ and $f^{-i,:} = \mathcal{A}(\mathcal{D}^{-i,:}; \xi)$ are evaluated using the same algorithmic randomness $\xi$. The expectation is interpreted according to the stability convention being used. In the uniform convention, the expectation is over algorithmic randomness and the bound holds for fixed $\mathcal{D}$ and $\boldsymbol{x}$. In distributional variants, the expectation may also average over the randomness in $\mathcal{D}$ and $\boldsymbol{x}$. With some abuse of notation, we call the left-hand side of (3) the instance instability of $\mathcal{A}$, with smaller values indicating greater instance stability.

The uniform convention is the standard instance-stability convention in the algorithmic-stability literature, while the distributional convention is weaker and is used in some of our average-case results, particularly in Section 3. The average over $i$ makes the criterion invariant to permutations of the dataset, even when $\mathcal{A}$ itself is not permutation-invariant. If $\mathcal{A}$ is permutation-invariant, then (3) reduces to $\mathbb{E}[(f(\boldsymbol{x}) - f^{-i,:}(\boldsymbol{x}))^2] \leq \phi^2$ for any $i \in [n]$, under the same expectation convention.

To study feature bagging, we need the analogous notion along the feature axis. Rather than modifying training instances, we remove one feature from the input representation and measure the resulting change in prediction.

**Definition 2.2.** An algorithm $\mathcal{A}$ is $\phi$-feature stable if

$$\mathrm{FI} := \frac{1}{d} \sum_{j=1}^{d} \mathbb{E}\left[\left(f(\boldsymbol{x}) - f^{:,-j}(\boldsymbol{x})\right)^2\right] \leq \phi^2, \qquad (4)$$

where $f = \mathcal{A}(\mathcal{D}; \xi)$ and $f^{:,-j} = \mathcal{A}\left(\mathcal{D}^{:,-j}; \xi\right)$ are evaluated using the same algorithmic randomness. By convention, we regard $f^{:,-j}$ as a predictor on the original feature space that ignores the $j$-th feature. The expectation follows the same convention as in Definition 2.1. We call the left-hand side of (4) the feature instability of $\mathcal{A}$, with smaller values indicating greater feature stability.

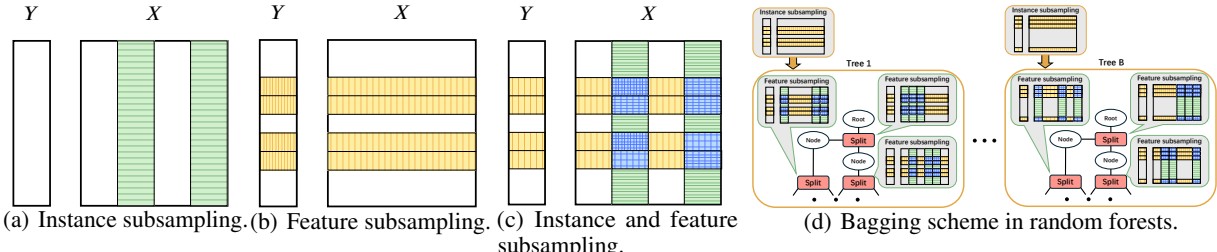

(a) Instance subsampling. (b) Feature subsampling. (c) Instance and feature subsampling. (d) Bagging scheme in random forests.

*Figure 1.* Illustration of subsampling and bagging schemes. In the random forest example (d), all splits within a single tree share the same instance subsample (yellow lines), while each split uses an independently drawn feature subsample (green columns).

The preceding definition fixes a shared seed across the full-data and feature-removed runs. We next allow these two runs to use coupled randomness. This is useful when removing a feature changes the admissible resampling space, as in the linear-model analysis of Section 3.

**Definition 2.3** (Coupled feature stability). For a dataset $\mathcal{D}$, let $\Xi(\mathcal{D})$ denote the seed space used to run $\mathcal{A}$ on $\mathcal{D}$. For each feature $j$, let $\Gamma_j(\mathcal{D})$ be a coupling on $\Xi(\mathcal{D}) \times \Xi(\mathcal{D}^{:,-j})$, that is, a joint law for the seeds $(\xi, \xi^j)$ used in the full-data and feature-removed runs. An algorithm $\mathcal{A}$ is $\phi$-feature stable under the coupling family $\Gamma = \{\Gamma_j(\mathcal{D})\}_{j=1}^d$ if

$$\frac{1}{d} \sum_{j=1}^{d} \mathbb{E}\left[\left(\mathcal{A}(\mathcal{D};\xi)(\boldsymbol{x}) - \mathcal{A}(\mathcal{D}^{:,-j};\xi^j)(\boldsymbol{x})\right)^2\right] \leq \phi^2, \quad (5)$$

where, in the $j$-th summand, $(\xi, \xi^j) \sim \Gamma_j(\mathcal{D})$, and any additional averaging follows the same convention as above. The left-hand side is the coupled feature instability under $\Gamma$.

Definition 2.2 is recovered as the shared-randomness special case of Definition 2.3: after identifying the two seed spaces, $\Gamma_j(\mathcal{D})$ is the diagonal coupling $\xi^j = \xi$. Unless otherwise stated, FI refers to this shared-randomness convention. Section 3 is an exception. In the linear-model analysis, the reduced-feature run shares the unaffected sketching randomness but redraws the feature sketch from the reduced feature set, corresponding to a affected-axis resampling coupling with $\xi^j \neq \xi$. In contrast, Section 4 returns to the shared-randomness coupling $\xi^j = \xi$, because its goal is a conditional algorithmic stability guarantee.

The FI criterion in (4) differs from the II criterion in (3) in two important respects. First, FI perturbs only the input matrix and leaves the response vector $\boldsymbol{y}$ unchanged. Second, the average over features is essential: features are generally non-exchangeable, and learning algorithms are typically not permutation-invariant along the feature dimension.

### 2.2. Why Do We Care about Feature Stability?

Feature stability is motivated by several practical and methodological concerns. First, relevant features may be absent, unavailable, or deliberately excluded; feature inclusion can therefore depend on preprocessing decisions, acquisition constraints, and analyst judgment (Jeng et al., 2024; Ma et al., 2025; Shen & Xiu, 2025; Poudel et al., 2025). The FI measure quantifies how sensitive an algorithm is to this feature-level variation. Second, feature entries may themselves be perturbed because of corruption (McWilliams et al., 2014), missingness (Little & Rubin, 2019; Chen & Xu, 2023), or measurement error (Hou et al., 2026). Algorithms with stronger feature stability are therefore more robust to such perturbations.

Beyond these practical considerations, we empirically examine whether FI carries information about generalization, paralleling a central motivation for the literature on instance stability (Bousquet & Elisseeff, 2002; Hardt et al., 2016). In a controlled synthetic regression benchmark, we vary model complexity and subsampling behavior in random forest models, and measure how FI and II account for variation in the generalization gap. Section C provides the detailed experiments and analysis. The results indicate that FI captures generalization-relevant variation that is not explained by II alone. Across Tables 2–5, the conditional contribution of FI after accounting for II remains positive in all displayed synthetic and real-data settings. The noise-control results in Tables 6–9 further show that replacing FI with unrelated Gaussian or permuted noise does not replicate this signal. These findings indicate that FI and II provide complementary measures for capturing generalization-relevant variation.

### 2.3. Instance and Feature Bagging

We now formalize the bagging schemes analyzed in the paper. For both instances and features, we use sub-bagging: each base learner is trained on a uniformly sampled fixed-size subset without replacement. Other bagging schemes can lead to similar theoretical conclusions (Soloff et al., 2024a), but fixed-size subsampling is especially natural for features. Classical bagging (Breiman, 1996) and Poissonized bagging (Oza & Russell, 2001) sample with replacement, which can select the same feature multiple times and create non-identifiability issues. Bernoulli sub-bagging (Harrington, 2003; Wu & Sun, 2025) produces random sub-

set sizes, which can complicate models that require a fixed input dimension, such as neural networks.

**Instance bagging.** We first consider instance bagging, the classical bagging setting. In each sub-bagging round, $m$ of the $n$ instances are sampled uniformly without replacement. Let $p = m/n$ denote the instance-subsampling ratio, and write $\boldsymbol{\mu}^{(b)} = (\boldsymbol{\mu}_1^{(b)}, \ldots, \boldsymbol{\mu}_m^{(b)})^\top$ for the indices selected in the $b$-th subsample. For any algorithm $\mathcal{A}_0$, we define its evaluation on the $b$-th subsample as $\mathcal{A}(\mathcal{D}, \xi^{(b)}) := \mathcal{A}_0(\mathcal{D}^{\boldsymbol{\mu}^{(b)}, :})$, where $\mathcal{D}^{\boldsymbol{\mu}^{(b)}, :} = \{(\boldsymbol{x}_{\boldsymbol{\mu}_1^{(b)}}, y_{\boldsymbol{\mu}_1^{(b)}}), \ldots, (\boldsymbol{x}_{\boldsymbol{\mu}_m^{(b)}}, y_{\boldsymbol{\mu}_m^{(b)}})\}$. See Figure 1(a) for an illustration. This construction defines a randomized algorithm $\mathcal{A}$, with randomness induced by instance subsampling. Let $f^{(b)} := \mathcal{A}(\mathcal{D}, \xi^{(b)})$ be the predictor trained in the $b$-th round. The bagged predictor averages $B$ such predictors:

$$f^B(\boldsymbol{x}) := \frac{1}{B} \sum_{b=1}^{B} f^{(b)}(\boldsymbol{x}). \tag{6}$$

We refer to $B$ as the number of bagging rounds. The infinite-bagging limit is the corresponding expectation over the subsampling randomness:

$$\mathcal{A}_\infty(\mathcal{D})(\cdot) := \mathbb{E}_{\xi^{(b)}}[f^{(b)}(\cdot)] = \mathbb{E}_{\xi^{(b)}}[\mathcal{A}(\mathcal{D}, \xi^{(b)})(\cdot)]. \tag{7}$$

We write its output as $f^\infty$, so $f^\infty(\boldsymbol{x}) = \mathcal{A}_\infty(\mathcal{D})(\boldsymbol{x}) = \mathbb{E}_{\xi^{(b)}}[f^{(b)}(\boldsymbol{x})]$. This predictor is the large-$B$ limit of $f^B$.

**Feature bagging.** Feature bagging is defined analogously. Let $s \leq d$ denote the number of selected features and $q = s/d$ the subsampling ratio. In each round, a subset $\boldsymbol{\nu}^{(b)} = (\boldsymbol{\nu}_1^{(b)}, \ldots, \boldsymbol{\nu}_s^{(b)})$ is drawn uniformly at random without replacement. The corresponding randomized algorithm is $\mathcal{A}(\mathcal{D}, \xi^{(b)}) := \mathcal{A}_0(\mathcal{D}^{:, \boldsymbol{\nu}^{(b)}})$. Here, $\mathcal{D}^{:, \boldsymbol{\nu}^{(b)}}$ denotes the dataset restricted to the selected features and is illustrated in Figure 1(b). The finite and infinite bagged predictors are defined as in (6) and (7). At prediction time, the same test point is evaluated by all base learners, but the $b$-th learner only receives the coordinates $\boldsymbol{x}^{\boldsymbol{\nu}^{(b)}}$ that match its sampled feature subset. The final prediction is the average of these base-learner predictions. This differs from instance bagging, where every base learner receives the full test point.

**General bagging.** Instance and feature bagging can be combined by jointly subsampling both axes, yielding an ensemble whose randomness arises from both sources; see Figure 1(c). Section 3 explores their interaction. In random forests, feature bagging is implemented recursively rather than once per tree: as Figure 1(d) illustrates, all splits in a tree share the same instance subsample, while feature subsampling is performed independently at each node. Section 4 analyzes this recursive feature-subsampling strategy.

## 3. Stability under Model Assumptions

This section studies the effect of bagging on the II and FI measures in linear regression, where explicit model assumptions allow sharp comparisons.

### 3.1. Basic Settings

**Generating Distribution.** We consider the standard linear regression model (LeJeune et al., 2020; Chen et al., 2023):

$$y_i = \boldsymbol{x}_i^\top \boldsymbol{\beta}^* + \varepsilon_i, \tag{8}$$

where $\boldsymbol{x}_i \in \mathbb{R}^d$ is the feature vector, $y_i \in \mathbb{R}$ is the response, $\varepsilon_i \sim \mathcal{N}(0, \sigma^2)$ is Gaussian noise, and $(\boldsymbol{x}_i, \varepsilon_i)_{i=1}^n$ are i.i.d. copies of $(\boldsymbol{x}, \varepsilon)$. Additionally, we assume that $\boldsymbol{x} \sim \mathcal{N}(\boldsymbol{0}, \boldsymbol{I})$, $\boldsymbol{\beta}^* \sim \mathcal{N}(\boldsymbol{0}, \boldsymbol{I}/d)$, and that $\boldsymbol{x}$, $\boldsymbol{\beta}^*$, and the noise $\varepsilon$ are mutually independent. The Gaussian assumptions on $\boldsymbol{x}$ and $\varepsilon$ can be generally relaxed to bounded moment conditions under which the same theoretical results hold, a phenomenon known as universality (Hastie et al., 2022; Chen et al., 2023; Wu & Sun, 2025). To avoid introducing technical overhead, we adopt the Gaussian setting. Let $\boldsymbol{X} = (\boldsymbol{x}_1, \ldots, \boldsymbol{x}_n)^\top$ denote the feature matrix, $\boldsymbol{y} = (y_1, \ldots, y_n)^\top$ the response vector, and $\boldsymbol{\varepsilon} = (\varepsilon_1, \ldots, \varepsilon_n)^\top$ the noise vector. We work under the proportional asymptotic regime, where $d, n \to \infty$ with $d/n \to \gamma$ for some constant $\gamma$. This regime has been widely adopted in recent studies analyzing the exact risk behavior of linear models; see, e.g., Hastie et al. (2022); Chen et al. (2023); Wu & Sun (2025).

**Bagged least square estimator.** Following Wu & Sun (2025), we reformulate the bagged least square estimator as an average of sketched least square estimators. As described in Section 2.3, let $\boldsymbol{\mu}^{(b)}$ and $\boldsymbol{\nu}^{(b)}$ denote the instance and feature indices selected in the $b$-th subsample. Define the sketching matrices $\boldsymbol{U}_b \in \mathbb{R}^{n \times m}$ and $\boldsymbol{V}_b \in \mathbb{R}^{d \times s}$ as the corresponding selection matrices: $\boldsymbol{U}_b$ has ones at entries $(\boldsymbol{\mu}_i^{(b)}, i)$ for $i \in [m]$, and zeros elsewhere; similarly, $\boldsymbol{V}_b$ has ones at entries $(\boldsymbol{\nu}_j^{(b)}, j)$ for $j \in [s]$. The matrix $\boldsymbol{U}_b^\top \boldsymbol{X} \boldsymbol{V}_b \in \mathbb{R}^{m \times s}$ extracts a submatrix of $\boldsymbol{X}$ formed by the sampled rows and columns.

For each subsample, we compute the sketched minimum-norm least squares estimator

$$\boldsymbol{\beta}^{(b)} = \boldsymbol{V}_b \left(\boldsymbol{U}_b^\top \boldsymbol{X} \boldsymbol{V}_b\right)^\dagger \boldsymbol{U}_b^\top \boldsymbol{y},$$

where $(\cdot)^\dagger$ denotes the Moore–Penrose pseudoinverse. The final bagged estimator is obtained by averaging over $B$ such subsamples $\boldsymbol{\beta} := \frac{1}{B} \sum_{b=1}^B \boldsymbol{\beta}^{(b)}$.

**Affected-axis resampling coupling.** This section uses an affected-axis resampling coupling for the sketching randomness: after an instance or feature is removed, we redraw the sketching matrix on the perturbed axis from the reduced index set, while keeping the sketching matrix on the other

*Table 1.* Closed-form asymptotic limits for the four $\Delta_\ell$-terms in Theorem 3.1.

| Setting and Case | $\Delta_\ell^{V=}$ | $n\Delta_\ell^{V\neq}$ | $\Delta_\ell^{B=}$ | $n\Delta_\ell^{B\neq}$ |
|---|---|---|---|---|
| II ($\ell=-i$), $\gamma q < p$ | $\frac{2\gamma q(1-p)}{(p-\gamma q)(1-\gamma q)}$ | $\frac{\gamma q^2}{(1-\gamma q^2)^2}$ | $\frac{2\gamma q(1-q)(1-p)}{(p-\gamma q)(1-\gamma q)}$ | $\frac{\gamma q^2(1-q)^2}{(1-\gamma q^2)^2}$ |
| II ($\ell=-i$), $\gamma q > p$ | $\frac{2\gamma q(p-p^2)}{(\gamma q-p)(\gamma q-p^2)}$ | $\frac{\gamma p^2}{(\gamma-p^2)^2}$ | $\frac{2p(1-p)(\gamma^2 q-2\gamma pq+p^2)}{\gamma(\gamma q-p)(\gamma q-p^2)}$ | $\frac{p^2(\gamma-p)^2}{\gamma(\gamma-p^2)^2}$ |
| FI ($\ell=-j$), $\gamma q < p$ | $\frac{2\gamma pq(1-q)}{(p-\gamma q)(p-\gamma q^2)}$ | $\frac{q^2}{(1-\gamma q^2)^2}$ | $\frac{2pq(1-q)(p+\gamma-2\gamma q)}{(p-\gamma q)(p-\gamma q^2)}$ | $\frac{q^2(1+\gamma-2\gamma q)}{\gamma(1-\gamma q^2)^2}$ |
| FI ($\ell=-j$), $\gamma q > p$ | $\frac{2\gamma p(1-q)}{(\gamma q-p)(\gamma-p)}$ | $\frac{p^2}{(\gamma-p^2)^2}$ | $\frac{2p(1-q)}{\gamma q-p}$ | $\frac{p^2(1-2p+\gamma)}{(\gamma-p^2)^2}$ |

axis fixed. Thus the reduced run is neither a shared-seed run nor a fully independent rerun. After removing the $i$-th instance from the dataset, we independently resample a sketching matrix $\boldsymbol{U}_b'$ from the reduced index set $[n] \setminus \{i\}$. The resulting bagged estimator after instance removal is

$$\boldsymbol{\beta}^{-i,:} := \frac{1}{B} \sum_{b=1}^{B} \boldsymbol{V}_b \left( \boldsymbol{U}_b'^\top \boldsymbol{X} \boldsymbol{V}_b \right)^\dagger \boldsymbol{U}_b'^\top \boldsymbol{y}. \qquad (9)$$

Similarly, removing the $j$-th feature corresponds to resampling $\boldsymbol{V}_b'$ from $[d] \setminus \{j\}$, yielding

$$\boldsymbol{\beta}^{:,-j} := \frac{1}{B} \sum_{b=1}^{B} \boldsymbol{V}_b' \left( \boldsymbol{U}_b^\top \boldsymbol{X} \boldsymbol{V}_b' \right)^\dagger \boldsymbol{U}_b^\top \boldsymbol{y}. \qquad (10)$$

### 3.2. Stability Analysis

Under this affected-axis resampling coupling, we study the expected instability measures $\frac{1}{n} \sum_{i=1}^{n} \mathbb{E}_{\boldsymbol{x},\boldsymbol{\beta},\boldsymbol{\beta}^{-i,:}}[(\boldsymbol{\beta}^\top \boldsymbol{x} - (\boldsymbol{\beta}^{-i,:})^\top \boldsymbol{x})^2]$ for II and $\frac{1}{d} \sum_{j=1}^{d} \mathbb{E}_{\boldsymbol{x},\boldsymbol{\beta},\boldsymbol{\beta}^{:,-j}}[(\boldsymbol{\beta}^\top \boldsymbol{x} - (\boldsymbol{\beta}^{:,-j})^\top \boldsymbol{x})^2]$ for FI. Since the features are isotropic, taking expectation over $\boldsymbol{x}$ reduces both quantities to

$$\text{Instability} = \mathbb{E}\left[ \|\boldsymbol{\beta} - \boldsymbol{\beta}^\ell\|_2^2 \right], \quad \ell \in \{-i, -j\}, \qquad (11)$$

where the subscript of $\mathbb{E}_{\boldsymbol{\beta},\boldsymbol{\beta}^\ell}$ is omitted for brevity, $\ell = -i$ denotes $i$-th instance removal, and $\ell = -j$ denotes $j$-th feature removal. We slightly abuse notation by writing $\boldsymbol{\beta}^\ell$ for either $\boldsymbol{\beta}^{-i,:}$ or $\boldsymbol{\beta}^{:,-j}$. Rather than upper bounding this instability, we characterize the exact limiting value of (11) under proportional asymptotics. Our first result gives a finite-$B$ decomposition and the corresponding asymptotic limits.

**Theorem 3.1.** *Assume the Gaussian linear model in Section 3.1. Then, for $\ell \in \{-i, -j\}$,*

$$\mathbb{E}\big[\|\boldsymbol{\beta} - \boldsymbol{\beta}^\ell\|_2^2\big] = \sigma^2 \left( \frac{\Delta_\ell^{V=}}{B} + \frac{B-1}{B} \Delta_\ell^{V\neq} \right) \qquad (12)$$
$$+ \frac{\Delta_\ell^{B=}}{B} + \frac{B-1}{B} \Delta_\ell^{B\neq}.$$

*Moreover, as $n, d \to \infty$ with $d/n \to \gamma$,*

$$\Delta_\ell^{V=}, \Delta_\ell^{B=} \to \Theta(1), \text{ and } \Delta_\ell^{V\neq}, \Delta_\ell^{B\neq} \to \Theta(\tfrac{1}{n}), \qquad (13)$$

*where $\Theta(\cdot)$ denotes asymptotic equality up to constants. The exact asymptotic limits of $\Delta_\ell^{V=}$, $n\Delta_\ell^{V\neq}$, $\Delta_\ell^{B=}$, and $n\Delta_\ell^{B\neq}$ are given in Table 1 for instance removal and feature removal.*

The proof of Theorem 3.1 is given in Appendix D. The theorem decomposes the expected instance and feature instability of bagged least-squares estimators into noise variance terms ($V$) and signal-bias terms ($B$). The $=$ terms compare paired estimators built from the same subsampling realization and receive weight $1/B$, whereas the $\neq$ terms compare different submodels in the ensemble and receive weight $(B-1)/B$.

The decomposition gives three takeaways. First, it explains why bagging stabilizes the subsampled least-squares estimator. As $B$ grows, the $1/B$-weighted same-submodel terms vanish, and the cross terms $\Delta_\ell^{V\neq}$ and $\Delta_\ell^{B\neq}$ dominate. These cross terms are of order $1/n$, giving the dashed large-$B$ curves in Figure 2. Thus, compared with a single subsampled estimator in Appendix B, bagging preserves the qualitative double-descent shape but removes the interpolation divergence by averaging over different submodels, consistent with Wu & Sun (2025).

Second, the closed forms show that feature removal has a different natural scale from instance removal. In the large-$B$ limit, most cross terms are reported on the $1/n$ scale, but the feature-removal bias term also reflects the number of available coordinates. Specifically, in the underparameterized regime $\gamma q < p$,

$$n\Delta_{-j}^{B\neq} \to \frac{q^2(1+\gamma-2\gamma q)}{\gamma(1-\gamma q^2)^2} \sim \frac{q^2}{\gamma} \quad (\gamma \downarrow 0).$$

Equivalently, since $d = \gamma n$, the unscaled term behaves like $\Delta_{-j}^{B\neq} \asymp q^2/d$. Thus this part of feature instability is controlled by the feature dimension, not only by the sample size. This explains why feature removal is more consequential when $d$ is small relative to $n$, while the same feature-specific contribution vanishes in the high-dimensional regime $\gamma \to \infty$.

Third, the same decomposition explains the role of the ensemble size $B$, as shown in Figure 2. For small $B$, the $1/B$-weighted same-submodel terms $\Delta_\ell^{V=}$ and $\Delta_\ell^{B=}$ re-

main visible and inherit the instability of individual subsampled estimators. Increasing $B$ transfers mass from these terms to the cross terms, producing an explicit averaging gain. Once $B$ is of order $n$, the instability is already near its order-$1/n$ large-$B$ limit, so further increases in $B$ give diminishing returns except near interpolation, where larger ensembles are still useful for flattening the peak. Finally, the closed forms show that, for fixed $\gamma$, smaller instance or feature subsampling ratios improve the large-$B$ limiting stability for both II and FI.

# 4. Model-free Feature Stability

In this section, we investigate FI under model-free settings. We first develop a general-space feature-stability tool, with the scalar $\mathbb{R}$ result appearing as a special case (Section 4.1), and then extend it to recursive subsampling (Section 4.2). These results are applied to random forward selection and random forests in Section 4.2.1 and Appendix E.1, and extended to finite bagging rounds in Appendix E.3. Unlike linear regression analysis in Section 3, the model-free results use the shared-randomness, conditional-on-the-dataset coupling in Definitions 2.1–2.2, namely $\xi^j = \xi$. The aim is therefore a uniform algorithmic guarantee rather than an exact distributional perturbation calculation.

## 4.1. Model-free Stability in General Spaces

This subsection develops the general-space stability tool used later for recursive feature subsampling, used in random forests (Breiman, 2001; Zhang et al., 2024). As illustrated in Figure 1(d), each tree uses a fixed subset of instances, while features are subsampled recursively at each node. This recursive, data-dependent feature subsampling makes feature-bagging analysis more challenging than instance-bagging analysis. To isolate the feature-subsampling effect, we focus on FI and first present the scalar $\mathbb{R}$ guarantee before giving its Hilbert-space generalization.

**Proposition 4.1** (FI of feature bagging). *Assume the base algorithm $\mathcal{A}_0$ produces predictors satisfying $f(\boldsymbol{x}) \in \mathcal{Y} \subseteq [-M, M]$ for every $\boldsymbol{x} \in \mathcal{X}$, and let $q < 1$. Then the infinitely feature-bagged algorithm $\mathcal{A}_\infty$ based on $\mathcal{A}_0$ is $\phi$-feature stable whenever $\phi^2 \geq \frac{M^2}{d-1} \frac{q}{1-q}$.*

Proposition 4.1 is the feature-wise dual of the infinite instance-bagging guarantee of Soloff et al. (2024a). Under the same bounded-output assumption, their result states that infinite instance bagging with subsampling ratio $p < 1$ is $\phi$-instance stable whenever $\phi^2 \geq \frac{M^2}{n-1} \frac{p}{1-p}$. Thus, their guarantee controls II under instance subsampling, whereas Proposition 4.1 controls FI under feature subsampling; formally, $(n, p, \mathrm{II})$ is replaced by $(d, q, \mathrm{FI})$. This feature-side guarantee is the scalar starting point for the general-space and recursive results below.

We now lift the scalar feature-bagging result to general output spaces. Consider algorithms of the form $\boldsymbol{w} = \mathcal{A}(\mathcal{D}, \xi)$ whose outputs lie in a Hilbert space $\mathcal{H}$ with inner product $\langle \cdot, \cdot \rangle$ and induced norm $\| \cdot \|_{\mathcal{H}}$. We assume that the output belongs to a convex, bounded set $\mathcal{W} \subset \mathcal{H}$ and that the zero element of $\mathcal{H}$ lies in $\mathcal{W}$. The associated feature instability over $\mathcal{W}$ is measured using $\| \cdot \|_{\mathcal{H}}$.

**Definition 4.2.** An algorithm $\mathcal{A}$ with output in $\mathcal{W}$ is $\phi$-feature stable if, for all datasets $\mathcal{D} = \{(\boldsymbol{x}_i, y_i)\}_{i=1}^n$,

$$\mathrm{FI}_{\mathcal{H}} := \frac{1}{d} \sum_{j=1}^d \mathbb{E}\left[\|\boldsymbol{w} - \boldsymbol{w}^{-j}\|_{\mathcal{H}}^2\right] \leq \phi^2,$$

where $\boldsymbol{w} = \mathcal{A}(\mathcal{D}; \xi)$ and $\boldsymbol{w}^{-j} = \mathcal{A}(\mathcal{D}^{:, -j}; \xi)$.

The bagged estimator is defined analogously. Let $\boldsymbol{w}^{(b)} = \mathcal{A}(\mathcal{D}, \xi^{(b)}) = \mathcal{A}_0(\mathcal{D}^{:, \boldsymbol{\nu}^{(b)}})$ be the output of the $b$-th feature subsample. Since $\mathcal{W}$ is convex, $\boldsymbol{w}^B = B^{-1} \sum_{b=1}^B \boldsymbol{w}^{(b)}$ also lies in $\mathcal{W}$. The infinite-bagged estimator is $\boldsymbol{w}^\infty := \mathbb{E}[\mathcal{A}(\mathcal{D}, \xi)]$. This framework extends the scalar case $\mathcal{W} \subseteq \mathbb{R}$ above. We use the following Hilbert-space analogue of the bounded-range assumption of Soloff et al. (2024c).

**Assumption 4.3.** Let $\mathcal{W}$ be a closed and convex subset of a Hilbert space with norm $\| \cdot \|_{\mathcal{H}}$. We assume that its radius $\mathrm{rad}(\mathcal{W}) := \inf_{\boldsymbol{w} \in \mathcal{W}} \sup_{\boldsymbol{w}' \in \mathcal{W}} \|\boldsymbol{w} - \boldsymbol{w}'\|_{\mathcal{H}}$ is finite.

When $\mathcal{H} = \mathbb{R}$, $\| \cdot \|_{\mathcal{H}} = |\cdot|$, and $\mathcal{W} = [-M, M]$, Assumption 4.3 recovers the bounded-range condition used in Proposition 4.1.

**Proposition 4.4** (General-space FI of feature bagging). *Assume Assumption 4.3 holds, and let $q < 1$. For any base algorithm $\mathcal{A}_0$ with output in $\mathcal{W}$, infinite feature bagging $\mathcal{A}_\infty$ is $\phi$-feature stable whenever*

$$\phi^2 \geq \frac{\mathrm{rad}^2(\mathcal{W})}{d-1} \frac{q}{1-q}.$$

Proposition 4.4 generalizes Proposition 4.1 by replacing the scalar range bound $M$ with the Hilbert-space radius $\mathrm{rad}(\mathcal{W})$. This general-space form is the key input for the recursive procedures analyzed next, including random forward selection and random forests.

## 4.2. Recursive Subsampling

The previous subsection studies feature bagging where each base learner only subsamples features once. Recursive algorithms is more complex: each base learner performs multiple rounds of feature subsampling, and the subsampled features at each round are data-dependent. This recursive structure makes the analysis more challenging because the effect from one feature removal can propogate in later updates. We model this propagation through a stochastic process

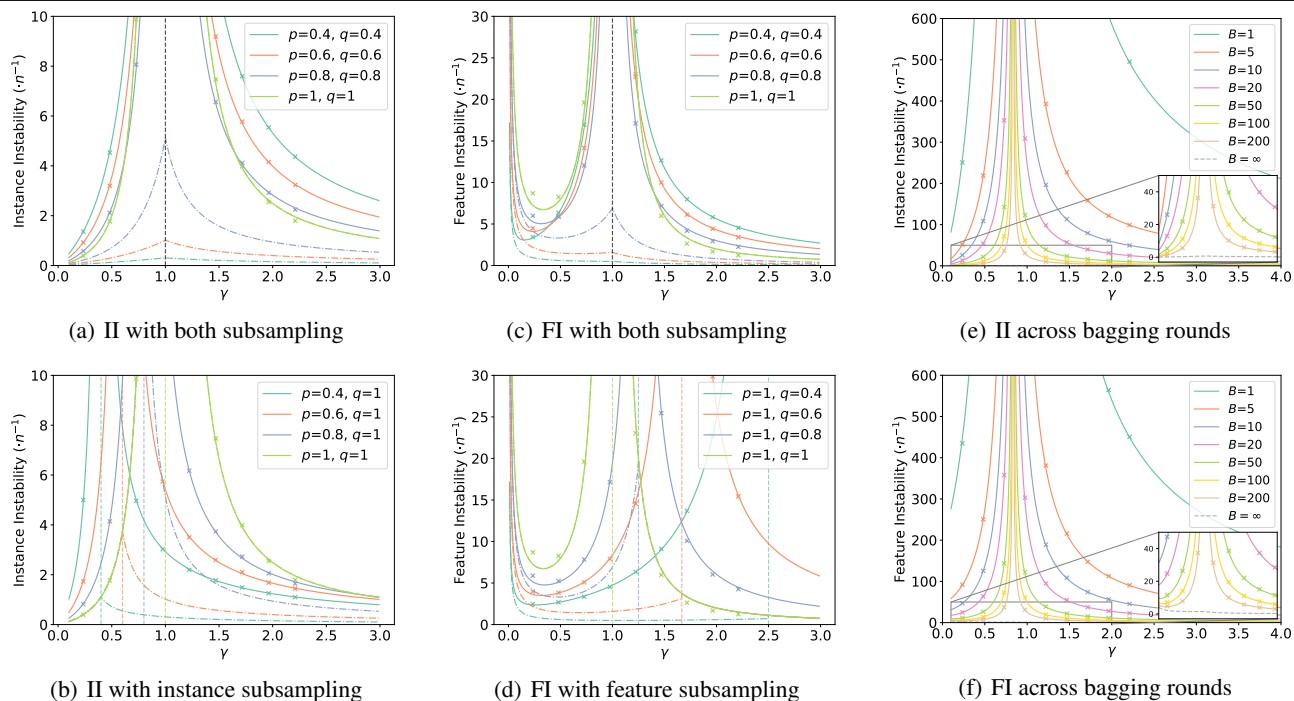

*Figure 2.* Instability $\mathbb{E}[\|\boldsymbol{\beta} - \boldsymbol{\beta}^\ell\|_2^2]$ of the bagged least square estimator. Panels (a)–(d) use $B = 200$ bagging rounds and vary the aspect ratio $\gamma$; solid lines denote theoretical predictions, crosses mark empirical averages, and dashed curves show the infinite-bagging limit. Panels (e)–(f) compare different numbers of bagging rounds on an $n^{-1}$ scale; colors indicate $B$, solid lines denote theoretical predictions, and crosses mark empirical averages. All panels use the same simulation setting as in Appendix B.

$\{\boldsymbol{w}_t\}_{t\geq 0}$, where each state $\boldsymbol{w}_t$ lies in a set $\mathcal{W}_t \subseteq \mathcal{H}$ and $\mathcal{H}$ is equipped with the norm $\|\cdot\|_{\mathcal{H}}$. At iteration $t$, the update is

$$\boldsymbol{w}_t = \mathcal{A}(\mathcal{D}, (\boldsymbol{w}_{t-1}, \xi_t)),$$

where $\xi_t$ denotes the randomness injected at step $t$. The admissible set $\mathcal{W}_t$ may depend on $\boldsymbol{w}_{t-1}$, allowing the state space itself to be data-dependent and recursive. We initialize the process at $\boldsymbol{w}_0$, the zero element of $\mathcal{H}$, and assume that $\boldsymbol{w}_0 \in \mathcal{W}_t$ for all $t$.

For a fixed dataset, define the update maps

$$S_\xi(\boldsymbol{w}) := \mathcal{A}(\mathcal{D}, (\boldsymbol{w}, \xi)), \quad S_\xi^{-j}(\boldsymbol{w}) := \mathcal{A}(\mathcal{D}, (\boldsymbol{w}, \xi, j)).$$

Here the extra argument $j$ means that the same seed $\xi$ first generates the candidate feature set as in the original run, and then feature $j$ is removed from that set if it appears. Using the same randomness $\xi_{1:T}$, the original trajectory is

$$\mathcal{A}^{(T)}(\mathcal{D}, \xi_{1:T}) = \boldsymbol{w}_T = S_{\xi_T} \circ \cdots \circ S_{\xi_1}(\boldsymbol{w}_0), \quad (14)$$

whereas the trajectory after removing feature $j$ is

$$\mathcal{A}^{(T)}(\mathcal{D}, (\xi_{1:T}, j)) = \boldsymbol{w}_T^{-j} = S_{\xi_T}^{-j} \circ \cdots \circ S_{\xi_1}^{-j}(\boldsymbol{w}_0). \quad (15)$$

Independent copies of this recursive trajectory give the finite- and infinite-bagged outputs

$$\mathcal{A}_B^{(T)}(\mathcal{D}) = \frac{1}{B}\sum_{b=1}^{B} \boldsymbol{w}_T^{(b)}, \quad \mathcal{A}_\infty^{(T)}(\mathcal{D}) = \mathbb{E}_{\xi_{1:T}}[\boldsymbol{w}_T],$$

with their feature-removed counterparts denoted by a superscript $-j$. This framework covers random forward selection (Mentch & Zhou, 2020) and random forests (Breiman, 2001), both described in Appendix E.4.

To state the recursive guarantee, let $\Delta_t^{-j} := \boldsymbol{w}_t - \boldsymbol{w}_t^{-j}$. We isolate the part of the step-$t$ perturbation that comes from propagating the previous feature-removal discrepancy: $\mathrm{pith}_t^j := \mathbb{E}[S_{\xi_t}^{-j}(\boldsymbol{w}_{t-1}) - S_{\xi_t}^{-j}(\boldsymbol{w}_{t-1}^{-j})]$. The following one-step condition requires this propagated discrepancy to grow by at most a factor $1+\delta_t$: for each step $t$, there exists $\delta_t \geq 0$ such that, for every $j \in [d]$,

$$\left\|\mathrm{pith}_t^j\right\|_{\mathcal{H}} \leq (1+\delta_t)\left\|\mathbb{E}\left[\Delta_{t-1}^{-j}\right]\right\|_{\mathcal{H}}. \quad (16)$$

This is a Lipschitz-type bound on the expected transition map, analogous to stepwise stability assumptions for iterative algorithms (Hardt et al., 2016; Lei & Ying, 2020).

**Proposition 4.5.** *Let Assumption 4.3 hold. Suppose that for each recursion step $t$, the one-step condition (16) holds for every $j \in [d]$. Then*

$$\frac{1}{d}\sum_{j=1}^{d}\left\|\mathbb{E}[\boldsymbol{w}_T] - \mathbb{E}\left[\boldsymbol{w}_T^{-j}\right]\right\|_{\mathcal{H}}^2 \leq \frac{M_T q}{(d-1)(1-q)}, \quad (17)$$

*where*

$$M_T := \left(\sum_{t=1}^{T}\left(\prod_{t'=t+1}^{T}(1+\delta_{t'})\right)\sup_{\xi_{1:(t-1)}} \mathrm{rad}(\mathcal{W}_t)\right)^2.$$

The bound preserves the same feature-subsampling factor $q/((d-1)(1-q))$ as the one-step[1] result in Proposition 4.4. The new factor $M_T$ is the price of recursion: it accumulates the radii of the intermediate state spaces and the one-step inflation factors. In the simple case $\mathrm{rad}(\mathcal{W}_t) \equiv r$ and $\delta_t \equiv \delta$, we obtain $M_T = O(r^2 T^2)$ when $\delta = 0$, and $M_T = O(r^2(1+\delta)^{2T})$ for fixed $\delta > 0$.

Proposition 4.5 is stated for infinite bagging to isolate the mechanism that improves feature stability. Proposition E.3 in Appendix E.11 shows that using $B$ finite bagging rounds adds only an $O(1/B)$ concentration error to the infinite-bagging instability term. Thus, for moderately large $B$, finite bagging inherits the same stability behavior up to this vanishing error.

### 4.2.1. RANDOM FORWARD SELECTION

We first apply the recursive bound to random forward selection. The algorithm builds a feature set one coordinate at a time: at each step, it samples a fraction $q$ of the available features and adds the sampled feature that gives the largest reduction in residual error. This places random forward selection within the class of data-dependent recursion algorithms covered by Proposition 4.5.

To represent the recursive state, encode the selected variables by $\boldsymbol{w} = (w^1, \ldots, w^d)$, where $w^j \in \{0,1\}$ and $w^j = 1$ means that feature $j$ is included in the fitted linear model. In this specialization, $\mathcal{W}_t \subseteq \{0,1\}^d \subset \mathcal{H}$, with $\mathcal{H} = \mathbb{R}^d$ and $\|\cdot\|_{\mathcal{H}} = \|\cdot\|_2$. For feature-subsampled ordinary least squares, $w^j = 1$ exactly when $j \in \boldsymbol{\nu}$. Let $\boldsymbol{E_w}$ denote the diagonal matrix with diagonal $\boldsymbol{w}$. The $b$-th subsampled ordinary least-squares estimator is

$$\boldsymbol{\beta}^{(b)} = \boldsymbol{E_{w^{(b)}}} \left(\boldsymbol{X E_{w^{(b)}}}\right)^\dagger \boldsymbol{y}.$$

where the inverse is understood as the Moore–Penrose pseudoinverse in the overparameterized regime. Let $\boldsymbol{\beta} = \boldsymbol{X}^\dagger \boldsymbol{y}$.

We consider the orthogonal-design case, where $\boldsymbol{X}^\top \boldsymbol{X} = n\boldsymbol{I}_d$. Then the feature-subsampled estimator reduces to $\boldsymbol{\beta}^{(b)} = \boldsymbol{E_{w^{(b)}}} \boldsymbol{\beta}$ and, with $\boldsymbol{w}_B := B^{-1} \sum_{b=1}^B \boldsymbol{w}^{(b)}$,

$$\boldsymbol{\beta}_B = \frac{1}{B} \sum_{b=1}^B \boldsymbol{\beta}^{(b)} = \frac{1}{B} \sum_{b=1}^B \boldsymbol{E_{w^{(b)}}} \boldsymbol{\beta} = \boldsymbol{E_{w_B}} \boldsymbol{\beta}.$$

Thus, in the orthogonal setting, the stability of the averaged estimator is governed by the selection-frequency vector $\boldsymbol{w}_B$. Under uniform feature subsampling, each feature is selected with probability $q$, so $\mathbb{E}[\boldsymbol{w}_B] = (q, \ldots, q)$. Under random forward selection, $\boldsymbol{w}_B$ is data-dependent and records how frequently each feature is selected, thereby capturing feature importance. We measure discrepancies between selection

---

[1]Here one step means that each base learner only subsamples features once.

vectors by the $\ell_2$ distance. The following theorem bounds the resulting feature instability.

**Theorem 4.6.** *Let $\mathcal{A}$ be the random forward selection algorithm in Appendix E.4 with feature-subsampling ratio $q < 1$. Then $\mathrm{rad}(\mathcal{W}_t) \leq 1$ for $t \in [T]$. Moreover, the one-step condition* (16) *in Proposition 4.5 holds with $\delta_t = 4^{-1} \max\{q(1+q), q^2 t\}$. When $T \lesssim q^{-1}$, we have*

$$\frac{1}{d} \sum_{j=1}^d \left\| \mathbb{E}\left[\boldsymbol{w}_T\right] - \mathbb{E}\left[\boldsymbol{w}_T^{-j}\right] \right\|^2 \lesssim \frac{T^2}{d-1} \cdot \frac{q}{1-q}. \quad (18)$$

The theorem specializes Proposition 4.5 to the selection path of random forward selection. The bound in (18) gives a stronger stability guarantee for smaller feature-subsampling ratios $q$, showing that more aggressive feature subsampling improves stability. Its quadratic dependence on $T$ reflects the other side of the recursive analysis: instability can accumulate along a long selection path. For comparison, deterministic forward selection without bagging has feature instability

$$\frac{1}{d} \sum_{j=1}^d \left\| \mathbb{E}\left[\boldsymbol{w}_T\right] - \mathbb{E}\left[\boldsymbol{w}_T^{-j}\right] \right\|^2 = \frac{2T}{d},$$

whose derivation is deferred to the appendix. Hence, feature bagging is provably more stable whenever

$$\frac{2T}{d} \gtrsim \frac{T^2}{d-1} \cdot \frac{q}{1-q}, \quad \text{or equivalently,} \quad T \lesssim \frac{1-q}{q}.$$

This is consistent with the condition $T \lesssim q^{-1}$; for example, it is satisfied when $T = o(\sqrt{d})$ and $q = 1/\sqrt{d}$.

Figure 3 evaluates this prediction under the experimental settings detailed in Appendix E.2. Figure 3(a) shows that feature bagging consistently improves empirical feature stability, with smaller subsampling ratios $q$ giving larger stability gains. Figure 3(b) shows a corresponding decrease in parameter estimation difference, suggesting that the stability gain does not come at the expense of estimation accuracy.

When $qT \leq 1$, the theoretical curve upper bounds the averaged empirical instability and certifies better stability than the non-bagged baseline. For small $T$, it also tracks the empirical curve closely, indicating that the bound captures the early-stage behavior of random forward selection. As $T$ grows, the gap widens, consistent with the quadratic dependence on $T$ and the conservative one-step condition used in the analysis. Overall, the experiment supports the same message as the theorem: feature bagging stabilizes random forward selection most clearly in the shallow-recursion regime.

### 4.2.2. RANDOM FORESTS

Random forests provide the second application. In a full classification-and-regression tree, both the split feature and

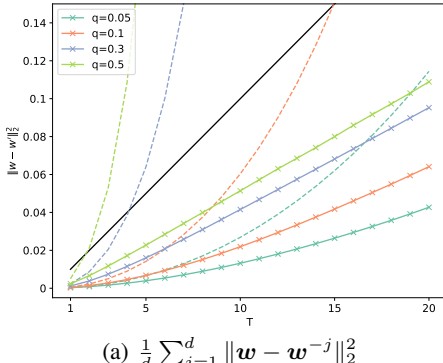

(a) $\frac{1}{d}\sum_{j=1}^{d}\|\boldsymbol{w}-\boldsymbol{w}^{-j}\|_2^2$

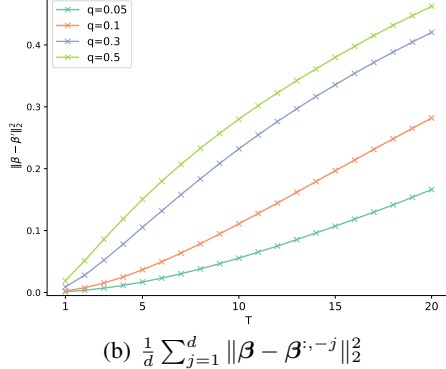

(b) $\frac{1}{d}\sum_{j=1}^{d}\|\boldsymbol{\beta}-\boldsymbol{\beta}^{\cdot,-j}\|_2^2$

*Figure 3.* Left: Feature instability of feature-bagged random forward selection and non-bagged baseline (forward selection). Solid curves with crosses show empirical averages over 100 repetitions for different $q$, dashed curves show the upper bound in (18), and the solid black curve gives the non-bagged baseline. Right: Parameter estimation difference of feature-bagged random forward selection for different $q$.

the split threshold are data-dependent. we analyze a simplified dyadic forest: each split is made at the midpoint of the selected coordinate, so the local cell containing a fixed test point $\boldsymbol{x}$ is determined only by the sequence of split features along the path to $\boldsymbol{x}$.

For this local path, we encode split usage by a binary matrix $\boldsymbol{w}_T \in [0,1]^{T\times d}$, written as $\boldsymbol{w}_T = (\boldsymbol{w}_T^1, \ldots, \boldsymbol{w}_T^d)$. If feature $j$ is used $r$ times along the path to $\boldsymbol{x}$, then the first $r$ entries of the $j$-th column of $\boldsymbol{w}_T$ are one and the remaining entries are zero. Thus, $\|\boldsymbol{w}_T^j\|_1$ records the number of path splits using feature $j$. Appendix E.1 gives the precise construction and connects it to prediction. Under the additive model considered there, the conditional expected prediction is an affine function of $\boldsymbol{w}_T$, so controlling the perturbation of $\boldsymbol{w}_T$ also controls the corresponding prediction perturbation.

**Theorem 4.7.** *Let $\mathcal{A}$ be the max-edge dyadic random forest algorithm described in Appendix E.1. Then $\mathrm{rad}(\mathcal{W}_t) \leq 1$, and the one-step condition (16) holds with $\delta_t = \sqrt{2}$. Let $A_T = \sum_{t=1}^{T}(1+\sqrt{2})^{T-t}$. Then*

$$\frac{1}{d}\sum_{j=1}^{d}\left\|\mathbb{E}\left[\boldsymbol{w}_T\right]-\mathbb{E}\left[\boldsymbol{w}_T^{-j}\right]\right\|_{\mathrm{F}}^2 \leq \frac{q \cdot A_T^2}{(d-1)(1-q)}. \quad (19)$$

Theorem 4.7 shows that recursive feature bagging stabilizes the local partition induced by a dyadic tree. Its dependence on the feature-subsampling ratio matches the one-step bound in Proposition 4.4: smaller $q$ yields a stronger feature-stability guarantee. The factor $A_T^2$ is the price of recursion, capturing how perturbations accumulate along the tree path. For comparison, a non-bagged depth-$T$ dyadic tree has feature instability $2T/d$, as derived in Appendix E.7. Thus, for shallow trees and sufficiently small $q$, feature bagging provides a strict stability improvement.

## 5. Conclusion

We showed that feature bagging provides a principled route to algorithmic stabilization. By introducing FI as the feature-side counterpart to II, we made it possible to study how bagging improves stability along both the instance and feature axes. In linear regression, we derived sharp asymptotic characterizations of II and FI; in a model-free setting inspired by recursive feature subsampling in random forests, we obtained tight upper bounds for FI in early steps. Together, these results show that feature bagging consistently improves stability, that more aggressive feature subsampling can yield stronger stabilization, and that a modest number of bagging rounds is enough to approach the infinite-bagging stability level.

## Acknowledgments

We thank the anonymous reviewers for their constructive comments and suggestions, which have helped improve the clarity and presentation of this work. The bulk of the work was carried out while YM was a visiting student at MBZUAI and University of Toronto. QS was partially supported by NSERC Grant RGPIN-2026-06888, Compute Canada, and MBZUAI.

## Impact Statement

This paper presents work whose goal is to advance the field of Machine Learning. There are many potential societal consequences of our work, none which we feel must be specifically highlighted here.

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

## A. Related Works

This appendix provides literature reviews.

**Stability.** As noted by Soloff et al. (2024a), the definitions of "sensitivity" and "small changes" vary across the literature. Following Bousquet & Elisseeff (2002); Elisseeff et al. (2005), many studies define a small change as the removal of a single instance (Kutin & Niyogi, 2002; Mukherjee et al., 2006; Liu et al., 2017). This notion is particularly convenient for deriving generalization bounds (Bousquet & Elisseeff, 2002; Feldman & Vondrak, 2018; 2019; Bousquet et al., 2020). A related notion is averaging stability (Shalev-Shwartz et al., 2010; Lei & Ying, 2020; Soloff et al., 2024a), which measures the average sensitivity over all possible single-sample removals. Recent works including Adrian et al. (2024); Soloff et al. (2024b;a;c); Liang et al. (2025) have used this definition to study the stability of bagging methods.

Throughout this paper, we primarily focus on the instance-instability measure defined by comparing the prediction losses using the original dataset $\mathcal{D}$ and the same dataset but without the $i$-th instance $\mathcal{D}^{-i,:}$. An alternative notion (Bousquet & Elisseeff, 2002; Soloff et al., 2024a) is by comparing the prediction losses of the algorithms based on $\mathcal{D}$ and the same dataset but with the $i$-th instance replaced with an independent copy $\mathcal{D}^{(i),:}$. This alternative notion is weaker than the one defined in (3), in the sense that it can be bounded by twice the value of (3). This implication follows by relating $\mathcal{D}$ to $\mathcal{D}^{-i,:}$, and subsequently $\mathcal{D}^{-i,:}$ to $\mathcal{D}^{-i,:} \cup \{(\boldsymbol{x}_i', y_i')\}$. The same convention applies along the feature axis. We use removal as the primary definition because feature replacement requires specifying how the replaced coordinate is generated, for example unconditionally or conditionally on $(Y_i, \boldsymbol{x}_i^{-j})$. Replacement-based feature perturbations can still be related to removal-based ones by the same triangle-inequality argument.

**Bagging.** Instance bagging is one of the most widely adopted paradigms in machine learning, with applications across tree-based models (Breiman, 1996; Geurts et al., 2006), nearest neighbors (Biau et al., 2010; Cai et al., 2025), and neural networks (Hansen & Salamon, 1990; Perrone & Cooper, 1995; Lakshminarayanan et al., 2017; Zaidi et al., 2021). In contrast, feature bagging has been predominantly employed in tree-based methods (Breiman, 1996; Geurts et al., 2006). Recent high-dimensional linear-model studies provide a complementary view of subsampling, regularization, and ensembling: sketching can shift interpolation thresholds and stabilize risk curves (Chen et al., 2023), reference-panel regularization admits exact high-dimensional risk characterizations (Su et al., 2024), and bagged linear interpolators reveal how ensembling controls interpolation-driven variance through implicit regularization (Wu & Sun, 2025).

## B. Additional Linear-Model Subsampling Results

This appendix collects the single-submodel subsampling baseline and the subsampling-specific discussion deferred from Section 3.

Without bagging ($B = 1$), the relevant instability measure is that of a single subsampled estimator and exhibits the classical double-descent phenomenon for both II and FI, as shown in Figure 4. The instability diverges at the interpolation threshold $\gamma q = p$, where the subsampled design matrix becomes nearly singular. Changing the subsampling ratios $p$ or $q$ shifts this threshold, but does not remove the divergence. This is the main contrast with the bagged curves in Figure 2, where ensemble averaging suppresses the interpolation peak.

The explicit formulas in Table 1 also reveal how the large-$B$ limit depends on the subsampling ratios. For fixed $\gamma$, more aggressive instance or feature subsampling yields stronger asymptotic stability in the bagged regime. Taken together, these observations separate the two effects: subsampling controls where the interpolation behavior appears, whereas bagging controls whether the associated instability peak persists.

Classical stability-based generalization bounds (Bousquet & Elisseeff, 2002; Elisseeff et al., 2005; Mukherjee et al., 2006) do not directly apply to linear regression, due to its unbounded output space. Nonetheless, the generalization-error curves observed in (LeJeune et al., 2020; Wu & Sun, 2025) display notable qualitative similarities with our instability curves. In particular, both exhibit a double-descent-like phenomenon around the interpolation threshold. In both cases, subsampling shifts the threshold, while bagging suppresses the peak and flattens the curve, rendering the instability bounded.

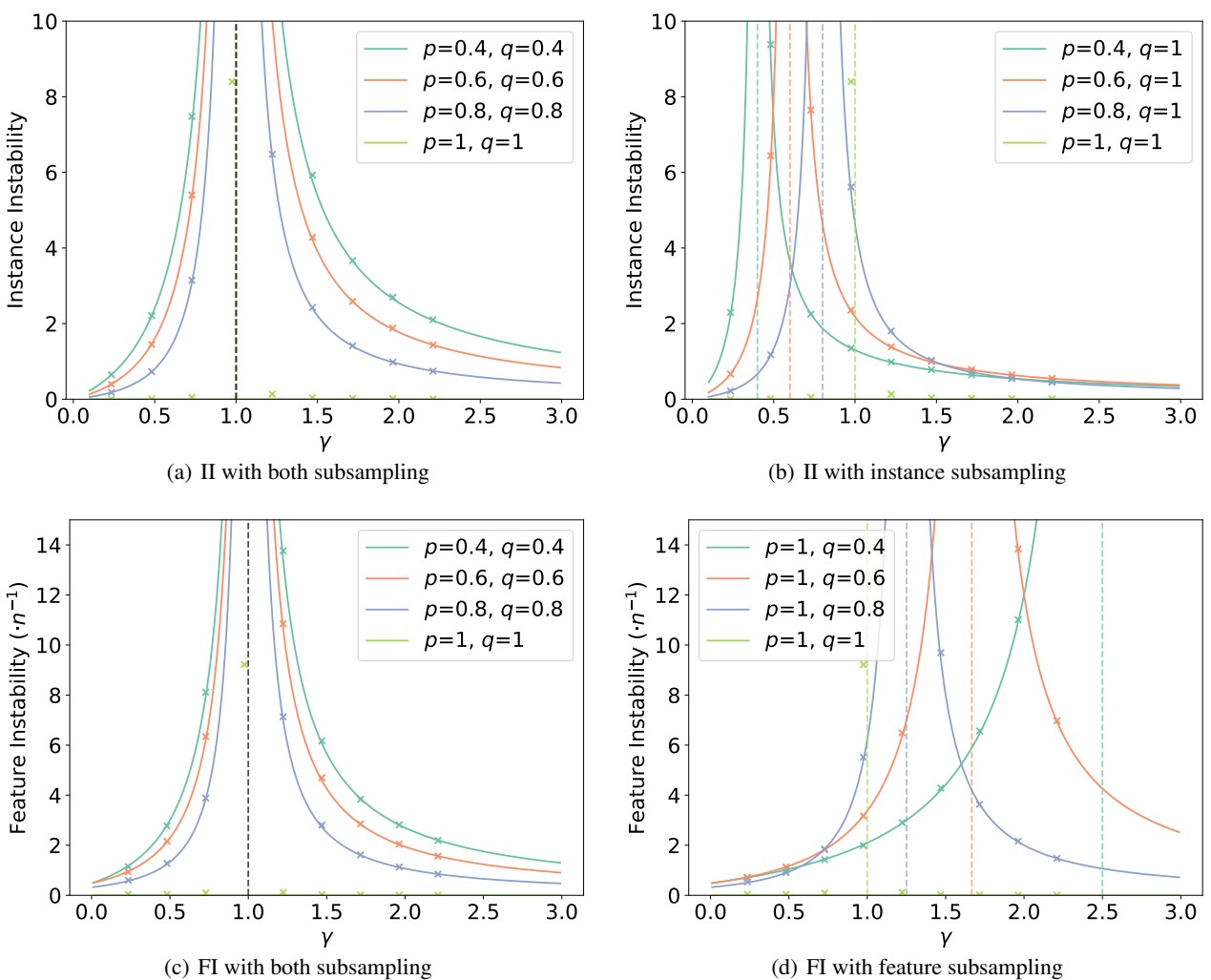

*Figure 4.* Instability of the single subsampled minimum-norm OLS estimator as a function of the aspect ratio $\gamma$. Colors indicate subsampling ratios $(p, q)$; solid lines denote theoretical predictions and crosses denote empirical averages over 300 realizations with $\sigma = 1$, $n = 400$, and $d \in \{100, 200, \ldots, 900\}$. Vertical dashed lines indicate the interpolation threshold $\gamma q = p$.

## C. Experiment Details of PMI between Feature Instability and Generalization

This appendix provides full experimental details and quantitative analyses supporting Section 2.2.

### C.1. Data Generating Process

We adopt the MARSadd setting from Friedman (1991), a standard benchmark for evaluating random forests (Mentch & Zhou, 2020; Curth et al., 2024). For each experiment, we independently sample $X \in [0,1]^{n \times d}$ from the uniform distribution and generate responses according to

$$f(x) = (1 - \nu)f_{\text{weak}}(x) + \nu f_{\text{strong}}(x),$$

where

$$f_{\text{weak}}(x) = \frac{1}{\sqrt{d-4}} \sum_{j=5}^{d} x_j$$

represents a linear but low signal-to-noise component, in which the signal is distributed across many features. Let

$$f_{\text{strong}}(x) = 0.1e^{4x_1} + \frac{4}{1 + e^{-20x_2 + 10}} + 3x_3 + 2x_4$$

represents a concentrated but nonlinear signal. Independent Gaussian noise $\epsilon \sim \mathcal{N}(0, \sigma^2)$ is added to form $y = f(x) + \epsilon$. An independent test set of equal size is generated using the same procedure. The mixing parameter $\nu \in [0,1]$ controls the dominance of nonlinear structure.

### C.2. Model Training and Instability Estimation

The random forest models were trained over a predefined hyperparameter grid to systematically explore the effects of model complexity and ensemble diversity. The number of estimators was fixed at $n_{\text{estimators}} = 256$. To control tree diversity, the maximum feature ratio (`max_features`) and the maximum sample ratio (`max_samples`) were varied over $\{0.1, 0.3, 0.5, 0.7\}$. Model complexity was adjusted by varying the maximum tree depth (`max_depth`) from 1 to 7, while the minimum number of samples per leaf (`min_samples_leaf`) was fixed at 1.

This design enables a systematic investigation of model complexity along both the feature-subsampling and depth dimensions. In particular, `max_features` and `max_samples` control the diversity of individual trees, whereas `max_depth` governs the bias-variance trade-off of the ensemble. Fixing the number of estimators improves comparability across runs.

For each configuration of hyperparameters {max_features, max_samples, max_depth}, we train a RandomForestRegressor on the training data and compute both training and test mean squared errors (MSE). Both instance and feature instability metrics are then measured as in (1) and (2), by retraining the model on the data with the removed feature or instance.

All configurations are repeated 500 times with different random seeds, ensuring robust Monte Carlo estimates. For each unique hyperparameter configuration, we compute the mean instability scores, mean squared errors, and define the generalization gap as

$$\text{Gap} = \text{MSE}_{\text{test}} - \text{MSE}_{\text{train}}.$$

Results are filtered for fixed conditions ($n = 500, d = 10, \sigma = 1$). The $\text{MSE}_{\text{test}}$ is computed using additional independent test samples, and $\text{MSE}_{\text{train}}$ is the in-sample training error.

### C.3. Statistical Dependence Measures

We then investigate the statistical dependence between instability measures and the generalization gap. Three types of associations are evaluated:

- Linear dependence: $R^2$ values from linear regression models of the form $\text{Gap} \sim S_{\text{feature}}$, $\text{Gap} \sim S_{\text{instance}}$, and $\text{Gap} \sim (S_{\text{feature}}, S_{\text{instance}})$.

- Mutual information (MI): Nonlinear associations are estimated using the mutual_info_regression function from scikit-learn.

- Partial mutual information (PMI): To isolate the unique contribution of each instability measure, we apply a residual-based approach using random forest regressions: we regress out the conditioning variable (e.g., $S_{\text{instance}}$ ) from both the predictor and target, and compute the mutual information between the residuals via the MINE estimator. This yields estimates of PMI ( $S_{\text{feature}}$ ; Gap | $S_{\text{instance}}$ ) and PMI( $S_{\text{instance}}$ ; Gap | $S_{\text{feature}}$ ).

## C.4. More Explanation about Partial Mutual Information

We provide an overview of partial mutual information used in the experiments.

Mutual information (MI) quantifies the amount of shared information between two random variables $X$ and $Y$. The partial mutual information (PMI) aims to measure the direct association between $X$ and $Y$ while removing the indirect influence of $Z$, typically by regressing out $Z$ from both $X$ and $Y$ and then computing the mutual information between the residuals. Formally, if $\tilde{X}$ and $\tilde{Y}$ denote the residuals obtained from regressing $X$ and $Y$ on $Z$, respectively, then

$$\text{PMI}(X; Y \mid Z) = I(\tilde{X}; \tilde{Y}). \tag{20}$$

Thus, PMI can be viewed as a nonparametric extension of the partial correlation concept to information-theoretic measures. Consider three Gaussian random variables satisfying

$$X = g_1(Z) + \epsilon_X, \tag{21}$$
$$Y = g_2(Z) + \epsilon_Y, \tag{22}$$

where $\epsilon_X$ and $\epsilon_Y$ are independent Gaussian noises and $Z$ is a common latent variable. In this case, $X$ and $Y$ are dependent due to their shared dependence on $Z$, yet they contain no direct interaction. If we compute the partial mutual information $\text{PMI}(X; Y \mid Z)$, it is zero because, after regressing out $Z$, the residuals $\tilde{X}$ and $\tilde{Y}$ are independent. However, if we slightly modify the model by introducing a direct coupling:

$$Y = g_1(Z) + g_2(X) + \epsilon_Y, \tag{23}$$

then $\text{PMI}(X; Y \mid Z) > 0$, indicating a direct dependence between $X$ and $Y$. If we treat the generalization gap as $Y$, and take $(X, Z)$ to be either $(\text{FI}, \text{II})$ or $(\text{II}, \text{FI})$, then both PMIs should be positive when both measures contain generalization-relevant information.

## C.5. Analysis

We investigate how FI and II jointly and individually explain variation in the generalization gap $Y$, under two configurations of the hyperparameter $\nu$ ($\nu = 1$ and $\nu = 0$). We report both deterministic correlations (via $R^2$) and information-theoretic dependencies (via mutual information and partial mutual information).

**Case 1: $\nu = 1$.** When $\nu = 1$, the individual explanatory powers of II and FI are moderate: $R^2(\text{II} \to Y) = 0.143$ and $R^2(\text{FI} \to Y) = 0.120$. However, their joint contribution increases substantially, achieving $R^2(\text{FI} + \text{II} \to Y) = 0.345$. This suggests a complementary effect between II and FI: neither measure alone suffices to capture the variability of the generalization gap, but together they explain a larger fraction of variance.

The mutual information (MI) values support this observation. The joint dependency between $(\text{FI}, \text{II})$ and $Y$ is substantial $(I((\text{FI}, \text{II}); Y) = 1.345)$, with asymmetric contributions: $\text{PMI}(\text{FI}; Y \mid \text{II}) = 0.469$, $\text{PMI}(\text{II}; Y \mid \text{FI}) = 0.439$, and mutual dependence between FI and II $(I(\text{FI}; \text{II}) = 0.558)$. These results indicate that both measures convey overlapping but not redundant information about generalization. FI contributes slightly more conditional information about the generalization gap once II is known.

**Case 2: $\nu = 0$.** At $\nu = 0$, the relationships become markedly stronger. $R^2(\text{II} \to Y) = 0.231$, $R^2(\text{FI} \to Y) = 0.546$, and $R^2(\text{FI} + \text{II} \to Y) = 0.675$. The dominant role of FI indicates that feature-level perturbations are the primary source of generalization-relevant variation, while II contributes additional but smaller explanatory power.

Information-theoretic results are consistent: $I((\text{FI}, \text{II}); Y) = 1.062$ (with FI dominating), $\text{PMI}(\text{FI}; Y \mid \text{II}) = 0.498$, $\text{PMI}(\text{II}; Y \mid \text{FI}) = 0.402$, and $I(\text{FI}; \text{II}) = 0.748$. Compared with the $\nu = 1$ case, the interdependence between FI and II increases, suggesting that when regularization or noise is reduced, the two measures become more correlated. Partial mutual information values also reveal that the unique contribution of each variable to generalization decreases slightly, consistent with the growing redundancy between FI and II.

**Interpretation**   Overall, these findings highlight two trends: (1) complementarity, where FI and II jointly enhance generalization predictability, especially when they are less correlated (as in $\nu = 1$), and (2) a redundancy-dominance shift, where for smaller $\nu$, FI becomes the dominant predictor while its correlation with II strengthens, reducing the independent information each measure provides. This suggests that different regimes of model regularization or noise change how II and FI interact with generalization: from complementary (when $\nu = 1$) to partially redundant but stronger overall predictors (when $\nu = 0$).

### C.6. Additional Robustness Checks

The preceding analysis is a focused synthetic random-forest study at two representative values of $\nu$. The robustness study here strictly broadens that analysis along four axes: six synthetic signal regimes $\nu \in \{0, 0.2, 0.4, 0.6, 0.8, 1\}$, four base estimator families (RF, DT, AdaBoost, and GBRT), four PMI residualizers (RF, DT, AdaBoost, and GBRT), and four completed real tabular regression benchmarks (`diabetes`, `abalone`, `cpu_act`, and `house_prices`). The base estimator grids contain 105, 35, 63, and 63 hyperparameter settings, respectively, and each reported summary uses 100 bootstrap replicates, each subsampling one third of the corresponding grid.

Tables 2–5 report the conditional contribution of FI after accounting for II. The entries are positive across all displayed synthetic and real-data settings under both $\mathrm{PMI}(\mathrm{FI}; \mathrm{gap} \mid \mathrm{II})$ and incremental $\Delta \mathrm{R}^2$. Across the 96 synthetic estimator/residualizer/$\nu$ groups and the 64 real-data estimator/residualizer/dataset groups, the bootstrap positive rate is 1.0 for every group. For $\Delta \mathrm{R}^2$, the values repeat across residualizer columns because the residualizer is only used in the PMI calculation. Thus, the conclusion from the focused analysis above persists under a broader estimator grid, multiple dependence residualizers, and real tabular data.

*Table 2.* Synthetic $\mathrm{PMI(FI; gap \mid II)}$ across estimator families and PMI residualizers (6 seeds/config, 100 bootstrap repetitions).

| Synthetic setting | RF residualizer | | | | DT residualizer | | | | AdaBoost residualizer | | | | GBRT residualizer | | | |
|---|---|---|---|---|---|---|---|---|---|---|---|---|---|---|---|---|
| | RF | DT | Ada | GBRT | RF | DT | Ada | GBRT | RF | DT | Ada | GBRT | RF | DT | Ada | GBRT |
| $\nu = 0.0$ | 0.380 | 0.633 | 0.485 | 0.376 | 0.417 | 0.841 | 0.598 | 0.696 | 0.375 | 0.582 | 0.460 | 0.350 | 0.337 | 0.486 | 0.432 | 0.343 |
| $\nu = 0.2$ | 0.251 | 0.496 | 0.454 | 0.443 | 0.325 | 0.723 | 0.590 | 0.644 | 0.263 | 0.441 | 0.455 | 0.404 | 0.239 | 0.391 | 0.378 | 0.360 |
| $\nu = 0.4$ | 0.238 | 0.378 | 0.347 | 0.375 | 0.252 | 0.555 | 0.428 | 0.525 | 0.239 | 0.342 | 0.349 | 0.365 | 0.223 | 0.335 | 0.303 | 0.340 |
| $\nu = 0.6$ | 0.228 | 0.373 | 0.522 | 0.306 | 0.299 | 0.495 | 0.636 | 0.424 | 0.249 | 0.344 | 0.512 | 0.281 | 0.226 | 0.352 | 0.427 | 0.255 |
| $\nu = 0.8$ | 0.223 | 0.322 | 0.582 | 0.368 | 0.236 | 0.404 | 0.682 | 0.550 | 0.229 | 0.307 | 0.595 | 0.348 | 0.222 | 0.294 | 0.477 | 0.318 |
| $\nu = 1.0$ | 0.250 | 0.303 | 0.458 | 0.414 | 0.291 | 0.378 | 0.585 | 0.537 | 0.265 | 0.291 | 0.488 | 0.397 | 0.224 | 0.302 | 0.379 | 0.381 |

*Table 3.* Synthetic $\Delta \mathrm{R}^2$ across estimator families and PMI residualizers (6 seeds/config, 100 bootstrap repetitions).

| Synthetic setting | RF residualizer | | | | DT residualizer | | | | AdaBoost residualizer | | | | GBRT residualizer | | | |
|---|---|---|---|---|---|---|---|---|---|---|---|---|---|---|---|---|
| | RF | DT | Ada | GBRT | RF | DT | Ada | GBRT | RF | DT | Ada | GBRT | RF | DT | Ada | GBRT |
| $\nu = 0.0$ | 0.152 | 0.485 | 0.246 | 0.231 | 0.152 | 0.485 | 0.246 | 0.231 | 0.152 | 0.485 | 0.246 | 0.231 | 0.152 | 0.485 | 0.246 | 0.231 |
| $\nu = 0.2$ | 0.064 | 0.402 | 0.300 | 0.273 | 0.064 | 0.402 | 0.300 | 0.273 | 0.064 | 0.402 | 0.300 | 0.273 | 0.064 | 0.402 | 0.300 | 0.273 |
| $\nu = 0.4$ | 0.014 | 0.320 | 0.204 | 0.334 | 0.014 | 0.320 | 0.204 | 0.334 | 0.014 | 0.320 | 0.204 | 0.334 | 0.014 | 0.320 | 0.204 | 0.334 |
| $\nu = 0.6$ | 0.028 | 0.310 | 0.510 | 0.332 | 0.028 | 0.310 | 0.510 | 0.332 | 0.028 | 0.310 | 0.510 | 0.332 | 0.028 | 0.310 | 0.510 | 0.332 |
| $\nu = 0.8$ | 0.011 | 0.133 | 0.496 | 0.420 | 0.011 | 0.133 | 0.496 | 0.420 | 0.011 | 0.133 | 0.496 | 0.420 | 0.011 | 0.133 | 0.496 | 0.420 |
| $\nu = 1.0$ | 0.016 | 0.095 | 0.316 | 0.454 | 0.016 | 0.095 | 0.316 | 0.454 | 0.016 | 0.095 | 0.316 | 0.454 | 0.016 | 0.095 | 0.316 | 0.454 |

*Table 4.* Real-data $\mathrm{PMI(FI; gap \mid II)}$ across estimator families and PMI residualizers (6 seeds/config, 100 bootstrap repetitions).

| Real dataset | RF residualizer | | | | DT residualizer | | | | AdaBoost residualizer | | | | GBRT residualizer | | | |
|---|---|---|---|---|---|---|---|---|---|---|---|---|---|---|---|---|
| | RF | DT | Ada | GBRT | RF | DT | Ada | GBRT | RF | DT | Ada | GBRT | RF | DT | Ada | GBRT |
| Abalone | 0.219 | 0.334 | 0.384 | 0.292 | 0.232 | 0.387 | 0.452 | 0.573 | 0.227 | 0.316 | 0.402 | 0.283 | 0.216 | 0.390 | 0.363 | 0.276 |
| CPU Act | 0.264 | 0.387 | 0.278 | 0.283 | 0.330 | 0.438 | 0.296 | 0.361 | 0.279 | 0.372 | 0.281 | 0.266 | 0.252 | 0.471 | 0.293 | 0.269 |
| Diabetes | 0.358 | 0.368 | 0.549 | 0.437 | 0.412 | 0.444 | 0.597 | 0.538 | 0.364 | 0.332 | 0.524 | 0.404 | 0.339 | 0.328 | 0.505 | 0.398 |
| House prices | 0.216 | 0.310 | 0.404 | 0.365 | 0.222 | 0.313 | 0.399 | 0.436 | 0.219 | 0.288 | 0.373 | 0.355 | 0.213 | 0.294 | 0.368 | 0.347 |

*Table 5.* Real-data $\Delta \mathrm{R}^2$ across estimator families and PMI residualizers (6 seeds/config, 100 bootstrap repetitions).

| Real dataset | RF residualizer | | | | DT residualizer | | | | AdaBoost residualizer | | | | GBRT residualizer | | | |
|---|---|---|---|---|---|---|---|---|---|---|---|---|---|---|---|---|
| | RF | DT | Ada | GBRT | RF | DT | Ada | GBRT | RF | DT | Ada | GBRT | RF | DT | Ada | GBRT |
| Abalone | 0.017 | 0.212 | 0.295 | 0.114 | 0.017 | 0.212 | 0.295 | 0.114 | 0.017 | 0.212 | 0.295 | 0.114 | 0.017 | 0.212 | 0.295 | 0.114 |
| CPU Act | 0.087 | 0.202 | 0.028 | 0.027 | 0.087 | 0.202 | 0.028 | 0.027 | 0.087 | 0.202 | 0.028 | 0.027 | 0.087 | 0.202 | 0.028 | 0.027 |
| Diabetes | 0.100 | 0.217 | 0.469 | 0.314 | 0.100 | 0.217 | 0.469 | 0.314 | 0.100 | 0.217 | 0.469 | 0.314 | 0.100 | 0.217 | 0.469 | 0.314 |
| House prices | 0.016 | 0.047 | 0.238 | 0.048 | 0.016 | 0.047 | 0.238 | 0.048 | 0.016 | 0.047 | 0.238 | 0.048 | 0.016 | 0.047 | 0.238 | 0.048 |

We also performed a noise-control check by replacing FI with Gaussian and permuted-noise baselines and rerunning the same conditional-dependence analysis. Tables 6–9 report the differences between the FI-based statistic and the corresponding noise-based statistic, averaged over the two noise controls. The incremental $\Delta \mathrm{R}^2$ advantage of FI over the noise baseline is positive in all 320 noise-control aggregate entries. The $\mathrm{PMI}(\mathrm{FI}; \mathrm{gap} \mid \mathrm{II})$ advantage over $\mathrm{PMI}(\mathrm{noise}; \mathrm{gap} \mid \mathrm{II})$ is positive in 319 out of 320 entries; the only negative entry is $-7.7 \times 10^{-4}$, which is numerically negligible. Thus, the observed conditional signal is not explained by simply adding another random covariate.

# D. Contents Related to Stability of Bagged Linear Regression

## D.1. Error Analysis

In the general setting, we may remove both instance $i$ and feature $j$ simultaneously. The updated sketching matrices $U_b'$ and $V_b'$ reflect these removals as defined above. The corresponding modified bagged estimator is:

$$\boldsymbol{\beta}' := \frac{1}{B} \sum_{b=1}^{B} V_b' (U_b'^\top X V_b')^\dagger U_b'^\top \boldsymbol{y}.$$

For notational simplicity, we use $U_b'$ and $V_b'$ throughout to denote the sketching matrices under the appropriate removal operation. If only instance subsampling is applied (i.e., no feature subsampling), then $V_b' = V_b$ for all $b$; conversely, if only feature sampling is used, we have $U_b' = U_b$. For notational simplicity, we define the weighting matrices

$$g(X) = \frac{1}{B} \sum_{b=1}^{B} V_b (U_b^\top X V_b)^\dagger U_b^\top, g'(X) = \frac{1}{B} \sum_{b=1}^{B} V_b' (U_b'^\top X V_b')^\dagger U_b'^\top,$$

so that $\boldsymbol{\beta} = g(X)\boldsymbol{y}$, $\boldsymbol{\beta}' = g'(X)\boldsymbol{y}$. We decompose the instability into

$$
\mathbb{E}\left[\|\boldsymbol{\beta} - \boldsymbol{\beta}'\|_2^2\right] = \mathbb{E}_{X,y,U,V}\left[\boldsymbol{y}^\top (g(X) - g'(X))^\top (g(X) - g'(X)) \boldsymbol{y}\right]
$$

$$
= \sigma^2 \underbrace{\mathbb{E}_{X,U,V}\left[\mathrm{tr}\left((g(X) - g'(X))^\top (g(X) - g'(X))\right)\right]}_{\textbf{Variance}}
$$

$$
+ \frac{1}{d} \underbrace{\mathbb{E}_{X,U,V}\left[\mathrm{tr}\left(X^\top (g(X) - g'(X))^\top (g(X) - g'(X)) X\right)\right]}_{\textbf{Bias}}.
$$

The two terms correspond to the impact of removal on the variance and bias terms, respectively. The following propositions give the exact values of the variance and the bias term.

**Proposition D.1** (**Variance term of instability**). *Assume the above assumptions hold. Then the expectation of the variance term is*

$$
\sigma^2 \mathbb{E}_{X,U,V}\left[\mathrm{tr}\left((g(X) - g'(X))^\top (g(X) - g'(X))\right)\right] = \sigma^2 \left(\frac{\Delta^{V=}}{B} + \frac{(B-1)\Delta^{V\neq}}{B}\right),
$$

*where*

$$
\lim_{n,d\to\infty} \Delta^{V=} := \begin{cases} \frac{2\gamma q(1-p)}{(p-\gamma q)(1-\gamma q)} & \text{if } \gamma q < p \text{ and } i\text{-th instance removed} \\ \frac{2\gamma pq(1-q)}{(p-\gamma q)(p-\gamma q^2)} & \text{if } \gamma q < p \text{ and } j\text{-th feature removed} \\ \frac{2\gamma q(p-p^2)}{(\gamma q-p)(\gamma q-p^2)} & \text{if } \gamma q > p \text{ and } i\text{-th instance removed} \\ \frac{2\gamma q(1-p)}{(\gamma q-p)(\gamma-p)} & \text{if } \gamma q > p \text{ and } j\text{-th feature removed} \end{cases}
$$

*and*

$$
\lim_{n,d\to\infty} n\Delta^{V\neq} := \begin{cases} \frac{\gamma q^2}{(1-\gamma q^2)^2} & \text{if } \gamma q < p \text{ and } i\text{-th instance removed} \\ \frac{q^2}{(1-\gamma q^2)^2} & \text{if } \gamma q < p \text{ and } j\text{-th feature removed} \\ \frac{\gamma p^2}{(\gamma-p^2)^2} & \text{if } \gamma q > p \text{ and } i\text{-th instance removed} \\ \frac{p^2}{(\gamma-p^2)^2} & \text{if } \gamma q > p \text{ and } j\text{-th feature removed} . \end{cases}
$$

*Table 6.* Synthetic noise-control: average PMI difference $\mathrm{PMI}(\mathrm{FI}; \mathrm{gap} \mid \mathrm{II}) - \mathrm{PMI}(\mathrm{noise}; \mathrm{gap} \mid \mathrm{II})$ across estimator families and PMI residualizers (averaged over Gaussian and permuted noise; 6 seeds/config, 100 bootstrap repetitions).

| Synthetic setting | RF residualizer | | | | DT residualizer | | | | AdaBoost residualizer | | | | GBRT residualizer | | | |
|---|---|---|---|---|---|---|---|---|---|---|---|---|---|---|---|---|
| | RF | DT | Ada | GBRT | RF | DT | Ada | GBRT | RF | DT | Ada | GBRT | RF | DT | Ada | GBRT |
| $\nu = 0.0$ | 0.167 | 0.371 | 0.241 | 0.116 | 0.201 | 0.565 | 0.357 | 0.452 | 0.170 | 0.325 | 0.216 | 0.101 | 0.118 | 0.191 | 0.188 | 0.096 |
| $\nu = 0.2$ | 0.042 | 0.239 | 0.206 | 0.198 | 0.119 | 0.454 | 0.344 | 0.394 | 0.045 | 0.185 | 0.204 | 0.160 | 0.027 | 0.128 | 0.112 | 0.120 |
| $\nu = 0.4$ | 0.028 | 0.115 | 0.103 | 0.134 | 0.040 | 0.280 | 0.181 | 0.270 | 0.027 | 0.100 | 0.104 | 0.122 | 0.007 | 0.074 | 0.053 | 0.099 |
| $\nu = 0.6$ | 0.020 | 0.095 | 0.262 | 0.049 | 0.079 | 0.241 | 0.382 | 0.173 | 0.042 | 0.094 | 0.247 | 0.037 | 0.011 | 0.063 | 0.174 | 0.012 |
| $\nu = 0.8$ | 0.008 | 0.050 | 0.325 | 0.135 | 0.026 | 0.139 | 0.424 | 0.323 | 0.021 | 0.039 | 0.344 | 0.120 | 0.006 | 0.029 | 0.209 | 0.086 |
| $\nu = 1.0$ | 0.042 | 0.039 | 0.221 | 0.162 | 0.079 | 0.098 | 0.338 | 0.298 | 0.047 | 0.019 | 0.241 | 0.150 | 0.011 | 0.029 | 0.131 | 0.123 |

*Table 7.* Synthetic noise-control: average $\Delta \mathrm{R}^2$ difference $\Delta \mathrm{R}^2(\mathrm{FI}) - \Delta \mathrm{R}^2(\mathrm{noise})$ across estimator families and PMI residualizers (averaged over Gaussian and permuted noise; 6 seeds/config, 100 bootstrap repetitions).

| Synthetic setting | RF residualizer | | | | DT residualizer | | | | AdaBoost residualizer | | | | GBRT residualizer | | | |
|---|---|---|---|---|---|---|---|---|---|---|---|---|---|---|---|---|
| | RF | DT | Ada | GBRT | RF | DT | Ada | GBRT | RF | DT | Ada | GBRT | RF | DT | Ada | GBRT |
| $\nu = 0.0$ | 0.152 | 0.477 | 0.240 | 0.235 | 0.152 | 0.475 | 0.241 | 0.234 | 0.152 | 0.475 | 0.240 | 0.235 | 0.153 | 0.476 | 0.240 | 0.234 |
| $\nu = 0.2$ | 0.062 | 0.408 | 0.306 | 0.254 | 0.061 | 0.407 | 0.307 | 0.255 | 0.061 | 0.407 | 0.307 | 0.253 | 0.061 | 0.411 | 0.305 | 0.254 |
| $\nu = 0.4$ | 0.011 | 0.294 | 0.194 | 0.333 | 0.011 | 0.295 | 0.195 | 0.334 | 0.011 | 0.295 | 0.195 | 0.333 | 0.011 | 0.296 | 0.195 | 0.334 |
| $\nu = 0.6$ | 0.027 | 0.292 | 0.516 | 0.335 | 0.027 | 0.290 | 0.517 | 0.335 | 0.027 | 0.289 | 0.515 | 0.335 | 0.027 | 0.290 | 0.516 | 0.336 |
| $\nu = 0.8$ | 0.011 | 0.137 | 0.448 | 0.422 | 0.011 | 0.141 | 0.447 | 0.423 | 0.011 | 0.142 | 0.447 | 0.423 | 0.011 | 0.140 | 0.448 | 0.423 |
| $\nu = 1.0$ | 0.013 | 0.083 | 0.316 | 0.463 | 0.013 | 0.085 | 0.318 | 0.465 | 0.013 | 0.082 | 0.319 | 0.463 | 0.013 | 0.085 | 0.317 | 0.463 |

*Table 8.* Real-data noise-control: average PMI difference $\mathrm{PMI}(\mathrm{FI}; \mathrm{gap} \mid \mathrm{II}) - \mathrm{PMI}(\mathrm{noise}; \mathrm{gap} \mid \mathrm{II})$ across estimator families and PMI residualizers (averaged over Gaussian and permuted noise; 6 seeds/config, 100 bootstrap repetitions).

| Real dataset | RF residualizer | | | | DT residualizer | | | | AdaBoost residualizer | | | | GBRT residualizer | | | |
|---|---|---|---|---|---|---|---|---|---|---|---|---|---|---|---|---|
| | RF | DT | Ada | GBRT | RF | DT | Ada | GBRT | RF | DT | Ada | GBRT | RF | DT | Ada | GBRT |
| Abalone | 0.007 | 0.068 | 0.153 | 0.051 | 0.022 | 0.096 | 0.224 | 0.325 | 0.020 | 0.047 | 0.167 | 0.043 | 0.000 | 0.025 | 0.116 | 0.033 |
| CPU Act | 0.052 | 0.114 | 0.032 | 0.040 | 0.119 | 0.149 | 0.045 | 0.119 | 0.067 | 0.093 | 0.027 | 0.021 | 0.039 | 0.039 | 0.042 | 0.017 |
| Diabetes | 0.140 | 0.115 | 0.294 | 0.200 | 0.192 | 0.172 | 0.337 | 0.302 | 0.151 | 0.062 | 0.272 | 0.149 | 0.118 | 0.063 | 0.243 | 0.152 |
| House prices | 0.005 | 0.045 | 0.150 | 0.120 | 0.004 | 0.043 | 0.143 | 0.194 | 0.009 | 0.030 | 0.131 | 0.108 | 0.001 | 0.007 | 0.122 | 0.104 |

*Table 9.* Real-data noise-control: average $\Delta \mathrm{R}^2$ difference $\Delta \mathrm{R}^2(\mathrm{FI}) - \Delta \mathrm{R}^2(\mathrm{noise})$ across estimator families and PMI residualizers (averaged over Gaussian and permuted noise; 6 seeds/config, 100 bootstrap repetitions).

| Real dataset | RF residualizer | | | | DT residualizer | | | | AdaBoost residualizer | | | | GBRT residualizer | | | |
|---|---|---|---|---|---|---|---|---|---|---|---|---|---|---|---|---|
| | RF | DT | Ada | GBRT | RF | DT | Ada | GBRT | RF | DT | Ada | GBRT | RF | DT | Ada | GBRT |
| Abalone | 0.011 | 0.184 | 0.312 | 0.103 | 0.013 | 0.186 | 0.312 | 0.103 | 0.012 | 0.184 | 0.311 | 0.103 | 0.013 | 0.183 | 0.311 | 0.103 |
| CPU Act | 0.083 | 0.173 | 0.018 | 0.020 | 0.084 | 0.171 | 0.017 | 0.021 | 0.082 | 0.175 | 0.019 | 0.021 | 0.082 | 0.172 | 0.017 | 0.021 |
| Diabetes | 0.092 | 0.201 | 0.430 | 0.327 | 0.092 | 0.200 | 0.430 | 0.328 | 0.092 | 0.201 | 0.431 | 0.327 | 0.091 | 0.199 | 0.431 | 0.327 |
| House prices | 0.010 | 0.035 | 0.238 | 0.046 | 0.011 | 0.035 | 0.236 | 0.046 | 0.010 | 0.033 | 0.238 | 0.047 | 0.009 | 0.036 | 0.239 | 0.046 |

*Table 10.* Noise-control consistency against the Gaussian baseline. "FI wins on" is the average, within each scope, of the bootstrap win rate for a fixed completed row; that row-level win rate is the fraction of bootstrap replicates in which the FI-based statistic exceeds its Gaussian-noise counterpart.

| Scope | mean $\Delta \mathrm{R}^2$ diff | FI wins on $\Delta \mathrm{R}^2$ | mean PMI diff | FI wins on PMI |
|---|---|---|---|---|
| Real | 0.143 | 0.935 | 0.103 | 0.852 |
| Synthetic subset | 0.252 | 0.972 | 0.155 | 0.907 |

**Proposition D.2** (**Bias term of instability**). *Assume the above assumptions hold. Then, the expectation of the bias term is*

$$\frac{1}{d}\,\mathbb{E}_{\boldsymbol{X},\boldsymbol{U},\boldsymbol{V}}\left[\mathrm{tr}\left(\boldsymbol{X}^\top\left(g(\boldsymbol{X})-g'(\boldsymbol{X})\right)^\top\left(g(\boldsymbol{X})-g'(\boldsymbol{X})\right)\boldsymbol{X}\right)\right]=\frac{B-1}{B}\Delta^{B\neq}+\frac{1}{B}\Delta^{B=},$$

*where*

$$\lim_{n,d\to\infty}\Delta^{B=}:=\begin{cases}\frac{2\gamma q(1-q)(1-p)}{(p-\gamma q)(1-\gamma q)} & \text{if }\gamma q<p\text{ and }i\text{-th instance removed}\\[2mm]\frac{2\gamma pq(1-q)^2}{(p-\gamma q)(p-\gamma q^2)}+\frac{2pq(1-q)}{p-\gamma q^2} & \text{if }\gamma q<p\text{ and }j\text{-th feature removed}\\[2mm]\frac{2\gamma pq(\gamma-p)(1-p)}{\gamma(\gamma q-p^2)}-\frac{2(1-p)p^2}{\gamma(\gamma q-p^2)} & \text{if }\gamma q>p\text{ and }i\text{-th instance removed}\\[2mm]\frac{2p(1-q)}{\gamma q-p} & \text{if }\gamma q>p\text{ and }j\text{-th feature removed}.\end{cases}$$

*and*

$$\lim_{n,d\to\infty}n\Delta^{B\neq}:=\begin{cases}\frac{\gamma q^2(1-q)^2}{(1-\gamma q^2)^2} & \text{if }\gamma q<p\text{ and }i\text{-th instance removed}\\[2mm]\frac{q^2(1+\gamma-2\gamma q)}{\gamma(1-\gamma q^2)^2} & \text{if }\gamma q<p\text{ and }j\text{-th feature removed}\\[2mm]\frac{p^2(\gamma-p)^2}{\gamma(\gamma-p^2)^2} & \text{if }\gamma q>p\text{ and }i\text{-th instance removed}\\[2mm]\frac{p^2(1-2p+\gamma)}{(\gamma-p^2)^2} & \text{if }\gamma q>p\text{ and }j\text{-th feature removed}.\end{cases}$$

*Proof of Theorem 3.1.* Theorem 3.1, including all entries of Table 1, follows by applying Propositions D.1 and D.2 to the two perturbations $\ell=-i$ and $\ell=-j$, and then collecting the corresponding variance and bias terms.

$\square$

## D.2. Useful Lemmas

**Lemma D.3** (Expectation of matrix under MP-law). *Let $\boldsymbol{X}$ be a $n\times d$ matrix with each row sampled from $\mathcal{N}(\boldsymbol{0},\Sigma)$, where $\Sigma$ is invertible. Then, there holds*

$$\mathbb{E}_{\boldsymbol{X}}\left[\boldsymbol{X}^\top\boldsymbol{X}\right]=n\Sigma,\quad\mathbb{E}_{\boldsymbol{X}}\left[(\boldsymbol{X}^\top\boldsymbol{X})^{-1}\right]=\frac{1}{n-d-1}\Sigma^{-1}.$$

*Proof of D.3.* The conclusions follow from standard properties of inverse Wishart distribution, see e.g. Haff (1979). $\square$

**Lemma D.4** (Lemma A.1 of LeJeune et al. (2020)). *Let $\boldsymbol{V}$ and $\boldsymbol{V}^c$ be the selection matrix corresponds to $\boldsymbol{\nu}$ and $\boldsymbol{\nu}^c$, respectively. Then for any random matrix $\boldsymbol{X}\in\mathbb{R}^{n\times d}$ whose entries are sampled i.i.d. from the standard Gaussian distribution, we have the following holds true*

$$\mathbb{E}_{\boldsymbol{XV}^c}\left[\boldsymbol{V}^\top\boldsymbol{X}^\dagger\right]=(\boldsymbol{XS})^\dagger\text{ if }n>d,\quad\text{and}\quad\mathbb{E}_{\boldsymbol{U}^{c\top}\boldsymbol{X}}\left[\boldsymbol{X}^\dagger T\right]=(\boldsymbol{U}^\top\boldsymbol{X})^\dagger\text{ if }n<d.$$

*Moreover, if $h(\boldsymbol{XS})$ and $\boldsymbol{XV}^c$ are independent, there holds*

$$\mathbb{E}_{\boldsymbol{XV}^c}\left[\boldsymbol{V}^{c\top}\boldsymbol{X}^\top h(\boldsymbol{XS})\boldsymbol{V}^\top\boldsymbol{X}^\dagger\right]=\boldsymbol{0}\text{ if }n>d,\quad\text{and}\quad\mathbb{E}_{\boldsymbol{U}^{c\top}X}\left[X^\top\boldsymbol{U}^c h(\boldsymbol{U}^\top X)X^\dagger T\right]=\boldsymbol{0}\text{ if }n<d.$$

The assumption $\mathbb{E}_T\left[\boldsymbol{U}_1^\top\boldsymbol{U}_1\right]=\mathbb{E}_T\left[\boldsymbol{U}_2^\top\boldsymbol{U}_2\right]=\frac{|\boldsymbol{\mu}_1|}{\mathrm{tr}(\Lambda)}\Lambda$ appears less natural, but is important to unify the results for $T$ and $T'$.

**Lemma D.5** (Variance). *For $\boldsymbol{\mu}_1,\boldsymbol{\mu}_2,\boldsymbol{\nu}_1,\boldsymbol{\nu}_2$ sampled independently, let $\boldsymbol{U}_1,\boldsymbol{U}_2,\boldsymbol{V}_1,\boldsymbol{V}_2$ be their selection matrix. Suppose $\Lambda_1,\Lambda_2$ and $\Theta_1,\Theta_2$ are diagonal matrices with only 1s and 0s on their diagonal. Then let $\mathbb{E}_T\left[\boldsymbol{U}_1\boldsymbol{U}_1^\top\right]=\frac{|\boldsymbol{\mu}_1|}{\mathrm{tr}(\Lambda_1)}\Lambda_1$, $\mathbb{E}_T\left[\boldsymbol{U}_2\boldsymbol{U}_2^\top\right]=\frac{|\boldsymbol{\mu}_2|}{\mathrm{tr}(\Lambda_2)}\Lambda_2$, $\mathbb{E}_S\left[\boldsymbol{V}_1\boldsymbol{V}_1^\top\right]=\frac{|\boldsymbol{\nu}_1|}{\mathrm{tr}(\Theta_1)}\Theta_1$, $\mathbb{E}_S\left[\boldsymbol{V}_2\boldsymbol{V}_2^\top\right]=\frac{|\boldsymbol{\nu}_2|}{\mathrm{tr}(\Theta_2)}\Theta_2$. Assume we always have $\Lambda_1\Lambda_2\in\{\Lambda_1,\Lambda_2\}$ and $\Theta_1\Theta\in\{\Theta_1,\Theta_2\}$. Define*

$$\Lambda^\wedge=\Lambda_1\Lambda_2,\quad\Lambda^\vee=\Lambda_1+\Lambda_2-\Lambda^\wedge,\quad\text{and}\quad\Theta^\wedge=\Theta_1\Theta_2,\quad\Theta^\vee=\Theta_1+\Theta_2-\Theta^\wedge.$$

*Then there holds*

$$\mathbb{E}_{\boldsymbol{X},\boldsymbol{S},\boldsymbol{T}}\left[\operatorname{tr}\left(\boldsymbol{U}_1(\boldsymbol{U}_1^\top \boldsymbol{X}\boldsymbol{V}_1)^{\dagger\top}\boldsymbol{V}_1^\top \boldsymbol{V}_2(\boldsymbol{U}_2^\top \boldsymbol{X}\boldsymbol{V}_2)^\dagger \boldsymbol{U}_2^\top)\right] = \begin{cases} \frac{|\boldsymbol{\nu}_1 \cap \boldsymbol{\nu}_2|}{\operatorname{tr}(\Lambda^\vee)-|\boldsymbol{\nu}_1 \cap \boldsymbol{\nu}_2|-1} & \text{if } |\boldsymbol{\nu}_1| < |\boldsymbol{\mu}_1|, |\boldsymbol{\nu}_2| < |\boldsymbol{\mu}_2|, \\ \frac{|\boldsymbol{\mu}_1 \cap \boldsymbol{\mu}_2|}{\operatorname{tr}(\Theta^\vee)-|\boldsymbol{\mu}_1 \cap \boldsymbol{\mu}_2|-1} & \text{if } |\boldsymbol{\nu}_1| > |\boldsymbol{\mu}_1|, |\boldsymbol{\nu}_2| > |\boldsymbol{\mu}_2|. \end{cases} \tag{24}$$

*Moreover, even if $\boldsymbol{\mu}_1$ and $\boldsymbol{\mu}_2$ are not independent, and $\boldsymbol{\nu}_1$ and $\boldsymbol{\nu}_2$ are not independent, yet we know $\boldsymbol{\mu}_1 \subset \boldsymbol{\mu}_2$, $\boldsymbol{\nu}_1 \subset \boldsymbol{\nu}_2$, the quantity becomes*

$$\mathbb{E}_{\boldsymbol{X},\boldsymbol{S},\boldsymbol{T}}\left[\operatorname{tr}\left(\boldsymbol{U}_1(\boldsymbol{U}_1^\top \boldsymbol{X}\boldsymbol{V}_1)^{\dagger\top}\boldsymbol{V}_1^\top \boldsymbol{V}_2(\boldsymbol{U}_2^\top \boldsymbol{X}\boldsymbol{V}_2)^\dagger \boldsymbol{U}_2^\top)\right] = \begin{cases} \frac{|\boldsymbol{\nu}_1|}{|\boldsymbol{\mu}_2|-|\boldsymbol{\nu}_1|-1} & \text{if } |\boldsymbol{\nu}_1| < |\boldsymbol{\mu}_1|, |\boldsymbol{\nu}_2| < |\boldsymbol{\mu}_2|, \\ \frac{|\boldsymbol{\mu}_1|}{|\boldsymbol{\nu}_2|-|\boldsymbol{\mu}_1|-1} & \text{if } |\boldsymbol{\nu}_1| > |\boldsymbol{\mu}_1|, |\boldsymbol{\nu}_2| > |\boldsymbol{\mu}_2|. \end{cases} \tag{25}$$

*Proof of Lemma D.5.* By definition, $\{\Lambda_1, \Lambda_2\} = \{\Lambda^\wedge, \Lambda^\vee\}$. W.o.l.g., we assume $(\Lambda_1, \Lambda_2) = (\Lambda^\wedge, \Lambda^\vee)$. Similarly, w.o.l.g., we assume $(\Theta_1, \Theta_2) = (\Theta^\wedge, \Theta^\vee)$.

**The first circumstance** We begin with the first circumstances. Let $\boldsymbol{V}_{1\cap2}$ be the selection matrix associated with $\boldsymbol{\nu}_1 \cap \boldsymbol{\nu}_2$. In this case, we can directly use the first argument of Lemma D.4 to get

$$\begin{aligned} &\mathbb{E}_{\boldsymbol{X},\boldsymbol{S},\boldsymbol{T}}\left[\operatorname{tr}\left(\boldsymbol{U}_1(\boldsymbol{U}_1^\top \boldsymbol{X}\boldsymbol{V}_1)^{\dagger\top}\boldsymbol{V}_1^\top \boldsymbol{V}_2(\boldsymbol{U}_2^\top \boldsymbol{X}\boldsymbol{V}_2)^\dagger \boldsymbol{U}_2^\top)\right] \\ =&\mathbb{E}_{\boldsymbol{X},\boldsymbol{S},\boldsymbol{T}}\left[\operatorname{tr}\left(\boldsymbol{U}_1(\boldsymbol{U}_1^\top \boldsymbol{X}\boldsymbol{V}_1)^{\dagger\top}\boldsymbol{V}_1^\top \boldsymbol{V}_{1\cap2}\boldsymbol{V}_{1\cap2}^\top \boldsymbol{V}_2(\boldsymbol{U}_2^\top \boldsymbol{X}\boldsymbol{V}_2)^\dagger \boldsymbol{U}_2^\top)\right] \\ =&\mathbb{E}_{\boldsymbol{X},\boldsymbol{S},\boldsymbol{T}}\left[\operatorname{tr}\left(\boldsymbol{U}_1(\boldsymbol{U}_1^\top \boldsymbol{X}\boldsymbol{V}_{1\cap2})^{\dagger\top}(\boldsymbol{U}_2^\top \boldsymbol{X}\boldsymbol{V}_{1\cap2})^\dagger \boldsymbol{U}_2^\top)\right]. \end{aligned}$$

Let $\Pi_A = I - A^{\top\dagger}A^\top$ be the projection matrix associated to $A$. Then, we can break the above equation to get

$$\begin{aligned} &\mathbb{E}_{\boldsymbol{X},\boldsymbol{S},\boldsymbol{T}}\left[\operatorname{tr}\left(\boldsymbol{U}_1(\boldsymbol{U}_1^\top \boldsymbol{X}\boldsymbol{V}_{1\cap2})^{\dagger\top}(\boldsymbol{U}_2^\top \boldsymbol{X}\boldsymbol{V}_{1\cap2})^\dagger \boldsymbol{U}_2^\top)\right] \\ =&\mathbb{E}_{\boldsymbol{X},\boldsymbol{S},\boldsymbol{T}}\left[\operatorname{tr}\left((\boldsymbol{U}_2^\top \boldsymbol{X}\boldsymbol{V}_{1\cap2})^\dagger \boldsymbol{U}_2^\top \boldsymbol{U}_1(\boldsymbol{U}_1^\top \boldsymbol{X}\boldsymbol{V}_{1\cap2})^{\dagger\top}\right)\right] \\ =&\mathbb{E}_{\boldsymbol{X},\boldsymbol{S},\boldsymbol{T}}\left[\operatorname{tr}\left((\boldsymbol{U}_2^\top \boldsymbol{X}\boldsymbol{V}_{1\cap2})^\dagger \boldsymbol{U}_2^\top \left(\Pi_{\boldsymbol{X}^\vee \boldsymbol{V}_{1\cap2}} + \boldsymbol{X}^\vee \boldsymbol{V}_{1\cap2}(\boldsymbol{V}_{1\cap2}^\top \boldsymbol{X}^{\vee\top}\boldsymbol{X}^\vee \boldsymbol{V}_{1\cap2})^{-1}\boldsymbol{V}_{1\cap2}^\top \boldsymbol{X}^{\vee\top}\right)\boldsymbol{U}_1(\boldsymbol{U}_1^\top \boldsymbol{X}\boldsymbol{V}_{1\cap2})^{\dagger\top}\right)\right], \end{aligned} \tag{26}$$

where we define $\boldsymbol{X}^\vee = \Lambda^\vee \boldsymbol{X}$ as a temporary notation. We scope in the first half of this quantity, where

$$(\boldsymbol{U}_2^\top \boldsymbol{X}\boldsymbol{V}_{1\cap2})^\dagger \boldsymbol{U}_2^\top \Pi_{\boldsymbol{X}^\vee \boldsymbol{V}_{1\cap2}} = (\boldsymbol{V}_{1\cap2}^\top \boldsymbol{X}^\top \boldsymbol{U}_2\boldsymbol{U}_2^\top \boldsymbol{X}\boldsymbol{V}_{1\cap2})^{-1}\boldsymbol{V}_{1\cap2}^\top \boldsymbol{X}^\top \boldsymbol{U}_2\boldsymbol{U}_2^\top \Pi_{\boldsymbol{X}^\vee \boldsymbol{V}_{1\cap2}}.$$

Since the distribution of $\boldsymbol{U}_2\boldsymbol{U}_2^\top$ is independent of the rest conditioned on we know $|\boldsymbol{\mu}_2|$, we have

$$\begin{aligned} \mathbb{E}_{\boldsymbol{T}}\left[(\boldsymbol{U}_2^\top \boldsymbol{X}\boldsymbol{V}_{1\cap2})^\dagger \boldsymbol{U}_2^\top \Pi_{\boldsymbol{X}^\vee \boldsymbol{V}_{1\cap2}}\right] =&(\boldsymbol{V}_{1\cap2}^\top \boldsymbol{X}^\top \boldsymbol{U}_2\boldsymbol{U}_2^\top \boldsymbol{X}\boldsymbol{V}_{1\cap2})^{-1}\boldsymbol{V}_{1\cap2}^\top \boldsymbol{X}^\top \frac{|\boldsymbol{\mu}_2|}{\operatorname{tr}(\Lambda_2)}\Lambda_2\Pi_{\boldsymbol{X}^\vee \boldsymbol{V}_{1\cap2}} \\ =&(\boldsymbol{V}_{1\cap2}^\top \boldsymbol{X}^\top \boldsymbol{U}_2\boldsymbol{U}_2^\top \boldsymbol{X}\boldsymbol{V}_{1\cap2})^{-1}\boldsymbol{V}_{1\cap2}^\top \boldsymbol{X}^{\vee\top}\Pi_{\boldsymbol{X}^\vee \boldsymbol{V}_{1\cap2}} = \boldsymbol{0}, \end{aligned}$$

where we recall that we assumed $\Lambda^\vee = \Lambda_2$. This brings (26) into

$$\begin{aligned} &\mathbb{E}_{\boldsymbol{X},\boldsymbol{S},\boldsymbol{T}}\left[\operatorname{tr}\left(\boldsymbol{U}_1(\boldsymbol{U}_1^\top \boldsymbol{X}\boldsymbol{V}_{1\cap2})^{\dagger\top}(\boldsymbol{U}_2^\top \boldsymbol{X}\boldsymbol{V}_{1\cap2})^\dagger \boldsymbol{U}_2^\top)\right] \\ =&\mathbb{E}_{\boldsymbol{X},\boldsymbol{S},\boldsymbol{T}}\left[\operatorname{tr}\left((\boldsymbol{U}_2^\top \boldsymbol{X}\boldsymbol{V}_{1\cap2})^\dagger \boldsymbol{U}_2^\top \boldsymbol{X}^\vee \boldsymbol{V}_{1\cap2}(\boldsymbol{V}_{1\cap2}^\top \boldsymbol{X}^{\vee\top}\boldsymbol{X}^\vee \boldsymbol{V}_{1\cap2})^{-1}\boldsymbol{V}_{1\cap2}^\top \boldsymbol{X}^{\vee\top}\boldsymbol{U}_1(\boldsymbol{U}_1^\top \boldsymbol{X}\boldsymbol{V}_{1\cap2})^{\dagger\top}\right)\right] \\ \overset{(i)}{=}&\mathbb{E}_{\boldsymbol{X},\boldsymbol{S},\boldsymbol{T}}\left[\operatorname{tr}\left((\boldsymbol{U}_2^\top \boldsymbol{X}\boldsymbol{V}_{1\cap2})^\dagger \boldsymbol{U}_2^\top \boldsymbol{X}\boldsymbol{V}_{1\cap2}(\boldsymbol{V}_{1\cap2}^\top \boldsymbol{X}^{\vee\top}\boldsymbol{X}^\vee \boldsymbol{V}_{1\cap2})^{-1}\boldsymbol{V}_{1\cap2}^\top \boldsymbol{X}^\top \boldsymbol{U}_1(\boldsymbol{U}_1^\top \boldsymbol{X}\boldsymbol{V}_{1\cap2})^{\dagger\top}\right)\right] \\ =&\mathbb{E}_{\boldsymbol{X},\boldsymbol{S},\boldsymbol{T}}\left[\operatorname{tr}\left((\boldsymbol{V}_{1\cap2}^\top \boldsymbol{X}^{\vee\top}\boldsymbol{X}^\vee \boldsymbol{V}_{1\cap2})^{-1}\right)\right]. \end{aligned}$$

To show step $(i)$, we observe the fact that

$$\boldsymbol{U}_1^\top \boldsymbol{X}^\vee = \boldsymbol{U}_1^\top \boldsymbol{X}, \text{ and } \boldsymbol{U}_2^\top \boldsymbol{X}^\vee = \boldsymbol{U}_2^\top \boldsymbol{X}. \tag{27}$$

This is because $\Lambda^\vee$ is a diagonal matrix with only $1$s and $0s$ on its diagonal, while on its positions with $0$s, $\boldsymbol{U}_1$ and $\boldsymbol{U}_2$ must have zero rows. Otherwise, the expectation of $\boldsymbol{U}_1$ and $\boldsymbol{U}_2$ can not have zero expectation on these positions. The last step is due to Lemma D.3, where we have

$$\mathbb{E}_{\boldsymbol{X},\boldsymbol{S},\boldsymbol{T}}\left[\operatorname{tr}\left((\boldsymbol{V}_{1\cap2}^\top \boldsymbol{X}^{\vee\top}\boldsymbol{X}^\vee \boldsymbol{V}_{1\cap2})^{-1}\right)\right] = \frac{|\boldsymbol{\nu}_1 \cap \boldsymbol{\nu}_2|}{\operatorname{tr}(\Lambda^\vee) - |\boldsymbol{\nu}_1 \cap \boldsymbol{\nu}_2| - 1}.$$

To show the second condition, we notice that the operations between

$$\mathbb{E}_{\boldsymbol{X},\boldsymbol{S},\boldsymbol{T}}\left[\operatorname{tr}\left(\boldsymbol{U}_1(\boldsymbol{U}_1^\top \boldsymbol{X}\boldsymbol{V}_1)^{\dagger\top}\boldsymbol{V}_1^\top \boldsymbol{V}_2(\boldsymbol{U}_2^\top \boldsymbol{X}\boldsymbol{V}_2)^\dagger \boldsymbol{U}_2^\top\right)\right] = \mathbb{E}_{\boldsymbol{X},\boldsymbol{S},\boldsymbol{T}}\left[\operatorname{tr}\left(\boldsymbol{U}_1(\boldsymbol{U}_1^\top \boldsymbol{X}\boldsymbol{V}_{1\cap2})^{\dagger\top}(\boldsymbol{U}_2^\top \boldsymbol{X}\boldsymbol{V}_{1\cap2})^\dagger \boldsymbol{U}_2^\top\right)\right]$$

are still lawful, where we continue to have

$$\begin{aligned}
&\mathbb{E}_{\boldsymbol{X},\boldsymbol{S},\boldsymbol{T}}\left[\operatorname{tr}\left(\boldsymbol{U}_1(\boldsymbol{U}_1^\top \boldsymbol{X}\boldsymbol{V}_{1\cap2})^{\dagger\top}(\boldsymbol{U}_2^\top \boldsymbol{X}\boldsymbol{V}_{1\cap2})^\dagger \boldsymbol{U}_2^\top\right)\right]\\
=&\mathbb{E}_{\boldsymbol{X},\boldsymbol{S},\boldsymbol{T}}\left[\operatorname{tr}\left((\boldsymbol{V}_{1\cap2}^\top \boldsymbol{X}^\top \boldsymbol{U}_2\boldsymbol{U}_2^\top \boldsymbol{X}\boldsymbol{V}_{1\cap2})^{-1}\boldsymbol{V}_{1\cap2}^\top \boldsymbol{X}^\top \boldsymbol{U}_2\boldsymbol{U}_2^\top \boldsymbol{U}_1\boldsymbol{U}_1^\top \boldsymbol{X}\boldsymbol{V}_{1\cap2}(\boldsymbol{V}_{1\cap2}^\top \boldsymbol{X}^\top \boldsymbol{U}_1\boldsymbol{U}_1^\top \boldsymbol{X}\boldsymbol{V}_{1\cap2})^{-1}\right)\right]\\
\overset{(ii)}{=}&\mathbb{E}_{\boldsymbol{X},\boldsymbol{S},\boldsymbol{T}}\left[\operatorname{tr}\left((\boldsymbol{V}_{1\cap2}^\top \boldsymbol{X}^\top \boldsymbol{U}_2\boldsymbol{U}_2^\top \boldsymbol{X}\boldsymbol{V}_{1\cap2})^{-1}\boldsymbol{V}_{1\cap2}^\top \boldsymbol{X}^\top \boldsymbol{U}_1\boldsymbol{U}_1^\top \boldsymbol{X}\boldsymbol{V}_{1\cap2}(\boldsymbol{V}_{1\cap2}^\top \boldsymbol{X}^\top \boldsymbol{U}_1\boldsymbol{U}_1^\top \boldsymbol{X}\boldsymbol{V}_{1\cap2})^{-1}\right)\right]\\
=&\mathbb{E}_{\boldsymbol{X},\boldsymbol{S},\boldsymbol{T}}\left[\operatorname{tr}\left((\boldsymbol{V}_{1\cap2}^\top \boldsymbol{X}^\top \boldsymbol{U}_2\boldsymbol{U}_2^\top \boldsymbol{X}\boldsymbol{V}_{1\cap2})^{-1}\right)\right] = \frac{|\boldsymbol{\nu}_1|}{|\boldsymbol{\mu}_2| - |\boldsymbol{\nu}_1| - 1},
\end{aligned}$$

where $(ii)$ used the assumption that $\boldsymbol{\mu}_1 \subset \boldsymbol{\mu}_2$, and the last step used Lemma D.3. We notice that $\boldsymbol{\nu}_1 \cap \boldsymbol{\nu}_2 = \boldsymbol{\nu}_1$.

**The second circumstance**  Then we proceed to the second circumstance where $|\boldsymbol{\nu}_1| > |\boldsymbol{\mu}_1|, |\boldsymbol{\nu}_2| > |\boldsymbol{\mu}_2|$. This is an analog to the first circumstance, yet we operate on the dual dimension. Let $\boldsymbol{U}_{1\cap2}$ be the selection matrix associated with $\boldsymbol{\mu}_1 \cap \boldsymbol{\mu}_2$. We use the first argument of Lemma D.4 to get

$$\begin{aligned}
&\mathbb{E}_{\boldsymbol{X},\boldsymbol{S},\boldsymbol{T}}\left[\operatorname{tr}\left(\boldsymbol{U}_1(\boldsymbol{U}_1^\top \boldsymbol{X}\boldsymbol{V}_1)^{\dagger\top}\boldsymbol{V}_1^\top \boldsymbol{V}_2(\boldsymbol{U}_2^\top \boldsymbol{X}\boldsymbol{V}_2)^\dagger \boldsymbol{U}_2^\top\right)\right]\\
=&\mathbb{E}_{\boldsymbol{X},\boldsymbol{S},\boldsymbol{T}}\left[\operatorname{tr}\left(\boldsymbol{V}_2(\boldsymbol{U}_2^\top \boldsymbol{X}\boldsymbol{V}_2)^\dagger \boldsymbol{U}_2^\top \boldsymbol{U}_{1\cap2}\boldsymbol{U}_{1\cap2}^\top \boldsymbol{U}_1(\boldsymbol{U}_1^\top \boldsymbol{X}\boldsymbol{V}_1)^{\dagger\top}\boldsymbol{V}_1^\top\right)\right]\\
=&\mathbb{E}_{\boldsymbol{X},\boldsymbol{S},\boldsymbol{T}}\left[\operatorname{tr}\left(\boldsymbol{V}_2(\boldsymbol{U}_{1\cap2}^\top \boldsymbol{X}\boldsymbol{V}_2)^\dagger (\boldsymbol{U}_{1\cap2}^\top \boldsymbol{X}\boldsymbol{V}_1)^{\dagger\top}\boldsymbol{V}_1^\top\right)\right]\\
=&\mathbb{E}_{\boldsymbol{X},\boldsymbol{S},\boldsymbol{T}}\left[\operatorname{tr}\left((\boldsymbol{U}_{1\cap2}^\top \boldsymbol{X}\boldsymbol{V}_1)^{\dagger\top}\boldsymbol{V}_1^\top \boldsymbol{V}_2(\boldsymbol{U}_{1\cap2}^\top \boldsymbol{X}\boldsymbol{V}_2)^\dagger\right)\right]\\
=&\mathbb{E}_{\boldsymbol{X},\boldsymbol{S},\boldsymbol{T}}\left[\operatorname{tr}\left((\boldsymbol{U}_{1\cap2}^\top \boldsymbol{X}\boldsymbol{V}_1)^{\dagger\top}\boldsymbol{V}_1^\top \left(\Pi_{\boldsymbol{X}^{\vee\top}\boldsymbol{U}_{1\cap2}} + \boldsymbol{X}^{\vee\top}\boldsymbol{U}_{1\cap2}(\boldsymbol{U}_{1\cap2}^\top \boldsymbol{X}^\vee \boldsymbol{X}^{\vee\top}\boldsymbol{U}_{1\cap2})^{-1}\boldsymbol{U}_{1\cap2}^\top \boldsymbol{X}^\vee\right)\boldsymbol{V}_2(\boldsymbol{U}_{1\cap2}^\top \boldsymbol{X}\boldsymbol{V}_2)^\dagger\right)\right],
\end{aligned}$$

where $\boldsymbol{X}^\vee = \boldsymbol{X}\Theta^\vee$. Following the same reasoning in the first circumstance, we get

$$\begin{aligned}
\mathbb{E}_{\boldsymbol{X},\boldsymbol{S},\boldsymbol{T}}\left[\operatorname{tr}\left(\boldsymbol{U}_1(\boldsymbol{U}_1^\top \boldsymbol{X}\boldsymbol{V}_1)^{\dagger\top}\boldsymbol{V}_1^\top \boldsymbol{V}_2(\boldsymbol{U}_2^\top \boldsymbol{X}\boldsymbol{V}_2)^\dagger \boldsymbol{U}_2^\top\right)\right] &=\mathbb{E}_{\boldsymbol{X},\boldsymbol{S},\boldsymbol{T}}\left[\operatorname{tr}\left((\boldsymbol{U}_{1\cap2}^\top \boldsymbol{X}^\vee \boldsymbol{X}^{\vee\top}\boldsymbol{U}_{1\cap2})^{-1}\right)\right]\\
&=\frac{|\boldsymbol{\mu}_1 \cap \boldsymbol{\mu}_2|}{\operatorname{tr}(\Theta^\vee) - |\boldsymbol{\mu}_1 \cap \boldsymbol{\mu}_2| - 1}.
\end{aligned}$$

To show the second condition, we notice that the operations between

$$\mathbb{E}_{\boldsymbol{X},\boldsymbol{S},\boldsymbol{T}}\left[\operatorname{tr}\left(\boldsymbol{U}_1(\boldsymbol{U}_1^\top \boldsymbol{X}\boldsymbol{V}_1)^{\dagger\top}\boldsymbol{V}_1^\top \boldsymbol{V}_2(\boldsymbol{U}_2^\top \boldsymbol{X}\boldsymbol{V}_2)^\dagger \boldsymbol{U}_2^\top\right)\right] = \mathbb{E}_{\boldsymbol{X},\boldsymbol{S},\boldsymbol{T}}\left[\operatorname{tr}\left((\boldsymbol{U}_{1\cap2}^\top \boldsymbol{X}\boldsymbol{V}_1)^{\dagger\top}\boldsymbol{V}_1^\top \boldsymbol{V}_2(\boldsymbol{U}_{1\cap2}^\top \boldsymbol{X}\boldsymbol{V}_2)^\dagger\right)\right]$$

are still lawful, where we continue to have

$$\begin{aligned}
&\mathbb{E}_{\boldsymbol{X},\boldsymbol{S},\boldsymbol{T}}\left[\operatorname{tr}\left((\boldsymbol{U}_{1\cap2}^\top \boldsymbol{X}\boldsymbol{V}_1)^{\dagger\top}\boldsymbol{V}_1^\top \boldsymbol{V}_2(\boldsymbol{U}_{1\cap2}^\top \boldsymbol{X}\boldsymbol{V}_2)^\dagger\right)\right]\\
=&\mathbb{E}_{\boldsymbol{X},\boldsymbol{S},\boldsymbol{T}}\left[\operatorname{tr}\left((\boldsymbol{U}_{1\cap2}^\top \boldsymbol{X}\boldsymbol{V}_1\boldsymbol{V}_1^\top \boldsymbol{X}^\top \boldsymbol{U}_{1\cap2})^{-1}\boldsymbol{U}_{1\cap2}^\top \boldsymbol{X}\boldsymbol{V}_1\boldsymbol{V}_1^\top \boldsymbol{V}_2\boldsymbol{V}_2^\top \boldsymbol{X}^\top \boldsymbol{U}_{1\cap2}(\boldsymbol{U}_{1\cap2}^\top \boldsymbol{X}\boldsymbol{V}_2\boldsymbol{V}_2^\top \boldsymbol{X}^\top \boldsymbol{U}_{1\cap2})^{-1}\right)\right]\\
=&\mathbb{E}_{\boldsymbol{X},\boldsymbol{S},\boldsymbol{T}}\left[\operatorname{tr}\left((\boldsymbol{U}_{1\cap2}^\top \boldsymbol{X}\boldsymbol{V}_1\boldsymbol{V}_1^\top \boldsymbol{X}^\top \boldsymbol{U}_{1\cap2})^{-1}\boldsymbol{U}_{1\cap2}^\top \boldsymbol{X}\boldsymbol{V}_1\boldsymbol{V}_1^\top \boldsymbol{X}^\top \boldsymbol{U}_{1\cap2}(\boldsymbol{U}_{1\cap2}^\top \boldsymbol{X}\boldsymbol{V}_2\boldsymbol{V}_2^\top \boldsymbol{X}^\top \boldsymbol{U}_{1\cap2})^{-1}\right)\right]\\
=&\mathbb{E}_{\boldsymbol{X},\boldsymbol{S},\boldsymbol{T}}\left[\operatorname{tr}\left((\boldsymbol{U}_{1\cap2}^\top \boldsymbol{X}\boldsymbol{V}_2\boldsymbol{V}_2^\top \boldsymbol{X}^\top \boldsymbol{U}_{1\cap2})^{-1}\right)\right] = \frac{|\boldsymbol{\mu}_1|}{|\boldsymbol{\nu}_2| - |\boldsymbol{\mu}_1| - 1}.
\end{aligned}$$

$\square$

**Lemma D.6** (Bias). *For $\boldsymbol{\mu}_1, \boldsymbol{\mu}_2, \boldsymbol{\nu}_1, \boldsymbol{\nu}_2$ sampled independently, let $\boldsymbol{U}_1, \boldsymbol{U}_2, \boldsymbol{V}_1, \boldsymbol{V}_2$ be their selection matrix. Moreover, let $\boldsymbol{U}_1^c$ be the selection matrix for $\boldsymbol{\mu}_1^c$, and so on for $\boldsymbol{U}_2^c, \boldsymbol{V}_1^c, \boldsymbol{V}_2^c$. Suppose $\Lambda_1, \Lambda_2$ and $\Theta_1, \Theta_2$ are diagonal matrices with only 1s and 0s on their diagonal. Then let $\mathbb{E}_{\boldsymbol{T}}\left[\boldsymbol{U}_1\boldsymbol{U}_1^\top\right] = \frac{|\boldsymbol{\mu}_1|}{\operatorname{tr}(\Lambda_1)}\Lambda_1$, $\mathbb{E}_{\boldsymbol{T}}\left[\boldsymbol{U}_2\boldsymbol{U}_2^\top\right] = \frac{|\boldsymbol{\mu}_2|}{\operatorname{tr}(\Lambda_2)}\Lambda_2$, $\mathbb{E}_{\boldsymbol{S}}\left[\boldsymbol{V}_1\boldsymbol{V}_1^\top\right] = \frac{|\boldsymbol{\nu}_1|}{\operatorname{tr}(\Theta_1)}\Theta_1$, $\mathbb{E}_{\boldsymbol{S}}\left[\boldsymbol{V}_2\boldsymbol{V}_2^\top\right] = \frac{|\boldsymbol{\nu}_2|}{\operatorname{tr}(\Theta_2)}\Theta_2$. We want to calculate*

$$(\%) = \mathbb{E}_{\boldsymbol{X},\boldsymbol{S},\boldsymbol{T}}\left[\operatorname{tr}\left(\boldsymbol{X}^\top \boldsymbol{U}_1(\boldsymbol{U}_1^\top \boldsymbol{X}\boldsymbol{V}_1)^{\dagger\top}\boldsymbol{V}_1^\top \boldsymbol{V}_2(\boldsymbol{U}_2^\top \boldsymbol{X}\boldsymbol{V}_2)^\dagger \boldsymbol{U}_2^\top \boldsymbol{X}\right)\right].$$

*Assume we always have $\Lambda_1\Lambda_2 \in \{\Lambda_1, \Lambda_2\}$ and $\Theta_1\Theta \in \{\Theta_1, \Theta_2\}$. Define*

$$\Lambda^\wedge = \Lambda_1\Lambda_2, \ \ \Lambda^\vee = \Lambda_1 + \Lambda_2 - \Lambda^\wedge, \ and \ \ \Theta^\wedge = \Theta_1\Theta_2, \ \ \Theta^\vee = \Theta_1 + \Theta_2 - \Theta^\wedge.$$

*Then there holds*

$$(\%) = \begin{cases} \frac{|\boldsymbol{\nu}_1 \cap \boldsymbol{\nu}_2||\boldsymbol{\nu}_1^c \cap \boldsymbol{\nu}_2^c|}{\mathrm{tr}(\Lambda^\vee) - |\boldsymbol{\nu}_1 \cap \boldsymbol{\nu}_2| - 1} + |\boldsymbol{\nu}_1 \cap \boldsymbol{\nu}_2| & \text{if } |\boldsymbol{\nu}_1| < |\boldsymbol{\mu}_1|, |\boldsymbol{\nu}_2| < |\boldsymbol{\mu}_2|, \\ \frac{|\boldsymbol{\mu}_1/\boldsymbol{\mu}_2||\boldsymbol{\mu}_2/\boldsymbol{\mu}_1|}{\mathrm{tr}(\Theta^\vee) - |\boldsymbol{\mu}_1 \cap \boldsymbol{\mu}_2|} + |\boldsymbol{\mu}_1 \cap \boldsymbol{\mu}_2|\frac{d - |\boldsymbol{\mu}_1 \cap \boldsymbol{\mu}_2| - 1}{\mathrm{tr}(\Theta^\vee) - |\boldsymbol{\mu}_1 \cap \boldsymbol{\mu}_2| - 1} & \text{if } |\boldsymbol{\nu}_1| > |\boldsymbol{\mu}_1|, |\boldsymbol{\nu}_2| > |\boldsymbol{\mu}_2|. \end{cases} \tag{28}$$

*Moreover, even if $\boldsymbol{\mu}_1$ and $\boldsymbol{\mu}_2$ are not independent, and $\boldsymbol{\nu}_1$ and $\boldsymbol{\nu}_2$ are not independent, yet we know $\boldsymbol{\mu}_1 \subset \boldsymbol{\mu}_2$, $\boldsymbol{\nu}_1 \subset \boldsymbol{\nu}_2$, the quantity becomes*

$$(\%) = \begin{cases} \frac{|\boldsymbol{\nu}_1|(d - |\boldsymbol{\nu}_2|)}{|\boldsymbol{\mu}_2| - |\boldsymbol{\nu}_1| - 1} + |\boldsymbol{\nu}_1| & \text{if } |\boldsymbol{\nu}_1| < |\boldsymbol{\mu}_1|, |\boldsymbol{\nu}_2| < |\boldsymbol{\mu}_2|, \\ \frac{|\boldsymbol{\mu}_1|(d - |\boldsymbol{\nu}_2|)}{|\boldsymbol{\nu}_2| - |\boldsymbol{\mu}_1| - 1} + |\boldsymbol{\mu}_1| & \text{if } |\boldsymbol{\nu}_1| > |\boldsymbol{\mu}_1|, |\boldsymbol{\nu}_2| > |\boldsymbol{\mu}_2|. \end{cases} \tag{29}$$

*Proof of Lemma D.6.* By definition, $\{\Lambda_1, \Lambda_2\} = \{\Lambda^\wedge, \Lambda^\vee\}$. W.o.l.g., we assume $(\Lambda_1, \Lambda_2) = (\Lambda^\wedge, \Lambda^\vee)$. Similarly, w.o.l.g., we assume $(\Theta_1, \Theta_2) = (\Theta^\wedge, \Theta^\vee)$.

**The first circumstance** We first write

$$(\%) = \underbrace{\mathbb{E}_{\boldsymbol{X}, \boldsymbol{S}, \boldsymbol{T}} \left[ \mathrm{tr} \left( \boldsymbol{V}_1^c \boldsymbol{V}_1^{c\top} \boldsymbol{X}^\top \boldsymbol{U}_1 (\boldsymbol{U}_1^\top \boldsymbol{X} \boldsymbol{V}_1)^{\dagger\top} \boldsymbol{V}_1^\top \boldsymbol{V}_2 (\boldsymbol{U}_2^\top \boldsymbol{X} \boldsymbol{V}_2)^\dagger \boldsymbol{U}_2^\top \boldsymbol{X} \boldsymbol{V}_2^c \boldsymbol{V}_2^{c\top} \right) \right]}_{(*)}$$

$$+ \underbrace{\mathbb{E}_{\boldsymbol{X}, \boldsymbol{S}, \boldsymbol{T}} \left[ \mathrm{tr} \left( \boldsymbol{V}_1 \boldsymbol{V}_1^\top \boldsymbol{X}^\top \boldsymbol{U}_1 (\boldsymbol{U}_1^\top \boldsymbol{X} \boldsymbol{V}_1)^{\dagger\top} \boldsymbol{V}_1^\top \boldsymbol{V}_2 (\boldsymbol{U}_2^\top \boldsymbol{X} \boldsymbol{V}_2)^\dagger \boldsymbol{U}_2^\top \boldsymbol{X} \boldsymbol{V}_2^c \boldsymbol{V}_2^{c\top} \right) \right]}_{(**)}$$

$$+ \underbrace{\mathbb{E}_{\boldsymbol{X}, \boldsymbol{S}, \boldsymbol{T}} \left[ \mathrm{tr} \left( \boldsymbol{V}_1^c \boldsymbol{V}_1^{c\top} \boldsymbol{X}^\top \boldsymbol{U}_1 (\boldsymbol{U}_1^\top \boldsymbol{X} \boldsymbol{V}_1)^{\dagger\top} \boldsymbol{V}_1^\top \boldsymbol{V}_2 (\boldsymbol{U}_2^\top \boldsymbol{X} \boldsymbol{V}_2)^\dagger \boldsymbol{U}_2^\top \boldsymbol{X} \boldsymbol{V}_2 \boldsymbol{V}_2^\top \right) \right]}_{(***)}$$

$$+ \underbrace{\mathbb{E}_{\boldsymbol{X}, \boldsymbol{S}, \boldsymbol{T}} \left[ \mathrm{tr} \left( \boldsymbol{V}_1 \boldsymbol{V}_1^\top \boldsymbol{X}^\top \boldsymbol{U}_1 (\boldsymbol{U}_1^\top \boldsymbol{X} \boldsymbol{V}_1)^{\dagger\top} \boldsymbol{V}_1^\top \boldsymbol{V}_2 (\boldsymbol{U}_2^\top \boldsymbol{X} \boldsymbol{V}_2)^\dagger \boldsymbol{U}_2^\top \boldsymbol{X} \boldsymbol{V}_2 \boldsymbol{V}_2^\top \right) \right]}_{(****)} .$$

We notice that

$$(**) = \mathbb{E}_{\boldsymbol{X}, \boldsymbol{S}, \boldsymbol{T}} \left[ \mathrm{tr} \left( \boldsymbol{V}_1 \boldsymbol{V}_1^\top \boldsymbol{V}_2 (\boldsymbol{U}_2^\top \boldsymbol{X} \boldsymbol{V}_2)^\dagger \boldsymbol{U}_2^\top \boldsymbol{X} \boldsymbol{V}_2^c \boldsymbol{V}_2^{c\top} \right) \right] = \boldsymbol{0},$$

where we used the fact that $\boldsymbol{X} \boldsymbol{V}_2^c \boldsymbol{V}_2^{c\top}$ is independent of the rest part and has zero mean. Similarly, $(***) = \boldsymbol{0}$. It is also straightforward to see $(****) = |\boldsymbol{\nu}_1 \cap \boldsymbol{\nu}_2|$. It remains to evaluate $(*)$. We continue to use the notation $\boldsymbol{X}^\vee = \Lambda^\vee \boldsymbol{X}$ and $\boldsymbol{X}^\wedge = \Lambda^\wedge \boldsymbol{X}$. We first notice that

$$\mathbb{E}_{\boldsymbol{T}} \left[ \boldsymbol{U}_1 (\boldsymbol{U}_1^\top \boldsymbol{X} \boldsymbol{V}_1)^{\dagger\top} \boldsymbol{V}_1^\top \boldsymbol{V}_2 (\boldsymbol{U}_2^\top \boldsymbol{X} \boldsymbol{V}_2)^\dagger \boldsymbol{U}_2^\top \right]$$
$$= \mathbb{E}_{\boldsymbol{T}} \left[ (\Pi_{\boldsymbol{X}^\wedge \boldsymbol{V}_1} + \boldsymbol{X}^\wedge \boldsymbol{V}_1 (\boldsymbol{V}_1^\top \boldsymbol{X}^{\wedge\top} \boldsymbol{X}^\wedge \boldsymbol{V}_1)^{-1} \boldsymbol{V}_1^\top \boldsymbol{X}^{\wedge\top}) \boldsymbol{U}_1 (\boldsymbol{U}_1^\top \boldsymbol{X} \boldsymbol{V}_1)^{\dagger\top} \boldsymbol{V}_1^\top \boldsymbol{V}_2 (\boldsymbol{U}_2^\top \boldsymbol{X} \boldsymbol{V}_2)^\dagger \boldsymbol{U}_2^\top \right].$$

The first term is a zero matrix since

$$\mathbb{E}_{\boldsymbol{T}} \left[ \Pi_{\boldsymbol{X}^\wedge \boldsymbol{V}_1} \boldsymbol{U}_1 (\boldsymbol{U}_1^\top \boldsymbol{X} \boldsymbol{V}_1)^{\dagger\top} \right] = \mathbb{E}_{\boldsymbol{T}} \left[ \Pi_{\boldsymbol{X}^\wedge \boldsymbol{V}_1} \boldsymbol{U}_1 \boldsymbol{U}_1^\top \boldsymbol{X} \boldsymbol{V}_1 (\boldsymbol{V}_1^\top \boldsymbol{X}^\top \boldsymbol{U}_1 \boldsymbol{U}_1^\top \boldsymbol{X} \boldsymbol{V}_1)^{-1} \right]$$
$$= \Pi_{\boldsymbol{X}^\wedge \boldsymbol{V}_1} \boldsymbol{X}^\wedge \boldsymbol{V}_1 (\boldsymbol{V}_1^\top \boldsymbol{X}^\top \boldsymbol{U}_1 \boldsymbol{U}_1^\top \boldsymbol{X} \boldsymbol{V}_1)^{-1} = \boldsymbol{0},$$

since the distribution of $\boldsymbol{U}_1 \boldsymbol{U}_1^\top$ is independent of the rest conditioned on we know $|\boldsymbol{\mu}_1|$ and $|\boldsymbol{\mu}_1 \cap \boldsymbol{\mu}_2|$. The second term can be computed as

$$\mathbb{E}_{\boldsymbol{T}} \left[ (\boldsymbol{X}^\wedge \boldsymbol{V}_1 (\boldsymbol{V}_1^\top \boldsymbol{X}^{\wedge\top} \boldsymbol{X}^\wedge \boldsymbol{V}_1)^{-1} \boldsymbol{V}_1^\top \boldsymbol{X}^{\wedge\top} \boldsymbol{U}_1 (\boldsymbol{U}_1^\top \boldsymbol{X} \boldsymbol{V}_1)^{\dagger\top} \boldsymbol{V}_1^\top \boldsymbol{V}_2 (\boldsymbol{U}_2^\top \boldsymbol{X} \boldsymbol{V}_2)^\dagger \boldsymbol{U}_2^\top \right]$$
$$= \mathbb{E}_{\boldsymbol{T}} \left[ (\boldsymbol{X}^\wedge \boldsymbol{V}_1 (\boldsymbol{V}_1^\top \boldsymbol{X}^{\wedge\top} \boldsymbol{X}^\wedge \boldsymbol{V}_1)^{-1} \boldsymbol{V}_1^\top \boldsymbol{X}^\top \boldsymbol{U}_1 (\boldsymbol{U}_1^\top \boldsymbol{X} \boldsymbol{V}_1)^{\dagger\top} \boldsymbol{V}_1^\top \boldsymbol{V}_2 (\boldsymbol{U}_2^\top \boldsymbol{X} \boldsymbol{V}_2)^\dagger \boldsymbol{U}_2^\top \right]$$
$$= \mathbb{E}_{\boldsymbol{T}} \left[ (\boldsymbol{X}^\wedge \boldsymbol{V}_1 (\boldsymbol{V}_1^\top \boldsymbol{X}^{\wedge\top} \boldsymbol{X}^\wedge \boldsymbol{V}_1)^{-1} \boldsymbol{V}_1^\top \boldsymbol{V}_2 (\boldsymbol{U}_2^\top \boldsymbol{X} \boldsymbol{V}_2)^\dagger \boldsymbol{U}_2^\top \right],$$

where we used (27) in the first equality. Via exactly the same strategy, the above quantity reduces to

$$\mathbb{E}_{\boldsymbol{T}} \left[ (\boldsymbol{X}^\wedge \boldsymbol{V}_1 (\boldsymbol{V}_1^\top \boldsymbol{X}^{\wedge\top} \boldsymbol{X}^\wedge \boldsymbol{V}_1)^{-1} \boldsymbol{V}_1^\top \boldsymbol{V}_2 (\boldsymbol{U}_2^\top \boldsymbol{X} \boldsymbol{V}_2)^\dagger \boldsymbol{U}_2^\top \right]$$
$$= \mathbb{E}_{\boldsymbol{T}} \left[ (\boldsymbol{X}^\wedge \boldsymbol{V}_1 (\boldsymbol{V}_1^\top \boldsymbol{X}^{\wedge\top} \boldsymbol{X}^\wedge \boldsymbol{V}_1)^{-1} \boldsymbol{V}_1^\top \boldsymbol{V}_2 (\boldsymbol{V}_2^\top \boldsymbol{X}^{\vee\top} \boldsymbol{X}^\vee \boldsymbol{V}_2)^{-1} \boldsymbol{V}_2^\top \boldsymbol{X}^{\vee\top} \right] = \mathbb{E}_{\boldsymbol{T}} \left[ (\boldsymbol{V}_1^\top \boldsymbol{X}^{\wedge\top})^\dagger \boldsymbol{V}_1^\top \boldsymbol{V}_2 (\boldsymbol{X}^\vee \boldsymbol{V}_2)^\dagger \right].$$

Bringing this into $(*)$, we get

$$(*) = \mathbb{E}_{\boldsymbol{X},\boldsymbol{S},\boldsymbol{T}} \left[ \mathrm{tr} \left( \boldsymbol{V}_1^c \boldsymbol{V}_1^{c\top} \boldsymbol{X}^\top (\boldsymbol{V}_1^\top \boldsymbol{X}^{\wedge\top})^\dagger \boldsymbol{V}_1^\top \boldsymbol{V}_2 (\boldsymbol{X}^\vee \boldsymbol{V}_2)^\dagger \boldsymbol{X} \boldsymbol{V}_2^c \boldsymbol{V}_2^{c\top} \right) \right]$$

Let $\boldsymbol{V}_{1\cap2}$, $\boldsymbol{V}_{1\cup2}$, $\boldsymbol{V}_{1/2}$, and $\boldsymbol{V}_{2/1}$ be the selection matrix associated to $\boldsymbol{\nu}_1 \cap \boldsymbol{\nu}_2$, $\boldsymbol{\nu}_1 \cup \boldsymbol{\nu}_2$, $\boldsymbol{\nu}_1/\boldsymbol{\nu}_2$, and $\boldsymbol{\nu}_2/\boldsymbol{\nu}_1$, respectively. Since order-invariant permutation is allowed, we arrive

$$\begin{aligned}
(*) =& \mathbb{E}_{\boldsymbol{X},S,T} \left[ \mathrm{tr}(\boldsymbol{V}_{1\cup2}^c \boldsymbol{V}_{1\cup2}^{c\top} \boldsymbol{X}^\top (\boldsymbol{V}_1^\top \boldsymbol{X}^{\wedge\top})^\dagger \boldsymbol{V}_1^\top \boldsymbol{V}_2 (\boldsymbol{X}^\vee \boldsymbol{V}_2)^\dagger \boldsymbol{X} \boldsymbol{V}_{1\cup2}^c \boldsymbol{V}_{1\cup2}^{c\top}) \right] \\
=& \mathbb{E}_{\boldsymbol{X},S,T} \left[ \mathrm{tr}(\boldsymbol{V}_{1\cup2}^c \boldsymbol{V}_{1\cup2}^{c\top} \boldsymbol{X}^\top (\boldsymbol{V}_1^\top \boldsymbol{X}^{\wedge\top})^\dagger \boldsymbol{V}_1^\top \boldsymbol{V}_{1\cap2} \boldsymbol{V}_{1\cap2}^\top \boldsymbol{V}_2 (\boldsymbol{X}^\vee \boldsymbol{V}_2)^\dagger \boldsymbol{X} \boldsymbol{V}_{1\cup2}^c \boldsymbol{V}_{1\cup2}^{c\top}) \right] \\
=& \mathbb{E}_{\boldsymbol{X},S,T} \left[ \mathrm{tr}(\boldsymbol{V}_{1\cup2}^c \boldsymbol{V}_{1\cup2}^{c\top} \boldsymbol{X}^\top (\boldsymbol{V}_{1\cap2}^\top \boldsymbol{X}^{\wedge\top})^\dagger (\boldsymbol{X}^\vee \boldsymbol{V}_{1\cap2})^\dagger \boldsymbol{X} \boldsymbol{V}_{1\cup2}^c \boldsymbol{V}_{1\cup2}^{c\top}) \right] .
\end{aligned}$$

where in the last step we used the first argument of Lemma D.4. Note that this is applicable since $\boldsymbol{X} \boldsymbol{V}_{1\cup2}^c \boldsymbol{V}_{1\cup2}^{c\top}$ is independent of both $\boldsymbol{X} \boldsymbol{V}_{1/2}$ and $\boldsymbol{X} \boldsymbol{V}_{2/1}$. We first note that

$$\mathbb{E}_{\boldsymbol{X} \boldsymbol{V}_{1\cup2}^c \boldsymbol{V}_{1\cup2}^{c\top}} \left[ \boldsymbol{X} \boldsymbol{V}_{1\cup2}^c \boldsymbol{V}_{1\cup2}^{c\top} \boldsymbol{V}_{1\cup2}^c \boldsymbol{V}_{1\cup2}^{c\top} \boldsymbol{X}^\top \right] = |\boldsymbol{\nu}_1^c \cap \boldsymbol{\nu}_2^c| \cdot I_n.$$

Also, there holds

$$\begin{aligned}
\mathrm{tr}((\boldsymbol{V}_{1\cap2}^\top \boldsymbol{X}^{\wedge\top})^\dagger (\boldsymbol{X}^\vee \boldsymbol{V}_{1\cap2})^\dagger) &= \mathrm{tr}((\boldsymbol{V}_{1\cap2}^\top \boldsymbol{X}^{\wedge\top} \boldsymbol{X}^\wedge \boldsymbol{V}_{1\cap2})^{-1} \boldsymbol{V}_{1\cap2}^\top \boldsymbol{X}^{\wedge\top} \boldsymbol{X}^\vee \boldsymbol{V}_{1\cap2} (\boldsymbol{V}_{1\cap2}^\top \boldsymbol{X}^{\vee\top} \boldsymbol{X}^\vee \boldsymbol{V}_{1\cap2})^{-1}) \\
&= \mathrm{tr}((\boldsymbol{V}_{1\cap2}^\top \boldsymbol{X}^{\wedge\top} \boldsymbol{X}^\wedge \boldsymbol{V}_{1\cap2})^{-1} \boldsymbol{V}_{1\cap2}^\top \boldsymbol{X}^{\wedge\top} \boldsymbol{X}^\wedge \boldsymbol{V}_{1\cap2} (\boldsymbol{V}_{1\cap2}^\top \boldsymbol{X}^{\vee\top} \boldsymbol{X}^\vee \boldsymbol{V}_{1\cap2})^{-1}) \\
&= \mathrm{tr}((\boldsymbol{V}_{1\cap2}^\top \boldsymbol{X}^{\vee\top} \boldsymbol{X}^\vee \boldsymbol{V}_{1\cap2})^{-1}) = \frac{|\boldsymbol{\nu}_1 \cap \boldsymbol{\nu}_2|}{\mathrm{tr}(\Lambda^\vee) - |\boldsymbol{\nu}_1 \cap \boldsymbol{\nu}_2| - 1}.
\end{aligned}$$

They together yield

$$(*) = \frac{|\boldsymbol{\nu}_1^c \cap \boldsymbol{\nu}_2^c| \cdot |\boldsymbol{\nu}_1 \cap \boldsymbol{\nu}_2|}{\mathrm{tr}(\Lambda^\vee) - |\boldsymbol{\nu}_1 \cap \boldsymbol{\nu}_2| - 1}.$$

Thus,

$$(\%) = (*) + (****) = \frac{|\boldsymbol{\nu}_1^c \cap \boldsymbol{\nu}_2^c| \cdot |\boldsymbol{\nu}_1 \cap \boldsymbol{\nu}_2|}{\mathrm{tr}(\Lambda^\vee) - |\boldsymbol{\nu}_1 \cap \boldsymbol{\nu}_2| - 1} + |\boldsymbol{\nu}_1 \cap \boldsymbol{\nu}_2|.$$

As for the second conclusion, it is still straightforward that $(**) = (***) = 0$. Also, $(****) = |\gamma_1|$. For $(****)$, we have

$$\begin{aligned}
(*) =& \mathbb{E}_{\boldsymbol{X},\boldsymbol{S},\boldsymbol{T}} \left[ \mathrm{tr} \left( \boldsymbol{V}_1^c \boldsymbol{V}_1^{c\top} \boldsymbol{X}^\top \boldsymbol{U}_1 (\boldsymbol{U}_1^\top \boldsymbol{X} \boldsymbol{V}_1)^{\dagger\top} \boldsymbol{V}_1^\top \boldsymbol{V}_2 (\boldsymbol{U}_2^\top \boldsymbol{X} \boldsymbol{V}_2)^\dagger \boldsymbol{U}_2^\top \boldsymbol{X} \boldsymbol{V}_2^c \boldsymbol{V}_2^{c\top} \right) \right] \\
=& \mathbb{E}_{\boldsymbol{X},\boldsymbol{S},\boldsymbol{T}} \left[ \mathrm{tr} \left( \boldsymbol{V}_2^c \boldsymbol{V}_2^{c\top} \boldsymbol{X}^\top \boldsymbol{U}_1 (\boldsymbol{U}_1^\top \boldsymbol{X} \boldsymbol{V}_1)^{\dagger\top} \boldsymbol{V}_1^\top \boldsymbol{V}_2 (\boldsymbol{U}_2^\top \boldsymbol{X} \boldsymbol{V}_2)^\dagger \boldsymbol{U}_2^\top \boldsymbol{X} \boldsymbol{V}_2^c \boldsymbol{V}_2^{c\top} \right) \right] \\
=& \mathbb{E}_{\boldsymbol{X},\boldsymbol{S},\boldsymbol{T}} \left[ \mathrm{tr} \left( \boldsymbol{V}_2^c \boldsymbol{V}_2^{c\top} \boldsymbol{X}^\top \boldsymbol{U}_1 (\boldsymbol{U}_1^\top \boldsymbol{X} \boldsymbol{V}_1)^{\dagger\top} \boldsymbol{V}_1^\top \boldsymbol{V}_1 (\boldsymbol{U}_2^\top \boldsymbol{X} \boldsymbol{V}_1)^\dagger \boldsymbol{U}_2^\top \boldsymbol{X} \boldsymbol{V}_2^c \boldsymbol{V}_2^{c\top} \right) \right] ,
\end{aligned}$$

where the last follows from Lemma D.4. Lemma D.3 yields

$$\mathbb{E}_{\boldsymbol{X} \boldsymbol{V}_2^c} \left[ \boldsymbol{X} \boldsymbol{V}_2^c \boldsymbol{V}_2^{c\top} \boldsymbol{V}_2^c \boldsymbol{V}_2^{c\top} \boldsymbol{X}^\top \right] = (d - |\boldsymbol{\nu}_2|) I_n.$$

Then,

$$\begin{aligned}
(*) =& (d - |\boldsymbol{\nu}_2|) \mathbb{E}_{\boldsymbol{X},\boldsymbol{S},\boldsymbol{T}} \left[ \mathrm{tr} \left( \boldsymbol{U}_1 (\boldsymbol{U}_1^\top \boldsymbol{X} \boldsymbol{V}_1)^{\dagger\top} \boldsymbol{V}_1^\top \boldsymbol{V}_1 (\boldsymbol{U}_2^\top \boldsymbol{X} \boldsymbol{V}_1)^\dagger \boldsymbol{U}_2^\top \right) \right] \\
=& (d - |\boldsymbol{\nu}_2|) \mathbb{E}_{\boldsymbol{X},\boldsymbol{S},\boldsymbol{T}} \left[ \mathrm{tr} \left( \boldsymbol{U}_1 (\boldsymbol{U}_1^\top \boldsymbol{X} \boldsymbol{V}_1)^{\dagger\top} \boldsymbol{V}_1^\top \boldsymbol{V}_1 (\boldsymbol{U}_2^\top \boldsymbol{X} \boldsymbol{V}_1)^\dagger \boldsymbol{U}_1^\top \right) \right] \\
=& (d - |\boldsymbol{\nu}_2|) \mathbb{E}_{\boldsymbol{X},\boldsymbol{S},\boldsymbol{T}} \left[ \mathrm{tr} \left( \boldsymbol{U}_1 \boldsymbol{U}_1^\top \boldsymbol{X} \boldsymbol{V}_1 (\boldsymbol{V}_1^\top \boldsymbol{X}^\top \boldsymbol{U}_1 \boldsymbol{U}_1^\top \boldsymbol{X} \boldsymbol{V}_1)^{-1} \boldsymbol{V}_1^\top \boldsymbol{V}_1 (\boldsymbol{V}_1^\top \boldsymbol{X}^\top \boldsymbol{U}_2 \boldsymbol{U}_2^\top \boldsymbol{X} \boldsymbol{V}_1)^{-1} \boldsymbol{V}_1^\top \boldsymbol{X}^\top \right) \right] \\
=& (d - |\boldsymbol{\nu}_2|) \mathbb{E}_{\boldsymbol{X},\boldsymbol{S},\boldsymbol{T}} \left[ \mathrm{tr} \left( (\boldsymbol{V}_1^\top \boldsymbol{X}^\top \boldsymbol{U}_2 \boldsymbol{U}_2^\top \boldsymbol{X} \boldsymbol{V}_1)^{-1} \right) \right] = \frac{|\boldsymbol{\nu}_1| (d - |\boldsymbol{\nu}_2|)}{|\boldsymbol{\mu}_2| - |\boldsymbol{\nu}_1| - 1}.
\end{aligned}$$

**The second circumstance** Then we proceed to the second circumstance where $|\boldsymbol{\nu}_1| > |\boldsymbol{\mu}_1|, |\boldsymbol{\nu}_2| > |\boldsymbol{\mu}_2|$. This is an analog to the first circumstance, yet we operate on the dual dimension. We switch the notation to use the notation $\boldsymbol{X}^\vee = \boldsymbol{X} \Theta^\vee$

and $\boldsymbol{X}^\wedge = \boldsymbol{X}\Theta^\wedge$. Let $\boldsymbol{U}_{1\cap2}$, $\boldsymbol{U}_{1\cup2}$, $\boldsymbol{U}_{1/2}$, and $\boldsymbol{U}_{2/1}$ be the selection matrix associated to $\boldsymbol{\mu}_1 \cap \boldsymbol{\mu}_2$, $\boldsymbol{\mu}_1 \cup \boldsymbol{\mu}_2$, $\boldsymbol{\mu}_1/\boldsymbol{\mu}_2$, and $\boldsymbol{\mu}_2/\boldsymbol{\mu}_1$, respectively. We first write

$$\underbrace{(\%) = \mathbb{E}_{\boldsymbol{X},\boldsymbol{S},\boldsymbol{T}}\left[\operatorname{tr}\left(\boldsymbol{X}^\top \boldsymbol{U}_2^c \boldsymbol{U}_2^{c\top}\boldsymbol{U}_1(\boldsymbol{U}_1^\top \boldsymbol{X}\boldsymbol{V}_1)^{\dagger\top}\boldsymbol{V}_1^\top\boldsymbol{V}_2(\boldsymbol{U}_2^\top\boldsymbol{X}\boldsymbol{V}_2)^\dagger\boldsymbol{U}_2^\top\boldsymbol{U}_1^c\boldsymbol{U}_1^{c\top}\boldsymbol{X}\right)\right]}_{(*)}$$

$$+\underbrace{\mathbb{E}_{\boldsymbol{X},\boldsymbol{S},\boldsymbol{T}}\left[\operatorname{tr}\left(\boldsymbol{X}^\top \boldsymbol{U}_2^c \boldsymbol{U}_2^{c\top}\boldsymbol{U}_1(\boldsymbol{U}_1^\top \boldsymbol{X}\boldsymbol{V}_1)^{\dagger\top}\boldsymbol{V}_1^\top\boldsymbol{V}_2(\boldsymbol{U}_2^\top\boldsymbol{X}\boldsymbol{V}_2)^\dagger\boldsymbol{U}_2^\top\boldsymbol{U}_1\boldsymbol{U}_1^\top X\right)\right]}_{(**)}$$

$$+\underbrace{\mathbb{E}_{\boldsymbol{X},\boldsymbol{S},\boldsymbol{T}}\left[\operatorname{tr}\left(\boldsymbol{X}^\top \boldsymbol{U}_2 \boldsymbol{U}_2^{\top}\boldsymbol{U}_1(\boldsymbol{U}_1^\top \boldsymbol{X}\boldsymbol{V}_1)^{\dagger\top}\boldsymbol{V}_1^\top\boldsymbol{V}_2(\boldsymbol{U}_2^\top\boldsymbol{X}\boldsymbol{V}_2)^\dagger\boldsymbol{U}_2^\top\boldsymbol{U}_1^c\boldsymbol{U}_1^{c\top}X\right)\right]}_{(***)}$$

$$+\underbrace{\mathbb{E}_{\boldsymbol{X},\boldsymbol{S},\boldsymbol{T}}\left[\operatorname{tr}\left(\boldsymbol{X}^\top \boldsymbol{U}_2 \boldsymbol{U}_2^{\top}\boldsymbol{U}_1(\boldsymbol{U}_1^\top \boldsymbol{X}\boldsymbol{V}_1)^{\dagger\top}\boldsymbol{V}_1^\top\boldsymbol{V}_2(\boldsymbol{U}_2^\top\boldsymbol{X}\boldsymbol{V}_2)^\dagger\boldsymbol{U}_2^\top\boldsymbol{U}_1\boldsymbol{U}_1^\top \boldsymbol{X}\right)\right]}_{(****)}$$

$(i)$ We begin with $(*)$. Due to a similar reasoning in the first circumstance, we know that $\mathbb{E}_S\left[(\boldsymbol{U}_1^\top \boldsymbol{X}\boldsymbol{V}_1)^{\dagger\top}\boldsymbol{V}_1^\top\Pi_{\boldsymbol{X}^{\vee\top}\boldsymbol{U}_1}\right] = \boldsymbol{0}$. Thus, we can do the following transformation

$$(*) = \mathbb{E}_{\boldsymbol{X},\boldsymbol{S},\boldsymbol{T}}\left[\operatorname{tr}\left(\boldsymbol{X}^\top \boldsymbol{U}_2^c\boldsymbol{U}_2^{c\top}\boldsymbol{U}_1(\boldsymbol{U}_1^\top\boldsymbol{X}\boldsymbol{V}_1)^{\dagger\top}\boldsymbol{V}_1^\top\boldsymbol{X}^{\vee\top}\boldsymbol{U}_1(\boldsymbol{U}_1^\top\boldsymbol{X}^\vee\boldsymbol{X}^{\vee\top}\boldsymbol{U}_1)^{-1}\boldsymbol{U}_1^\top\boldsymbol{X}^\vee\boldsymbol{V}_2(\boldsymbol{U}_2^\top\boldsymbol{X}\boldsymbol{V}_2)^\dagger\boldsymbol{U}_2^\top\boldsymbol{U}_1^c\boldsymbol{U}_1^{c\top}\boldsymbol{X}\right)\right]$$
$$= \mathbb{E}_{\boldsymbol{X},\boldsymbol{S},\boldsymbol{T}}\left[\operatorname{tr}\left(\boldsymbol{X}^\top \boldsymbol{U}_2^c\boldsymbol{U}_2^{c\top}\boldsymbol{U}_1(\boldsymbol{U}_1^\top\boldsymbol{X}^\vee\boldsymbol{X}^{\vee\top}\boldsymbol{U}_1)^{-1}\boldsymbol{U}_1^\top\boldsymbol{X}^\vee\boldsymbol{V}_2(\boldsymbol{U}_2^\top\boldsymbol{X}\boldsymbol{V}_2)^\dagger\boldsymbol{U}_2^\top\boldsymbol{U}_1^c\boldsymbol{U}_1^{c\top}\boldsymbol{X}\right)\right].$$

Via exactly the same strategy, the above quantity reduces to

$$(*) = \mathbb{E}_{\boldsymbol{X},\boldsymbol{S},\boldsymbol{T}}\left[\operatorname{tr}\left(\boldsymbol{X}^\top \boldsymbol{U}_2^c\boldsymbol{U}_2^{c\top}\boldsymbol{U}_1(\boldsymbol{U}_1^\top\boldsymbol{X}^\vee\boldsymbol{X}^{\vee\top}\boldsymbol{U}_1)^{-1}\boldsymbol{U}_1^\top\boldsymbol{X}^\vee\boldsymbol{X}^{\wedge\top}\boldsymbol{U}_2(\boldsymbol{U}_2^\top\boldsymbol{X}^\wedge\boldsymbol{X}^{\wedge\top}\boldsymbol{U}_2)^{-1}\boldsymbol{U}_2^\top\boldsymbol{U}_1^c\boldsymbol{U}_1^{c\top}\boldsymbol{X}\right)\right]$$
$$= \mathbb{E}_{\boldsymbol{X},\boldsymbol{S},\boldsymbol{T}}\left[\operatorname{tr}\left(\boldsymbol{X}^\top \boldsymbol{U}_2^c\boldsymbol{U}_2^{c\top}\boldsymbol{U}_1(\boldsymbol{U}_1^\top\boldsymbol{X}^\vee\boldsymbol{X}^{\vee\top}\boldsymbol{U}_1)^{-1}\boldsymbol{U}_1^\top\boldsymbol{X}^\wedge\boldsymbol{X}^{\wedge\top}\boldsymbol{U}_2(\boldsymbol{U}_2^\top\boldsymbol{X}^\wedge\boldsymbol{X}^{\wedge\top}\boldsymbol{U}_2)^{-1}\boldsymbol{U}_2^\top\boldsymbol{U}_1^c\boldsymbol{U}_1^{c\top}\boldsymbol{X}\right)\right]. \quad (30)$$

We also note that

$$\mathbb{E}_{\boldsymbol{X}\Theta^{\vee c}}\left[\boldsymbol{U}_2^\top\boldsymbol{X}\boldsymbol{X}^\top\boldsymbol{U}_2^c\boldsymbol{U}_2^{c\top}\right] = \boldsymbol{U}_2^\top\boldsymbol{X}^\vee\boldsymbol{X}^{\vee\top}\boldsymbol{U}_2^c\boldsymbol{U}_2^{c\top} + \mathbb{E}_{\boldsymbol{X}\Theta^{\vee c}}\left[\boldsymbol{U}_2^\top\boldsymbol{X}\Theta^{\vee c}\Theta^{\vee c}\boldsymbol{X}^\top\boldsymbol{U}_2^c\boldsymbol{U}_2^{c\top}\right]$$
$$= \boldsymbol{U}_2^\top\boldsymbol{X}^\vee\boldsymbol{X}^{\vee\top}\boldsymbol{U}_2^c\boldsymbol{U}_2^{c\top} + \operatorname{tr}(\Theta^{\vee c})\mathbb{E}_{\boldsymbol{X}\Theta^{\vee c}}\left[\boldsymbol{U}_2^\top\boldsymbol{U}_2^c\boldsymbol{U}_2^{c\top}\right] = \boldsymbol{U}_2^\top\boldsymbol{X}^\vee\boldsymbol{X}^{\vee\top}\boldsymbol{U}_2^c\boldsymbol{U}_2^{c\top}.$$

This means we can turn the $\boldsymbol{X}$s on the side into

$$(*) = \mathbb{E}_{\boldsymbol{X},\boldsymbol{S},\boldsymbol{T}}\left[\operatorname{tr}\left(\boldsymbol{X}^{\vee\top}\boldsymbol{U}_2^c\boldsymbol{U}_2^{c\top}\boldsymbol{U}_1(\boldsymbol{U}_1^\top\boldsymbol{X}^\vee\boldsymbol{X}^{\vee\top}\boldsymbol{U}_1)^{-1}\boldsymbol{U}_1^\top\boldsymbol{X}^\wedge\boldsymbol{X}^{\wedge\top}\boldsymbol{U}_2(\boldsymbol{U}_2^\top\boldsymbol{X}^\wedge\boldsymbol{X}^{\wedge\top}\boldsymbol{U}_2)^{-1}\boldsymbol{U}_2^\top\boldsymbol{U}_1^c\boldsymbol{U}_1^{c\top}\boldsymbol{X}^\vee\right)\right].$$

We further decompose $I_n = \boldsymbol{U}_2\boldsymbol{U}_2^\top + \boldsymbol{U}_2^c\boldsymbol{U}_2^{c\top}$ and have

$$(*) = \mathbb{E}_{\boldsymbol{X},\boldsymbol{S},\boldsymbol{T}}\left[\operatorname{tr}\left(\boldsymbol{X}^{\vee\top}\boldsymbol{U}_2^c\boldsymbol{U}_2^{c\top}\boldsymbol{U}_1(\boldsymbol{U}_1^\top\boldsymbol{X}^\vee\boldsymbol{X}^{\vee\top}\boldsymbol{U}_1)^{-1}\boldsymbol{U}_1^\top\boldsymbol{U}_2\boldsymbol{U}_2^\top\boldsymbol{X}^\wedge\boldsymbol{X}^{\wedge\top}\boldsymbol{U}_2(\boldsymbol{U}_2^\top\boldsymbol{X}^\wedge\boldsymbol{X}^{\wedge\top}\boldsymbol{U}_2)^{-1}\boldsymbol{U}_2^\top\boldsymbol{U}_1^c\boldsymbol{U}_1^{c\top}\boldsymbol{X}^\vee\right)\right]$$
$$+\mathbb{E}_{\boldsymbol{X},\boldsymbol{S},\boldsymbol{T}}\left[\operatorname{tr}\left(\boldsymbol{X}^{\vee\top}\boldsymbol{U}_2^c\boldsymbol{U}_2^{c\top}\boldsymbol{U}_1(\boldsymbol{U}_1^\top\boldsymbol{X}^\vee\boldsymbol{X}^{\vee\top}\boldsymbol{U}_1)^{-1}\boldsymbol{U}_1^\top\boldsymbol{U}_2^c\boldsymbol{U}_2^{c\top}\boldsymbol{X}^\wedge\boldsymbol{X}^{\wedge\top}\boldsymbol{U}_2(\boldsymbol{U}_2^\top\boldsymbol{X}^\wedge\boldsymbol{X}^{\wedge\top}\boldsymbol{U}_2)^{-1}\boldsymbol{U}_2^\top\boldsymbol{U}_1^c\boldsymbol{U}_1^{c\top}\boldsymbol{X}^\vee\right)\right].$$

The first term becomes

$$\mathbb{E}_{\boldsymbol{X},\boldsymbol{S},\boldsymbol{T}}\left[\operatorname{tr}\left(\boldsymbol{X}^{\vee\top}\boldsymbol{U}_2^c\boldsymbol{U}_2^{c\top}\boldsymbol{U}_1(\boldsymbol{U}_1^\top\boldsymbol{X}^\vee\boldsymbol{X}^{\vee\top}\boldsymbol{U}_1)^{-1}\boldsymbol{U}_1^\top\boldsymbol{U}_2\boldsymbol{U}_2^\top\boldsymbol{X}^\wedge\boldsymbol{X}^{\wedge\top}\boldsymbol{U}_2(\boldsymbol{U}_2^\top\boldsymbol{X}^\wedge\boldsymbol{X}^{\wedge\top}\boldsymbol{U}_2)^{-1}\boldsymbol{U}_2^\top\boldsymbol{U}_1^c\boldsymbol{U}_1^{c\top}\boldsymbol{X}^\vee\right)\right]$$
$$= \mathbb{E}_{\boldsymbol{X},\boldsymbol{S},\boldsymbol{T}}\left[\operatorname{tr}\left(\boldsymbol{X}^{\vee\top}\boldsymbol{U}_2^c\boldsymbol{U}_2^{c\top}\boldsymbol{U}_1(\boldsymbol{U}_1^\top\boldsymbol{X}^\vee\boldsymbol{X}^{\vee\top}\boldsymbol{U}_1)^{-1}\boldsymbol{U}_1^\top\boldsymbol{U}_2\boldsymbol{U}_2^\top\boldsymbol{U}_1^c\boldsymbol{U}_1^{c\top}\boldsymbol{X}^\vee\right)\right] = \boldsymbol{0}. \quad (31)$$

For the second term, we write $\boldsymbol{U}_{1\cap2}^\top\boldsymbol{X}^\vee = \boldsymbol{X}_{1\cap2}^\vee$, $\boldsymbol{U}_{2/1}^\top\boldsymbol{X}^\vee = \boldsymbol{X}_{2/1}^\vee$, $\boldsymbol{U}_{1/2}^\top\boldsymbol{X}^\vee = \boldsymbol{X}_{1/2}^\vee$, $\boldsymbol{U}_{1\cup2}^{c\top}\boldsymbol{X}^\vee = \boldsymbol{X}_{1^c\cap2^c}^\vee$. W.o.l.g., we assume $(\boldsymbol{\mu}_1 \cap \boldsymbol{\mu}_2, \boldsymbol{\mu}_1/\boldsymbol{\mu}_2, \boldsymbol{\mu}_2/\boldsymbol{\mu}_1, \boldsymbol{\mu}_1^c \cap \boldsymbol{\mu}_2^c) = (1,\ldots,d)$, which means $\boldsymbol{X} = (\boldsymbol{X}_{1\cap2}^{\vee\top}, \boldsymbol{X}_{1/2}^{\vee\top}, \boldsymbol{X}_{2/1}^{\vee\top}, \boldsymbol{X}_{1^c\cap2^c}^{\vee\top})^\top$. Then we have

$$\boldsymbol{U}_1^\top\boldsymbol{X}^\vee\boldsymbol{X}^{\vee\top}\boldsymbol{U}_1 = \begin{pmatrix} X_{1\cap2}^\vee\boldsymbol{X}_{1\cap2}^{\vee\top} & \boldsymbol{X}_{1\cap2}^\vee\boldsymbol{X}_{1/2}^{\vee\top} \\ X_{1/2}^\vee\boldsymbol{X}_{1\cap2}^{\vee\top} & \boldsymbol{X}_{1/2}^\vee\boldsymbol{X}_{1/2}^{\vee\top} \end{pmatrix}.$$

We can compute

$$\boldsymbol{X}^{\vee\top}\boldsymbol{U}_2^c\boldsymbol{U}_2^{c\top}\boldsymbol{U}_1(\boldsymbol{U}_1^\top\boldsymbol{X}^\vee\boldsymbol{X}^{\vee\top}\boldsymbol{U}_1)^{-1}\boldsymbol{U}_1^\top\boldsymbol{U}_2^c\boldsymbol{U}_2^{c\top}\boldsymbol{X}^\vee$$
$$= \boldsymbol{X}_{1/2}^{\vee\top}(\boldsymbol{U}_1^\top\boldsymbol{X}^\vee\boldsymbol{X}^{\vee\top}\boldsymbol{U}_1)^{-1}\boldsymbol{X}_{1/2}^\vee$$
$$= \boldsymbol{X}_{1/2}^{\vee\top}(\boldsymbol{X}_{1/2}^\vee\boldsymbol{X}_{1/2}^{\vee\top} - \boldsymbol{X}_{1/2}^\vee\boldsymbol{X}_{1\cap2}^{\vee\top}(\boldsymbol{X}_{1\cap2}^\vee\boldsymbol{X}_{1\cap2}^{\vee\top})^{-1}\boldsymbol{X}_{1\cap2}^\vee\boldsymbol{X}_{1/2}^{\vee\top})^{-1}\boldsymbol{X}_{1/2}^\vee = \boldsymbol{X}_{1/2}^{\vee\top}(\boldsymbol{X}_{1/2}^\vee\Pi_{\boldsymbol{X}_{1\cap2}^{\vee\top}}\boldsymbol{X}_{1/2}^{\vee\top})^{-1}\boldsymbol{X}_{1/2}^\vee. \quad (32)$$

Since $\boldsymbol{X}_{1/2}^{\vee}$ has i.i.d. standard Gaussian entries, decomposing it into $\boldsymbol{X}_{1/2}^{\vee\top} = \Pi_{\boldsymbol{X}_{1\cap2}^{\vee\top}}\boldsymbol{X}_{1/2}^{\vee\top} + \Pi_{\boldsymbol{X}_{1\cap2}^{\vee c}}\boldsymbol{X}_{1/2}^{\vee\top}$ yields the following, where the cross terms disappear.

$$\mathbb{E}_{\boldsymbol{X}_{1/2}^{\vee\top}}\left[X_{1/2}^{\vee\top}(\boldsymbol{X}_{1/2}^{\vee}\Pi_{\boldsymbol{X}_{1\cap2}^{\vee\top}}\boldsymbol{X}_{1/2}^{\vee\top})^{-1}\boldsymbol{X}_{1/2}^{\vee}\right]$$
$$=\mathbb{E}_{\boldsymbol{X}_{1/2}^{\vee\top}}\left[\Pi_{\boldsymbol{X}_{1\cap2}^{\vee}}\boldsymbol{X}_{1/2}^{\vee\top}(\boldsymbol{X}_{1/2}^{\vee}\Pi_{\boldsymbol{X}_{1\cap2}^{\vee\top}}\boldsymbol{X}_{1/2}^{\vee\top})^{-1}\boldsymbol{X}_{1/2}^{\vee}\Pi_{\boldsymbol{X}_{1\cap2}^{\vee}}\right] + \mathbb{E}_{\boldsymbol{X}_{1/2}^{\vee\top}}\left[\Pi_{\boldsymbol{X}_{1\cap2}^{\vee c}}\boldsymbol{X}_{1/2}^{\vee\top}(\boldsymbol{X}_{1/2}^{\vee}\Pi_{\boldsymbol{X}_{1\cap2}^{\vee\top}}\boldsymbol{X}_{1/2}^{\vee\top})^{-1}\boldsymbol{X}_{1/2}^{\vee}\Pi_{\boldsymbol{X}_{1\cap2}^{\vee c}}\right].$$

The first term has

$$\mathbb{E}_{\boldsymbol{X}_{1/2}^{\vee\top}}\left[\Pi_{\Pi_{\boldsymbol{X}_{1\cap2}^{\vee\top}}\boldsymbol{X}_{1/2}^{\vee\top}}\right] = \frac{|\boldsymbol{\mu}_1/\boldsymbol{\mu}_2|}{\mathrm{tr}(\Theta^{\vee}) - |\boldsymbol{\mu}_1 \cap \boldsymbol{\mu}_2|}\Pi_{\boldsymbol{X}_{1\cap2}^{\vee}},$$

which is by the fact that $\Pi_{\boldsymbol{X}_{1\cap2}^{\vee}}\boldsymbol{X}_{1/2}^{\vee\top}$ is distributed as $\mathcal{N}(0, \Pi_{\boldsymbol{X}_{1\cap2}^{\vee}})$. The second term has

$$\mathbb{E}_{\boldsymbol{X}_{1/2}^{\vee\top}}\left[\Pi_{\boldsymbol{X}_{1\cap2}^{\vee c}}\boldsymbol{X}_{1/2}^{\vee\top}(\boldsymbol{X}_{1/2}^{\vee}\Pi_{\boldsymbol{X}_{1\cap2}^{\vee\top}}\boldsymbol{X}_{1/2}^{\vee\top})^{-1}\boldsymbol{X}_{1/2}^{\vee}\Pi_{\boldsymbol{X}_{1\cap2}^{\vee c}}\right]$$
$$=\mathbb{E}_{\boldsymbol{X}_{1/2}^{\vee\top}}\left[\Pi_{\boldsymbol{X}_{1\cap2}^{\vee c}}\boldsymbol{X}_{1/2}^{\vee\top}\frac{I_{|\boldsymbol{\mu}_1/\boldsymbol{\mu}_2|}}{\mathrm{tr}(\Theta^{\vee}) - |\boldsymbol{\mu}_1 \cap \boldsymbol{\mu}_2| - |\boldsymbol{\mu}_1/\boldsymbol{\mu}_2| - 1}\boldsymbol{X}_{1/2}^{\vee}\Pi_{\boldsymbol{X}_{1\cap2}^{\vee c}}\right]$$
$$=\Pi_{\boldsymbol{X}_{1\cap2}^{\vee c}}\frac{|\boldsymbol{\mu}_1/\boldsymbol{\mu}_2|}{\mathrm{tr}(\Theta^{\vee}) - |\boldsymbol{\mu}_1 \cap \boldsymbol{\mu}_2| - |\boldsymbol{\mu}_1/\boldsymbol{\mu}_2| - 1}$$

due to Lemma D.3. This means the second term of $(*)$ is

$$(\cdot) := \frac{|\boldsymbol{\mu}_1/\boldsymbol{\mu}_2|}{\mathrm{tr}(\Theta^{\vee}) - |\boldsymbol{\mu}_1 \cap \boldsymbol{\mu}_2|}\Pi_{\boldsymbol{X}_{1\cap2}^{\vee}} + \Pi_{\boldsymbol{X}_{1\cap2}^{\vee c}}\frac{|\boldsymbol{\mu}_1/\boldsymbol{\mu}_2|}{\mathrm{tr}(\Theta^{\vee}) - |\boldsymbol{\mu}_1 \cap \boldsymbol{\mu}_2| - |\boldsymbol{\mu}_1/\boldsymbol{\mu}_2| - 1}$$
$$=\frac{|\boldsymbol{\mu}_1/\boldsymbol{\mu}_2|}{\mathrm{tr}(\Theta^{\vee}) - |\boldsymbol{\mu}_1 \cap \boldsymbol{\mu}_2|}I_{\mathrm{tr}(\Theta^{\vee})} + \Pi_{\boldsymbol{X}_{1\cap2}^{\vee c}}\frac{|\boldsymbol{\mu}_1/\boldsymbol{\mu}_2|}{(\mathrm{tr}(\Theta^{\vee}) - |\boldsymbol{\mu}_1 \cap \boldsymbol{\mu}_2| - |\boldsymbol{\mu}_1/\boldsymbol{\mu}_2| - 1)(\mathrm{tr}(\Theta^{\vee}) - |\boldsymbol{\mu}_1 \cap \boldsymbol{\mu}_2|)}. \tag{33}$$

Bringing this into $(*)$, we get

$$(*) = \mathbb{E}_{\boldsymbol{X},\boldsymbol{S},\boldsymbol{T}}\left[\mathrm{tr}\left((\cdot)\boldsymbol{X}^{\wedge\top}\boldsymbol{U}_2(\boldsymbol{U}_2^{\top}\boldsymbol{X}^{\wedge}\boldsymbol{X}^{\wedge\top}\boldsymbol{U}_2)^{-1}\boldsymbol{U}_2^{\top}\boldsymbol{U}_1^c\boldsymbol{U}_1^{c\top}\boldsymbol{X}^{\vee}\right)\right].$$

Note that $\boldsymbol{U}_1^c\boldsymbol{U}_1^{c\top}\boldsymbol{X}^{\vee}\Pi_{\boldsymbol{X}_{1\cap2}^{\vee c}} = \boldsymbol{0}$, and thus

$$(*) = \frac{|\boldsymbol{\mu}_1/\boldsymbol{\mu}_2|}{\mathrm{tr}(\Theta^{\vee}) - |\boldsymbol{\mu}_1 \cap \boldsymbol{\mu}_2|}\mathbb{E}_{\boldsymbol{X},\boldsymbol{S},\boldsymbol{T}}\left[\mathrm{tr}\left(\boldsymbol{X}^{\wedge\top}\boldsymbol{U}_2(\boldsymbol{U}_2^{\top}\boldsymbol{X}^{\wedge}\boldsymbol{X}^{\wedge\top}\boldsymbol{U}_2)^{-1}\boldsymbol{U}_2^{\top}\boldsymbol{U}_1^c\boldsymbol{U}_1^{c\top}\boldsymbol{X}^{\vee}\right)\right].$$

Note that we can change the last $\boldsymbol{X}^{\vee}$ into $\boldsymbol{X}^{\wedge}$ since trace operator can permutate and $\boldsymbol{X}^{\vee}\boldsymbol{X}^{\wedge\top} = \boldsymbol{X}^{\wedge}\boldsymbol{X}^{\wedge\top}$. Following the same notations, we have

$$\boldsymbol{X}^{\wedge\top}\boldsymbol{U}_2(\boldsymbol{U}_2^{\top}\boldsymbol{X}^{\wedge}\boldsymbol{X}^{\wedge\top}\boldsymbol{U}_2)^{-1}\boldsymbol{U}_2^{\top}\boldsymbol{U}_1^c\boldsymbol{U}_1^{c\top}\boldsymbol{X}^{\wedge}$$
$$= - \boldsymbol{X}_{1\cap2}^{\wedge\top}(\boldsymbol{X}_{1\cap2}^{\wedge}\boldsymbol{X}_{1\cap2}^{\wedge\top})^{-1}\boldsymbol{X}_{1\cap2}^{\wedge}\boldsymbol{X}_{2/1}^{\wedge\top}(\boldsymbol{X}_{2/1}^{\wedge}\boldsymbol{X}_{2/1}^{\wedge\top} - \boldsymbol{X}_{2/1}^{\wedge}\boldsymbol{X}_{1\cap2}^{\wedge\top}(\boldsymbol{X}_{1\cap2}^{\wedge}\boldsymbol{X}_{1\cap2}^{\wedge\top})^{-1}\boldsymbol{X}_{1\cap2}^{\wedge}\boldsymbol{X}_{2/1}^{\wedge\top})^{-1}\boldsymbol{X}_{2/1}^{\wedge}$$
$$+ \boldsymbol{X}_{2/1}^{\wedge\top}(\boldsymbol{X}_{2/1}^{\wedge}\boldsymbol{X}_{2/1}^{\wedge\top} - \boldsymbol{X}_{2/1}^{\wedge}\boldsymbol{X}_{1\cap2}^{\wedge\top}(\boldsymbol{X}_{1\cap2}^{\wedge}\boldsymbol{X}_{1\cap2}^{\wedge\top})^{-1}\boldsymbol{X}_{1\cap2}^{\wedge}\boldsymbol{X}_{2/1}^{\wedge\top})^{-1}\boldsymbol{X}_{2/1}^{\wedge}$$
$$=\Pi_{\boldsymbol{X}_{1\cap2}^{\wedge\top}}\boldsymbol{X}_{2/1}^{\wedge\top}(\boldsymbol{X}_{2/1}^{\wedge}\Pi_{\boldsymbol{X}_{1\cap2}^{\wedge\top}}\boldsymbol{X}_{2/1}^{\wedge\top})^{-1}\boldsymbol{X}_{2/1}^{\wedge},$$

whose trace is

$$\mathbb{E}_{\boldsymbol{X},\boldsymbol{S},\boldsymbol{T}}\left[\mathrm{tr}\left(X^{\wedge\top}\boldsymbol{U}_2(\boldsymbol{U}_2^{\top}\boldsymbol{X}^{\wedge}\boldsymbol{X}^{\wedge\top}\boldsymbol{U}_2)^{-1}\boldsymbol{U}_2^{\top}\boldsymbol{U}_1^c\boldsymbol{U}_1^{c\top}\boldsymbol{X}^{\wedge}\right)\right]$$
$$=\mathbb{E}_{\boldsymbol{X},\boldsymbol{S},\boldsymbol{T}}\left[\mathrm{tr}\left(\Pi_{\boldsymbol{X}_{1\cap2}^{\wedge\top}}\boldsymbol{X}_{2/1}^{\wedge\top}(\boldsymbol{X}_{2/1}^{\wedge}\Pi_{\boldsymbol{X}_{1\cap2}^{\wedge\top}}\boldsymbol{X}_{2/1}^{\wedge\top})^{-1}\boldsymbol{X}_{2/1}^{\wedge}\Pi_{\boldsymbol{X}_{1\cap2}^{\wedge\top}}\right)\right] = |\boldsymbol{\mu}_2/\boldsymbol{\mu}_1|.$$

Thus, we reach

$$(*) = \frac{|\boldsymbol{\mu}_1/\boldsymbol{\mu}_2||\boldsymbol{\mu}_2/\boldsymbol{\mu}_1|}{\mathrm{tr}(\Theta^{\vee}) - |\boldsymbol{\mu}_1 \cap \boldsymbol{\mu}_2|}.$$

$(ii)$ For $(**)$, notice that the derivations till (31) and (33) are still valid, except we need to replace $\boldsymbol{U}_1^c \boldsymbol{U}_1^{c\top}$ by $\boldsymbol{U}_1 \boldsymbol{U}_1^\top$, namely

$$(**) = \mathbb{E}_{\boldsymbol{X},\boldsymbol{S},\boldsymbol{T}}\left[\operatorname{tr}\left(\boldsymbol{X}^{\vee\top}\boldsymbol{U}_2^c\boldsymbol{U}_2^{c\top}\boldsymbol{U}_1(\boldsymbol{U}_1^\top\boldsymbol{X}^\vee\boldsymbol{X}^{\vee\top}\boldsymbol{U}_1)^{-1}\boldsymbol{U}_1^\top\boldsymbol{U}_2\boldsymbol{U}_2^\top\boldsymbol{U}_1\boldsymbol{U}_1^\top\boldsymbol{X}^\vee\right)\right]$$
$$+ \frac{|\boldsymbol{\mu}_1/\boldsymbol{\mu}_2|}{\operatorname{tr}(\Theta^\vee) - |\boldsymbol{\mu}_1 \cap \boldsymbol{\mu}_2| - |\boldsymbol{\mu}_1/\boldsymbol{\mu}_2| - 1}\mathbb{E}_{\boldsymbol{X},\boldsymbol{S},\boldsymbol{T}}\left[\operatorname{tr}\left(\boldsymbol{X}^{\wedge\top}\boldsymbol{U}_2(\boldsymbol{U}_2^\top\boldsymbol{X}^\wedge\boldsymbol{X}^{\wedge\top}\boldsymbol{U}_2)^{-1}\boldsymbol{U}_2^\top\boldsymbol{U}_1\boldsymbol{U}_1^\top\boldsymbol{X}^\vee\right)\right].$$

Under the same notation, we can compute

$$\boldsymbol{X}^{\vee\top}\boldsymbol{U}_2^c\boldsymbol{U}_2^{c\top}\boldsymbol{U}_1(\boldsymbol{U}_1^\top\boldsymbol{X}^\vee\boldsymbol{X}^{\vee\top}\boldsymbol{U}_1)^{-1}\boldsymbol{U}_1^\top\boldsymbol{U}_2\boldsymbol{U}_2^\top\boldsymbol{X}^\vee = -\boldsymbol{X}_{1/2}^{\vee\top}(\boldsymbol{X}_{1/2}^\vee\Pi_{\boldsymbol{X}_{1\cap 2}^{\vee\top}}\boldsymbol{X}_{1/2}^{\vee\top})^{-1}\boldsymbol{X}_{1/2}^\vee\Pi_{\boldsymbol{X}_{1\cap 2}^{\vee c\top}}$$

and

$$\boldsymbol{X}^{\wedge\top}\boldsymbol{U}_2(\boldsymbol{U}_2^\top\boldsymbol{X}^\wedge\boldsymbol{X}^{\wedge\top}\boldsymbol{U}_2)^{-1}\boldsymbol{U}_2^\top\boldsymbol{U}_1\boldsymbol{U}_1^\top\boldsymbol{X}^\vee$$
$$= \boldsymbol{X}_{1\cap 2}^{\wedge\top}(\boldsymbol{X}_{1\cap 2}^\wedge\boldsymbol{X}_{1\cap 2}^{\wedge\top})^{-1}\boldsymbol{X}_{1\cap 2}^\wedge$$
$$+ \boldsymbol{X}_{1\cap 2}^{\wedge\top}(\boldsymbol{X}_{1\cap 2}^\wedge\boldsymbol{X}_{1\cap 2}^{\wedge\top})^{-1}\boldsymbol{X}_{1\cap 2}^\wedge\boldsymbol{X}_{2/1}^{\wedge\top}(\boldsymbol{X}_{2/1}^\wedge\Pi_{\boldsymbol{X}_{1\cap 2}^{\wedge\top}}\boldsymbol{X}_{2/1}^{\wedge\top})^{-1}\boldsymbol{X}_{2/1}^\wedge\boldsymbol{X}_{1\cap 2}^{\wedge\top}(\boldsymbol{X}_{1\cap 2}^\wedge\boldsymbol{X}_{1\cap 2}^{\wedge\top})^{-1}\boldsymbol{X}_{1\cap 2}^\wedge$$
$$- \boldsymbol{X}_{2/1}^{\wedge\top}(\boldsymbol{X}_{2/1}^\wedge\Pi_{\boldsymbol{X}_{1\cap 2}^{\wedge\top}}\boldsymbol{X}_{2/1}^{\wedge\top})^{-1}\boldsymbol{X}_{2/1}^\wedge\boldsymbol{X}_{1\cap 2}^{\wedge\top}(\boldsymbol{X}_{1\cap 2}^\wedge\boldsymbol{X}_{1\cap 2}^{\wedge\top})^{-1}\boldsymbol{X}_{1\cap 2}^\wedge$$
$$= \boldsymbol{X}_{1\cap 2}^{\wedge\top}(\boldsymbol{X}_{1\cap 2}^\wedge\boldsymbol{X}_{1\cap 2}^{\wedge\top})^{-1}\boldsymbol{X}_{1\cap 2}^\wedge + \Pi_{\boldsymbol{X}_{1\cap 2}^{\wedge\top}}\boldsymbol{X}_{2/1}^{\wedge\top}(\boldsymbol{X}_{2/1}^\wedge\Pi_{\boldsymbol{X}_{1\cap 2}^{\wedge\top}}\boldsymbol{X}_{2/1}^{\wedge\top})^{-1}\boldsymbol{X}_{2/1}^\wedge\Pi_{\boldsymbol{X}_{1\cap 2}^{\wedge c\top}}.$$

Bringing in the same calculations as in (32) yields $(**) = 0$.

$(iii)$ The calculation of $(***)$ follows symmetrically to $(*)$.

$(iv)$ We directly apply Lemma D.4 to have

$$\mathbb{E}_{\boldsymbol{X},\boldsymbol{S},\boldsymbol{T}}\left[\operatorname{tr}\left(\boldsymbol{X}^\top\boldsymbol{U}_2\boldsymbol{U}_2^\top\boldsymbol{U}_1(\boldsymbol{U}_1^\top\boldsymbol{X}\boldsymbol{V}_1)^{\dagger\top}\boldsymbol{V}_1^\top\boldsymbol{V}_2(\boldsymbol{U}_2^\top\boldsymbol{X}\boldsymbol{V}_2)^\dagger\boldsymbol{U}_2^\top\boldsymbol{U}_1\boldsymbol{U}_1^\top\boldsymbol{X}\right)\right]$$
$$= \mathbb{E}_{\boldsymbol{X},\boldsymbol{S},\boldsymbol{T}}\left[\operatorname{tr}\left(\boldsymbol{X}^\top\boldsymbol{U}_{1\cap 2}(\boldsymbol{U}_{1\cap 2}^\top\boldsymbol{X}\boldsymbol{V}_1)^{\dagger\top}\boldsymbol{V}_1^\top\boldsymbol{V}_2(\boldsymbol{U}_{1\cap 2}^\top\boldsymbol{X}\boldsymbol{V}_2)^\dagger\boldsymbol{U}_{1\cap 2}^\top\boldsymbol{X}\right)\right],$$

which is followed by the transformations similar to (30) to have

$$(***) = \mathbb{E}_{\boldsymbol{X},\boldsymbol{S},\boldsymbol{T}}\left[\operatorname{tr}\left(\boldsymbol{X}^\top\boldsymbol{U}_{1\cap 2}(\boldsymbol{U}_{1\cap 2}^\top\boldsymbol{X}^\vee\boldsymbol{X}^{\vee\top}\boldsymbol{U}_{1\cap 2})^{-1}\boldsymbol{U}_{1\cap 2}^\top\boldsymbol{X}\right)\right]$$
$$= \mathbb{E}_{\boldsymbol{X},\boldsymbol{S},\boldsymbol{T}}\left[\operatorname{tr}\left(\boldsymbol{X}^{\vee\top}\boldsymbol{U}_{1\cap 2}(\boldsymbol{U}_{1\cap 2}^\top\boldsymbol{X}^\vee\boldsymbol{X}^{\vee\top}\boldsymbol{U}_{1\cap 2})^{-1}\boldsymbol{U}_{1\cap 2}^\top\boldsymbol{X}^\vee\right)\right]$$
$$+ \mathbb{E}_{\boldsymbol{X},\boldsymbol{S},\boldsymbol{T}}\left[\operatorname{tr}\left(\boldsymbol{X}^\top\boldsymbol{U}_{1\cap 2}(\boldsymbol{U}_{1\cap 2}^\top\boldsymbol{X}^\vee\boldsymbol{X}^{\vee\top}\boldsymbol{U}_{1\cap 2})^{-1}\boldsymbol{U}_{1\cap 2}^\top\boldsymbol{X}(I_d - \Theta^\vee)\right)\right].$$

The first term is $|\boldsymbol{\mu}_1 \cap \boldsymbol{\mu}_2|$, while the second is

$$\frac{(d - \operatorname{tr}(\Theta^\vee))|\boldsymbol{\mu}_1 \cap \boldsymbol{\mu}_2|}{\operatorname{tr}(\Theta^\vee) - |\boldsymbol{\mu}_1 \cap \boldsymbol{\mu}_2| - 1}.$$

They together yield

$$(****) = |\boldsymbol{\mu}_1 \cap \boldsymbol{\mu}_2|\frac{d - |\boldsymbol{\mu}_1 \cap \boldsymbol{\mu}_2| - 1}{\operatorname{tr}(\Theta^\vee) - |\boldsymbol{\mu}_1 \cap \boldsymbol{\mu}_2| - 1}.$$

Thus,

$$(\%) = (*) + (****) = \frac{|\boldsymbol{\mu}_1/\boldsymbol{\mu}_2||\boldsymbol{\mu}_2/\boldsymbol{\mu}_1|}{\operatorname{tr}(\Theta^\vee) - |\boldsymbol{\mu}_1 \cap \boldsymbol{\mu}_2|} + |\boldsymbol{\mu}_1 \cap \boldsymbol{\mu}_2|\frac{d - |\boldsymbol{\mu}_1 \cap \boldsymbol{\mu}_2| - 1}{\operatorname{tr}(\Theta^\vee) - |\boldsymbol{\mu}_1 \cap \boldsymbol{\mu}_2| - 1}.$$

For the second conclusion, we notice that $(*) = (**) = (***) = 0$ is obvious. For $(****)$, we have

$$(****) = \mathbb{E}_{\boldsymbol{X},\boldsymbol{S},\boldsymbol{T}}\left[\operatorname{tr}\left(\boldsymbol{X}^\top\boldsymbol{U}_1(\boldsymbol{U}_1^\top\boldsymbol{X}\boldsymbol{V}_1)^{\dagger\top}\boldsymbol{V}_1^\top\boldsymbol{V}_1(\boldsymbol{U}_2^\top\boldsymbol{X}\boldsymbol{V}_2)^\dagger\boldsymbol{U}_2^\top\boldsymbol{U}_1\boldsymbol{U}_1^\top\boldsymbol{X}\right)\right]$$
$$= \mathbb{E}_{\boldsymbol{X},\boldsymbol{S},\boldsymbol{T}}\left[\operatorname{tr}\left(\boldsymbol{X}^\top\boldsymbol{U}_1(\boldsymbol{U}_1^\top\boldsymbol{X}\boldsymbol{V}_1)^{\dagger\top}\boldsymbol{V}_1^\top\boldsymbol{V}_1(\boldsymbol{U}_1^\top\boldsymbol{X}\boldsymbol{V}_2)^\dagger\boldsymbol{U}_1^\top\boldsymbol{X}\right)\right]$$

by Lemma D.4, from where we proceed to have

$$(****) = \mathbb{E}_{\boldsymbol{X},\boldsymbol{S},\boldsymbol{T}}\left[\operatorname{tr}\left(\boldsymbol{X}^\top\boldsymbol{U}_1(\boldsymbol{U}_1^\top\boldsymbol{X}\boldsymbol{V}_1\boldsymbol{V}_1^\top\boldsymbol{X}^\top\boldsymbol{U}_1)^{-1}\boldsymbol{U}_1^\top\boldsymbol{X}\boldsymbol{V}_1\boldsymbol{V}_1^\top\boldsymbol{X}^\top\boldsymbol{U}_1(\boldsymbol{U}_1^\top\boldsymbol{X}\boldsymbol{V}_2\boldsymbol{V}_2^\top\boldsymbol{X}^\top\boldsymbol{U}_1)^{-1}\boldsymbol{U}_1^\top\boldsymbol{X}\right)\right]$$
$$= \mathbb{E}_{\boldsymbol{X},\boldsymbol{S},\boldsymbol{T}}\left[\operatorname{tr}\left(\boldsymbol{X}^\top\boldsymbol{U}_1(\boldsymbol{U}_1^\top\boldsymbol{X}\boldsymbol{V}_2\boldsymbol{V}_2^\top\boldsymbol{X}^\top\boldsymbol{U}_1)^{-1}\boldsymbol{U}_1^\top\boldsymbol{X}\right)\right]$$
$$= |\boldsymbol{\mu}_1| + \frac{|\boldsymbol{\mu}_1|(d - |\boldsymbol{\nu}_2|)}{|\boldsymbol{\nu}_2| - |\boldsymbol{\mu}_1| - 1}.$$

$\square$

## D.3. Proofs for Appendix D.1

*Proof of Proposition D.1.* Recall that

$$
\begin{aligned}
&\mathrm{tr}\left(\left(g(\boldsymbol{x}) - g'(\boldsymbol{x})\right)^{\top}\left(g(\boldsymbol{x}) - g'(\boldsymbol{x})\right)\right)\\
=&\frac{1}{B^2}\sum_{b_1,b_2=1}^{B}\underbrace{\mathrm{tr}\left(\left(\boldsymbol{V}_{b_1}(\boldsymbol{U}_{b_1}^{\top}\boldsymbol{X}\boldsymbol{V}_{b_1})^{\dagger}\boldsymbol{U}_{b_1}^{\top} - \boldsymbol{V}'_{b_1}(\boldsymbol{U}'^{\top}_{b_1}\boldsymbol{X}\boldsymbol{V}'_{b_1})^{\dagger}\boldsymbol{U}'^{\top}_{b_1}\right)^{\top}\left(\boldsymbol{V}_{b_2}(\boldsymbol{U}_{b_2}^{\top}\boldsymbol{X}\boldsymbol{V}_{b_2})^{\dagger}\boldsymbol{U}_{b_2}^{\top} - \boldsymbol{V}'_{b_2}(\boldsymbol{U}'^{\top}_{b_2}\boldsymbol{X}\boldsymbol{V}'_{b_2})^{\dagger}\boldsymbol{U}'^{\top}_{b_2}\right)\right)}_{V_{b_1,b_2}}.
\end{aligned}
$$

There are $B(B-1)$ cross terms which have $b_1 \neq b_2$, and $B$ squared terms with $b_1 = b_2$.

**Underparameterized case** For instance removal, the squared terms become

$$
V_{b,b} = \mathrm{tr}\left(\left(\boldsymbol{V}_b(\boldsymbol{U}_b^{\top}\boldsymbol{X}\boldsymbol{V}_b)^{\dagger}\boldsymbol{U}_b^{\top} - \boldsymbol{V}_b(\boldsymbol{U}'^{\top}_b\boldsymbol{X}\boldsymbol{V}_b)^{\dagger}\boldsymbol{U}'^{\top}_b\right)^{\top}\left(\boldsymbol{V}_b(\boldsymbol{U}_b^{\top}\boldsymbol{X}\boldsymbol{V}_b)^{\dagger}\boldsymbol{U}_b^{\top} - \boldsymbol{V}_b(\boldsymbol{U}'^{\top}_b\boldsymbol{X}\boldsymbol{V}_b)^{\dagger}\boldsymbol{U}'^{\top}_b\right)\right),
$$

where $\boldsymbol{U}_b$ and $\boldsymbol{U}'_b$ are independent. We apply (25) in Lemma D.5 to have

$$
\begin{aligned}
&\mathbb{E}_{\boldsymbol{X},\boldsymbol{S},\boldsymbol{T}}\left[\mathrm{tr}\left(\left(\boldsymbol{V}_b(\boldsymbol{U}_b^{\top}\boldsymbol{X}\boldsymbol{V}_b)^{\dagger}\boldsymbol{U}_b^{\top}\right)^{\top}\left(\boldsymbol{V}_b(\boldsymbol{U}_b^{\top}\boldsymbol{X}\boldsymbol{V}_b)^{\dagger}\boldsymbol{U}_b^{\top}\right)\right)\right]\\
=&\mathbb{E}_{\boldsymbol{X},\boldsymbol{S},\boldsymbol{T}}\left[\mathrm{tr}\left(\left(\boldsymbol{V}_b(\boldsymbol{U}'^{\top}_b\boldsymbol{X}\boldsymbol{V}_b)^{\dagger}\boldsymbol{U}'^{\top}_b\right)^{\top}\left(\boldsymbol{V}_b(\boldsymbol{U}'^{\top}_b\boldsymbol{X}\boldsymbol{V}_b)^{\dagger}\boldsymbol{U}'^{\top}_b\right)\right)\right] = \frac{|\boldsymbol{\nu}_b|}{|\boldsymbol{\mu}_b| - |\boldsymbol{\nu}_b| - 1}.
\end{aligned}
$$

For the other two cross terms, we treat them as only have $\boldsymbol{\nu}_b$ columns, and apply (24) in Lemma D.5 to have

$$
\mathbb{E}_{\boldsymbol{X},\boldsymbol{S},\boldsymbol{T}}\left[\mathrm{tr}\left(\left(\boldsymbol{V}_b(\boldsymbol{U}_b^{\top}\boldsymbol{X}\boldsymbol{V}_b)^{\dagger}\boldsymbol{U}_b^{\top}\right)^{\top}\left(\boldsymbol{V}_b(\boldsymbol{U}'^{\top}_b\boldsymbol{X}\boldsymbol{V}_b)^{\dagger}\boldsymbol{U}'^{\top}_b\right)\right)\right] = \frac{|\boldsymbol{\nu}_b|}{n - |\boldsymbol{\nu}_b| - 1}.
$$

Combined together, this is

$$
\mathbb{E}_{\boldsymbol{X},\boldsymbol{S},\boldsymbol{T}}\left[V_{b,b}\right] = 2\frac{|\boldsymbol{\nu}_b|}{|\boldsymbol{\mu}_b| - |\boldsymbol{\nu}_b| - 1} - 2\frac{|\boldsymbol{\nu}_b|}{n - |\boldsymbol{\nu}_b| - 1}.
$$

Taking the limit, this is

$$
\lim_{n,d\to\infty}\mathbb{E}_{\boldsymbol{X},\boldsymbol{S},\boldsymbol{T}}\left[V_{b,b}\right] = 2\frac{\gamma q}{p - \gamma q} - 2\frac{\gamma q}{1 - \gamma q} = \frac{2\gamma q(1-p)}{(p - \gamma q)(1 - \gamma q)}.
$$

For feature removal, the squared term becomes

$$
V_{b,b} = \mathrm{tr}\left(\left(\boldsymbol{V}_b(\boldsymbol{U}_b^{\top}\boldsymbol{X}\boldsymbol{V}_b)^{\dagger}\boldsymbol{U}_b^{\top} - \boldsymbol{V}'_b(\boldsymbol{U}_b^{\top}\boldsymbol{X}\boldsymbol{V}'_b)^{\dagger}\boldsymbol{U}_b^{\top}\right)^{\top}\left(\boldsymbol{V}_b(\boldsymbol{U}_b^{\top}\boldsymbol{X}\boldsymbol{V}_b)^{\dagger}\boldsymbol{U}_b^{\top} - \boldsymbol{V}'_b(\boldsymbol{U}_b^{\top}\boldsymbol{X}\boldsymbol{V}'_b)^{\dagger}\boldsymbol{U}_b^{\top}\right)\right),
$$

where $\boldsymbol{V}_b$ and $\boldsymbol{V}'_b$ are independent. We apply (25) in Lemma D.5 to have

$$
\begin{aligned}
&\mathbb{E}_{\boldsymbol{X},\boldsymbol{S},\boldsymbol{T}}\left[\mathrm{tr}\left(\left(\boldsymbol{V}_b(\boldsymbol{U}_b^{\top}\boldsymbol{X}\boldsymbol{V}_b)^{\dagger}\boldsymbol{U}_b^{\top}\right)^{\top}\left(\boldsymbol{V}_b(\boldsymbol{U}_b^{\top}\boldsymbol{X}\boldsymbol{V}_b)^{\dagger}\boldsymbol{U}_b^{\top}\right)\right)\right]\\
=&\mathbb{E}_{\boldsymbol{X},\boldsymbol{S},\boldsymbol{T}}\left[\mathrm{tr}\left(\left(\boldsymbol{V}'_b(\boldsymbol{U}_b^{\top}\boldsymbol{X}\boldsymbol{V}'_b)^{\dagger}\boldsymbol{U}_b^{\top}\right)^{\top}\left(\boldsymbol{V}'_b(\boldsymbol{U}_b^{\top}\boldsymbol{X}\boldsymbol{V}'_b)^{\dagger}\boldsymbol{U}_b^{\top}\right)\right)\right] = \frac{|\boldsymbol{\nu}_b|}{|\boldsymbol{\mu}_b| - |\boldsymbol{\nu}_b| - 1}.
\end{aligned}
$$

For the other two cross terms, we treat them as only have $\boldsymbol{\mu}_b$ rows, and apply (24) in Lemma D.5 to have

$$
\mathbb{E}_{\boldsymbol{X},\boldsymbol{S},\boldsymbol{T}}\left[\mathrm{tr}\left(\left(\boldsymbol{V}_b(\boldsymbol{U}_b^{\top}\boldsymbol{X}\boldsymbol{V}_b)^{\dagger}\boldsymbol{U}_b^{\top}\right)^{\top}\left(\boldsymbol{V}'_b(\boldsymbol{U}_b^{\top}\boldsymbol{X}\boldsymbol{V}'_b)^{\dagger}\boldsymbol{U}_b^{\top}\right)\right)\right] = \frac{|\boldsymbol{\nu}_b \cap \boldsymbol{\nu}'_b|}{np - |\boldsymbol{\nu}_b \cap \boldsymbol{\nu}'_b| - 1}.
$$

Combined together, this is

$$
\mathbb{E}_{\boldsymbol{X},\boldsymbol{S},\boldsymbol{T}}\left[V_{b,b}\right] = 2\frac{|\boldsymbol{\nu}_b|}{|\boldsymbol{\mu}_b| - |\boldsymbol{\nu}_b| - 1} - 2\frac{|\boldsymbol{\nu}_b \cap \boldsymbol{\nu}'_b|}{np - |\boldsymbol{\nu}_b \cap \boldsymbol{\nu}'_b| - 1}.
$$

Taking the limit, this is

$$\lim_{n,d\to\infty} \mathbb{E}_{\boldsymbol{X},\boldsymbol{S},\boldsymbol{T}}\left[V_{b,b}\right] = 2\frac{\gamma q}{p - \gamma q} - 2\frac{\gamma q^2}{p - \gamma q^2} = \frac{2\gamma pq(1-q)}{(p-\gamma q)(p-\gamma q^2)}.$$

For the cross terms ($b_1 \neq b_2$) for instance removal, we have

$$\lim_{n,d\to\infty} n \cdot \mathbb{E}_{\boldsymbol{X},\boldsymbol{S},\boldsymbol{T}}\left[V_{b_1,b_2}\right] = \lim_{n,d\to\infty} n \cdot \mathbb{E}_{\boldsymbol{X},\boldsymbol{S},\boldsymbol{T}}\left[\frac{|\boldsymbol{\nu}_1 \cap \boldsymbol{\nu}_2|}{n - 1 - |\boldsymbol{\nu}_1 \cap \boldsymbol{\nu}_2| - 1} - \frac{|\boldsymbol{\nu}_1 \cap \boldsymbol{\nu}_2|}{n - |\boldsymbol{\nu}_1 \cap \boldsymbol{\nu}_2| - 1}\right]$$

$$= \lim_{n,d\to\infty} n \cdot \left[\frac{\frac{s^2}{d}}{n - \frac{s^2}{d} - 2} - \frac{\frac{s^2}{d}}{n - \frac{s^2}{d} - 1}\right] = \frac{\gamma q^2}{(1 - \gamma q^2)^2}.$$

As for the cross terms ($b_1 \neq b_2$) for feature removal, we have

$$\lim_{n,d\to\infty} n \cdot \mathbb{E}_{\boldsymbol{X},\boldsymbol{S},\boldsymbol{T}}\left[V_{b_1,b_2}\right] = \lim_{n,d\to\infty} n \cdot \mathbb{E}_{\boldsymbol{X},\boldsymbol{S},\boldsymbol{T}}\left[\frac{|\boldsymbol{\nu}_1' \cap \boldsymbol{\nu}_2'|}{n - |\boldsymbol{\nu}_1' \cap \boldsymbol{\nu}_2'| - 1} - \frac{|\boldsymbol{\nu}_1 \cap \boldsymbol{\nu}_2|}{n - |\boldsymbol{\nu}_1 \cap \boldsymbol{\nu}_2| - 1}\right]$$

$$= \lim_{n,d\to\infty} n \cdot \left[\frac{\frac{s^2}{d-1}}{n - \frac{s^2}{d-1} - 1} - \frac{\frac{s^2}{d}}{n - \frac{s^2}{d} - 1}\right] = \frac{q^2}{(1 - \gamma q^2)^2}.$$

**Overparameterized case**  For instance removal, the squared term becomes

$$V_{b,b} = \mathrm{tr}\left(\left(\boldsymbol{V}_b(\boldsymbol{U}_b^\top \boldsymbol{X}\boldsymbol{V}_b)^\dagger \boldsymbol{U}_b^\top - \boldsymbol{V}_b(\boldsymbol{U}_b'^\top \boldsymbol{X}\boldsymbol{V}_b)^\dagger \boldsymbol{U}_b'^\top\right)^\top \left(\boldsymbol{V}_b(\boldsymbol{U}_b^\top \boldsymbol{X}\boldsymbol{V}_b)^\dagger \boldsymbol{U}_b^\top - \boldsymbol{V}_b(\boldsymbol{U}_b'^\top \boldsymbol{X}\boldsymbol{V}_b)^\dagger \boldsymbol{U}_b'^\top\right)\right),$$

where $\boldsymbol{U}_b$ and $\boldsymbol{U}_b'$ are independent. We apply (25) in Lemma D.5 to have

$$\mathbb{E}_{\boldsymbol{X},\boldsymbol{S},\boldsymbol{T}}\left[\mathrm{tr}\left(\left(\boldsymbol{V}_b(\boldsymbol{U}_b^\top \boldsymbol{X}\boldsymbol{V}_b)^\dagger \boldsymbol{U}_b^\top\right)^\top \left(\boldsymbol{V}_b(\boldsymbol{U}_b^\top \boldsymbol{X}\boldsymbol{V}_b)^\dagger \boldsymbol{U}_b^\top\right)\right)\right]$$

$$= \mathbb{E}_{\boldsymbol{X},\boldsymbol{S},\boldsymbol{T}}\left[\mathrm{tr}\left(\left(\boldsymbol{V}_b(\boldsymbol{U}_b'^\top \boldsymbol{X}\boldsymbol{V}_b)^\dagger \boldsymbol{U}_b'^\top\right)^\top \left(\boldsymbol{V}_b(\boldsymbol{U}_b'^\top \boldsymbol{X}\boldsymbol{V}_b)^\dagger \boldsymbol{U}_b'^\top\right)\right)\right] = \frac{|\boldsymbol{\mu}_b|}{|\boldsymbol{\nu}_b| - |\boldsymbol{\mu}_b| - 1}.$$

For the other two cross terms, we treat them as only have $\boldsymbol{\nu}_b$ columns, and apply (24) in Lemma D.5 to have

$$\mathbb{E}_{\boldsymbol{X},\boldsymbol{S},\boldsymbol{T}}\left[\mathrm{tr}\left(\left(\boldsymbol{V}_b(\boldsymbol{U}_b^\top \boldsymbol{X}\boldsymbol{V}_b)^\dagger \boldsymbol{U}_b^\top\right)^\top \left(\boldsymbol{V}_b'(\boldsymbol{U}_b^\top \boldsymbol{X}\boldsymbol{V}_b')^\dagger \boldsymbol{U}_b^\top\right)\right)\right] = \frac{|\boldsymbol{\mu}_b \cap \boldsymbol{\mu}_b'|}{qd - |\boldsymbol{\mu}_b \cap \boldsymbol{\mu}_b'| - 1}.$$

Combined together, this is

$$\mathbb{E}_{\boldsymbol{X},\boldsymbol{S},\boldsymbol{T}}\left[V_{b,b}\right] = 2\frac{|\boldsymbol{\mu}_b|}{|\boldsymbol{\nu}_b| - |\boldsymbol{\mu}_b| - 1} - 2\frac{|\boldsymbol{\mu}_b \cap \boldsymbol{\mu}_b'|}{qd - |\boldsymbol{\mu}_b \cap \boldsymbol{\mu}_b'| - 1}.$$

Taking the limit, this is

$$\lim_{n,d\to\infty} \mathbb{E}_{\boldsymbol{X},\boldsymbol{S},\boldsymbol{T}}\left[V_{b,b}\right] = 2\frac{p}{\gamma q - p} - 2\frac{p^2}{\gamma q - p^2} = \frac{2\gamma q(p - p^2)}{(\gamma q - p)(\gamma q - p^2)}.$$

For feature removal, the squared terms become

$$V_{b,b} = \mathrm{tr}\left(\left(\boldsymbol{V}_b(\boldsymbol{U}_b^\top \boldsymbol{X}\boldsymbol{V}_b)^\dagger \boldsymbol{U}_b^\top - \boldsymbol{V}_b'(\boldsymbol{U}_b^\top \boldsymbol{X}\boldsymbol{V}_b')^\dagger \boldsymbol{U}_b^\top\right)^\top \left(\boldsymbol{V}_b(\boldsymbol{U}_b^\top \boldsymbol{X}\boldsymbol{V}_b)^\dagger \boldsymbol{U}_b^\top - \boldsymbol{V}_b'(\boldsymbol{U}_b^\top \boldsymbol{X}\boldsymbol{V}_b')^\dagger \boldsymbol{U}_b^\top\right)\right),$$

where $\boldsymbol{V}_b$ and $\boldsymbol{V}_b'$ are independent. We apply (25) in Lemma D.5 to have

$$\mathbb{E}_{\boldsymbol{X},\boldsymbol{S},\boldsymbol{T}}\left[\mathrm{tr}\left(\left(\boldsymbol{V}_b(\boldsymbol{U}_b^\top \boldsymbol{X}\boldsymbol{V}_b)^\dagger \boldsymbol{U}_b^\top\right)^\top \left(\boldsymbol{V}_b(\boldsymbol{U}_b^\top \boldsymbol{X}\boldsymbol{V}_b)^\dagger \boldsymbol{U}_b^\top\right)\right)\right]$$

$$= \mathbb{E}_{\boldsymbol{X},\boldsymbol{S},\boldsymbol{T}}\left[\mathrm{tr}\left(\left(\boldsymbol{V}_b'(\boldsymbol{U}_b^\top \boldsymbol{X}\boldsymbol{V}_b')^\dagger \boldsymbol{U}_b^\top\right)^\top \left(\boldsymbol{V}_b'(\boldsymbol{U}_b^\top \boldsymbol{X}\boldsymbol{V}_b')^\dagger \boldsymbol{U}_b^\top\right)\right)\right] = \frac{|\boldsymbol{\mu}_b|}{|\boldsymbol{\nu}_b| - |\boldsymbol{\mu}_b| - 1}.$$

For the other two cross terms, we treat them as only have $\boldsymbol{\mu}_b$ rows, and apply (24) in Lemma D.5 to have

$$\mathbb{E}_{\boldsymbol{X},\boldsymbol{S},\boldsymbol{T}} \left[ \mathrm{tr} \left( \left( \boldsymbol{V}_b (\boldsymbol{U}_b^\top \boldsymbol{X} \boldsymbol{V}_b)^\dagger \boldsymbol{U}_b^\top \right)^\top \left( \boldsymbol{V}_b' (\boldsymbol{U}_b^\top \boldsymbol{X} \boldsymbol{V}_b')^\dagger \boldsymbol{U}_b^\top \right) \right) \right] = \frac{|\boldsymbol{\mu}_b|}{d - |\boldsymbol{\mu}_b| - 1}.$$

Combined together, this is

$$\mathbb{E}_{\boldsymbol{X},\boldsymbol{S},\boldsymbol{T}} \left[ V_{b,b} \right] = 2 \frac{|\boldsymbol{\mu}_b|}{|\boldsymbol{\nu}_b| - |\boldsymbol{\mu}_b| - 1} - 2 \frac{|\boldsymbol{\mu}_b|}{d - |\boldsymbol{\mu}_b| - 1}.$$

Taking the limit, this is

$$\lim_{n,d\to\infty} \mathbb{E}_{\boldsymbol{X},\boldsymbol{S},\boldsymbol{T}} \left[ V_{b,b} \right] = 2 \frac{p}{\gamma q - p} - 2 \frac{p}{\gamma - p} = \frac{2\gamma p(1-q)}{(\gamma q - p)(\gamma - p)}.$$

For the cross terms ($b_1 \neq b_2$) with instance removal, we have

$$\lim_{n,d\to\infty} n \cdot \mathbb{E}_{\boldsymbol{X},\boldsymbol{S},\boldsymbol{T}} \left[ V_{b_1,b_2} \right] = \lim_{n,d\to\infty} n \cdot \mathbb{E}_{\boldsymbol{X},\boldsymbol{S},\boldsymbol{T}} \left[ \frac{|\boldsymbol{\mu}_1' \cap \boldsymbol{\mu}_2'|}{d - |\boldsymbol{\mu}_1' \cap \boldsymbol{\mu}_2'| - 1} - \frac{|\boldsymbol{\mu}_1 \cap \boldsymbol{\mu}_2|}{d - |\boldsymbol{\mu}_1 \cap \boldsymbol{\mu}_2| - 1} \right]$$

$$= \lim_{n,d\to\infty} n \cdot \left[ \frac{\frac{m^2}{n-1}}{d - \frac{m^2}{n-1} - 1} - \frac{\frac{m^2}{n}}{d - \frac{m^2}{n} - 1} \right] = \frac{\gamma p^2}{(\gamma - p^2)^2}.$$

For the cross terms ($b_1 \neq b_2$) with feature removal, we have

$$\lim_{n,d\to\infty} n \cdot \mathbb{E}_{\boldsymbol{X},\boldsymbol{S},\boldsymbol{T}} \left[ V_{b_1,b_2} \right] = \lim_{n,d\to\infty} n \cdot \mathbb{E}_{\boldsymbol{X},\boldsymbol{S},\boldsymbol{T}} \left[ \frac{|\boldsymbol{\mu}_1 \cap \boldsymbol{\mu}_2|}{d - 1 - |\boldsymbol{\mu}_1 \cap \boldsymbol{\mu}_2| - 1} - \frac{|\boldsymbol{\mu}_1 \cap \boldsymbol{\mu}_2|}{d - |\boldsymbol{\mu}_1 \cap \boldsymbol{\mu}_2| - 1} \right]$$

$$= \lim_{n,d\to\infty} n \cdot \left[ \frac{\frac{m^2}{n}}{d - 1 - \frac{m^2}{n} - 1} - \frac{\frac{m^2}{n}}{d - \frac{m^2}{n} - 1} \right] = \frac{p^2}{(\gamma - p^2)^2}.$$

$\square$

*Proof of Proposition D.2.* Recall that

$$\mathbb{E}_{\boldsymbol{\beta}^*} \left[ \boldsymbol{\beta}^{*\top} \boldsymbol{X}^\top \left( g(\boldsymbol{x}) - g'(\boldsymbol{x}) \right)^\top \left( g(\boldsymbol{x}) - g'(\boldsymbol{x}) \right) \boldsymbol{X} \boldsymbol{\beta}^* \right]$$

$$= \frac{1}{dB^2} \sum_{b_1,b_2=1}^{B} \underbrace{\mathrm{tr} \left( \left( \boldsymbol{V}_{b_1} (\boldsymbol{U}_{b_1}^\top \boldsymbol{X} \boldsymbol{V}_{b_1})^\dagger \boldsymbol{U}_{b_1}^\top \boldsymbol{X} - \boldsymbol{V}_{b_1}' (\boldsymbol{U}_{b_1}'^\top \boldsymbol{X} \boldsymbol{V}_{b_1}')^\dagger \boldsymbol{U}_{b_1}'^\top \boldsymbol{X} \right)^\top \left( \boldsymbol{V}_{b_2} (\boldsymbol{U}_{b_2}^\top \boldsymbol{X} \boldsymbol{V}_{b_2})^\dagger \boldsymbol{U}_{b_2}^\top \boldsymbol{X} - \boldsymbol{V}_{b_2}' (\boldsymbol{U}_{b_2}'^\top \boldsymbol{X} \boldsymbol{V}_{b_2}')^\dagger \boldsymbol{U}_{b_2}'^\top \boldsymbol{X} \right) \right)}_{B_{b_1,b_2}}.$$

There are $B(B-1)$ terms which have $b_1 \neq b_2$, and $B$ terms with $b_1 = b_2$.

**Underparameterized case** For instance removal, the squared terms become

$$B_{b,b} = \mathrm{tr} \left( \left( \boldsymbol{V}_b (\boldsymbol{U}_b^\top \boldsymbol{X} \boldsymbol{V}_b)^\dagger \boldsymbol{U}_b^\top \boldsymbol{X} - \boldsymbol{V}_b (\boldsymbol{U}_b'^\top \boldsymbol{X} \boldsymbol{V}_b)^\dagger \boldsymbol{U}_b'^\top \boldsymbol{X} \right)^\top \left( \boldsymbol{V}_b (\boldsymbol{U}_b^\top \boldsymbol{X} \boldsymbol{V}_b)^\dagger \boldsymbol{U}_b^\top \boldsymbol{X} - \boldsymbol{V}_b (\boldsymbol{U}_b'^\top \boldsymbol{X} \boldsymbol{V}_b)^\dagger \boldsymbol{U}_b'^\top \boldsymbol{X} \right) \right),$$

where $\boldsymbol{U}_b$ and $\boldsymbol{U}_b'$ are independent. We apply (29) in Lemma D.6 to have

$$\mathbb{E}_{\boldsymbol{X},\boldsymbol{S},\boldsymbol{T}} \left[ \mathrm{tr} \left( \left( \boldsymbol{V}_b (\boldsymbol{U}_b^\top \boldsymbol{X} \boldsymbol{V}_b)^\dagger \boldsymbol{U}_b^\top \boldsymbol{X} \right)^\top \left( \boldsymbol{V}_b (\boldsymbol{U}_b^\top \boldsymbol{X} \boldsymbol{V}_b)^\dagger \boldsymbol{U}_b^\top \boldsymbol{X} \right) \right) \right]$$

$$= \mathbb{E}_{\boldsymbol{X},\boldsymbol{S},\boldsymbol{T}} \left[ \mathrm{tr} \left( \left( \boldsymbol{V}_b (\boldsymbol{U}_b'^\top \boldsymbol{X} \boldsymbol{V}_b)^\dagger \boldsymbol{U}_b'^\top \boldsymbol{X} \right)^\top \left( \boldsymbol{V}_b (\boldsymbol{U}_b'^\top \boldsymbol{X} \boldsymbol{V}_b)^\dagger \boldsymbol{U}_b'^\top \boldsymbol{X} \right) \right) \right] = \frac{|\boldsymbol{\nu}_b|(d - |\boldsymbol{\nu}_b|)}{|\boldsymbol{\mu}_b| - |\boldsymbol{\nu}_b| - 1} + |\boldsymbol{\nu}_b|.$$

For the other two cross terms, we treat them as only have $\boldsymbol{\nu}_b$ columns, and apply (28) in Lemma D.6 to have

$$\mathbb{E}_{\boldsymbol{X},\boldsymbol{S},\boldsymbol{T}} \left[ \mathrm{tr} \left( \left( \boldsymbol{V}_b (\boldsymbol{U}_b^\top \boldsymbol{X} \boldsymbol{V}_b)^\dagger \boldsymbol{U}_b^\top \boldsymbol{X} \right)^\top \left( \boldsymbol{V}_b (\boldsymbol{U}_b'^\top \boldsymbol{X} \boldsymbol{V}_b)^\dagger \boldsymbol{U}_b'^\top \boldsymbol{X} \right) \right) \right] = \frac{|\boldsymbol{\nu}_b| |\boldsymbol{\nu}_b^c|}{n - |\boldsymbol{\nu}_b| - 1} + |\boldsymbol{\nu}_b|.$$

Combined together, this is

$$\mathbb{E}_{\boldsymbol{X},\boldsymbol{S},\boldsymbol{T}}[B_{b,b}] = 2\left(\frac{|\boldsymbol{\nu}_b|(d-|\boldsymbol{\nu}_b|)}{|\boldsymbol{\mu}_b|-|\boldsymbol{\nu}_b|-1}+|\boldsymbol{\nu}_b|\right) - 2\left(\frac{|\boldsymbol{\nu}_b||\boldsymbol{\nu}_b^c|}{n-|\boldsymbol{\nu}_b|-1}+|\boldsymbol{\nu}_b|\right).$$

Taking the limit, this is

$$\lim_{n,d\to\infty}\frac{1}{d}\mathbb{E}_{\boldsymbol{X},\boldsymbol{S},\boldsymbol{T}}[B_{b,b}] = 2\frac{\gamma q(1-q)}{p-\gamma q} - 2\frac{\gamma q(1-q)}{1-\gamma q} = \frac{2\gamma q(1-q)(1-p)}{(p-\gamma q)(1-\gamma q)}.$$

For feature removal, the squared term becomes

$$B_{b,b} = \operatorname{tr}\left(\left(\boldsymbol{V}_b(\boldsymbol{U}_b^\top \boldsymbol{X}\boldsymbol{V}_b)^\dagger \boldsymbol{U}_b^\top \boldsymbol{X} - \boldsymbol{V}_b'(\boldsymbol{U}_b^\top \boldsymbol{X}\boldsymbol{V}_b')^\dagger \boldsymbol{U}_b^\top \boldsymbol{X}\right)^\top \left(\boldsymbol{V}_b(\boldsymbol{U}_b^\top \boldsymbol{X}\boldsymbol{V}_b)^\dagger \boldsymbol{U}_b^\top \boldsymbol{X} - \boldsymbol{V}_b'(\boldsymbol{U}_b^\top \boldsymbol{X}\boldsymbol{V}_b')^\dagger \boldsymbol{U}_b^\top \boldsymbol{X}\right)\right),$$

where $\boldsymbol{V}_b$ and $\boldsymbol{V}_b'$ are independent. We apply (29) in Lemma D.6 to have

$$\mathbb{E}_{\boldsymbol{X},\boldsymbol{S},\boldsymbol{T}}\left[\operatorname{tr}\left(\left(\boldsymbol{V}_b(\boldsymbol{U}_b^\top \boldsymbol{X}\boldsymbol{V}_b)^\dagger \boldsymbol{U}_b^\top \boldsymbol{X}\right)^\top \left(\boldsymbol{V}_b(\boldsymbol{U}_b^\top \boldsymbol{X}\boldsymbol{V}_b)^\dagger \boldsymbol{U}_b^\top \boldsymbol{X}\right)\right)\right]$$

$$=\mathbb{E}_{\boldsymbol{X},\boldsymbol{S},\boldsymbol{T}}\left[\operatorname{tr}\left(\left(\boldsymbol{V}_b'(\boldsymbol{U}_b^\top \boldsymbol{X}\boldsymbol{V}_b')^\dagger \boldsymbol{U}_b^\top \boldsymbol{X}\right)^\top \left(\boldsymbol{V}_b'(\boldsymbol{U}_b^\top \boldsymbol{X}\boldsymbol{V}_b')^\dagger \boldsymbol{U}_b^\top \boldsymbol{X}\right)\right)\right] = \frac{|\boldsymbol{\nu}_b|(d-|\boldsymbol{\nu}_b|)}{|\boldsymbol{\mu}_b|-|\boldsymbol{\nu}_b|-1}+|\boldsymbol{\nu}_b|.$$

For the other two cross terms, we treat them as only have $\boldsymbol{\nu}_b$ rows, and apply (28) in Lemma D.6 to have

$$\mathbb{E}_{\boldsymbol{X},\boldsymbol{S},\boldsymbol{T}}\left[\operatorname{tr}\left(\left(\boldsymbol{V}_b(\boldsymbol{U}_b^\top \boldsymbol{X}\boldsymbol{V}_b)^\dagger \boldsymbol{U}_b^\top \boldsymbol{X}\right)^\top \left(\boldsymbol{V}_b'(\boldsymbol{U}_b^\top \boldsymbol{X}\boldsymbol{V}_b')^\dagger \boldsymbol{U}_b^\top \boldsymbol{X}\right)\right)\right] = \frac{|\boldsymbol{\nu}_b\cap\boldsymbol{\nu}_b'||\boldsymbol{\nu}_b^c\cap\boldsymbol{\nu}_b'^c|}{np-|\boldsymbol{\nu}_b\cap\boldsymbol{\nu}_b'|-1}+|\boldsymbol{\nu}_b\cap\boldsymbol{\nu}_b'|.$$

Combined together, this is

$$\mathbb{E}_{\boldsymbol{X},\boldsymbol{S},\boldsymbol{T}}[B_{b,b}] = 2\left(\frac{|\boldsymbol{\nu}_b|(d-|\boldsymbol{\nu}_b|)}{|\boldsymbol{\mu}_b|-|\boldsymbol{\nu}_b|-1}+|\boldsymbol{\nu}_b|\right) - 2\left(\frac{|\boldsymbol{\nu}_b\cap\boldsymbol{\nu}_b'||\boldsymbol{\nu}_b^c\cap\boldsymbol{\nu}_b'^c|}{np-|\boldsymbol{\nu}_b\cap\boldsymbol{\nu}_b'|-1}+|\boldsymbol{\nu}_b\cap\boldsymbol{\nu}_b'|\right).$$

Taking the limit, this is

$$\lim_{n,d\to\infty}\frac{1}{d}\mathbb{E}_{\boldsymbol{X},\boldsymbol{S},\boldsymbol{T}}[B_{b,b}] = 2\frac{\gamma q(1-q)}{p-\gamma q} - 2\frac{\gamma q^2(1-q)^2}{p-\gamma q^2} + 2q(1-q) = \frac{2\gamma pq(1-q)^2}{(p-\gamma q)(p-\gamma q^2)} + \frac{2pq(1-q)}{p-\gamma q^2}.$$

For the cross terms ($b_1 \neq b_2$) for instance removal, we have

$$\lim_{n,d\to\infty}\frac{n}{d}\cdot\mathbb{E}_{\boldsymbol{X},\boldsymbol{S},\boldsymbol{T}}[B_{b_1,b_2}]$$

$$=\lim_{n,d\to\infty}\frac{n}{d}\cdot\mathbb{E}_{\boldsymbol{X},\boldsymbol{S},\boldsymbol{T}}\left[\frac{|\boldsymbol{\nu}_{b_1}\cap\boldsymbol{\nu}_{b_2}||\boldsymbol{\nu}_{b_1}^c\cap\boldsymbol{\nu}_{b_2}^c|}{\operatorname{tr}(\Lambda^\vee)-|\boldsymbol{\nu}_{b_1}\cap\boldsymbol{\nu}_{b_2}|-1}+|\boldsymbol{\nu}_{b_1}\cap\boldsymbol{\nu}_{b_2}|-\frac{|\boldsymbol{\nu}_{b_1}\cap\boldsymbol{\nu}_{b_2}||\boldsymbol{\nu}_{b_1}^c\cap\boldsymbol{\nu}_{b_2}^c|}{\operatorname{tr}(\Lambda^\vee)-|\boldsymbol{\nu}_{b_1}\cap\boldsymbol{\nu}_{b_2}|-1}-|\boldsymbol{\nu}_{b_1}\cap\boldsymbol{\nu}_{b_2}|\right]$$

$$=\lim_{n,d\to\infty}\frac{n}{d}\cdot\left[\frac{\frac{s^2}{d}\frac{(d-s)^2}{d}}{n-1-\frac{s^2}{d}-1}+\frac{s^2}{d}-\frac{\frac{s^2}{d}\frac{(d-s)^2}{d}}{n-\frac{s^2}{d}-1}-\frac{s^2}{d}\right] = \frac{\gamma q^2(1-q)^2}{(1-\gamma q^2)^2}.$$

As for the cross terms ($b_1 \neq b_2$) for feature removal, we have

$$\lim_{n,d\to\infty}\frac{n}{d}\cdot\mathbb{E}_{\boldsymbol{X},\boldsymbol{S},\boldsymbol{T}}[B_{b_1,b_2}]$$

$$=\lim_{n,d\to\infty}\frac{n}{d}\cdot\mathbb{E}_{\boldsymbol{X},\boldsymbol{S},\boldsymbol{T}}\left[\frac{|\boldsymbol{\nu}_{b_1}'\cap\boldsymbol{\nu}_{b_2}'||\boldsymbol{\nu}_{b_1}'^c\cap\boldsymbol{\nu}_{b_2}'^c|}{\operatorname{tr}(\Lambda^\vee)-|\boldsymbol{\nu}_{b_1}'\cap\boldsymbol{\nu}_{b_2}'|-1}+|\boldsymbol{\nu}_{b_1}'\cap\boldsymbol{\nu}_{b_2}'|-\frac{|\boldsymbol{\nu}_{b_1}\cap\boldsymbol{\nu}_{b_2}||\boldsymbol{\nu}_{b_1}^c\cap\boldsymbol{\nu}_{b_2}^c|}{\operatorname{tr}(\Lambda^\vee)-|\boldsymbol{\nu}_{b_1}\cap\boldsymbol{\nu}_{b_2}|-1}-|\boldsymbol{\nu}_{b_1}\cap\boldsymbol{\nu}_{b_2}|\right]$$

$$=\lim_{n,d\to\infty}\frac{n}{d}\cdot\left[\frac{\frac{s^2}{d-1}(\frac{(d-1-s)^2}{d-2}+1)}{n-\frac{s^2}{d-1}-1}+\frac{s^2}{d-1}-\frac{\frac{s^2}{d}\frac{(d-s)^2}{d-1}}{n-\frac{s^2}{d}-1}-\frac{s^2}{d}\right]$$

$$=\frac{\gamma q^4(1-q^2)-2q^3(1-q)}{(1-\gamma q^2)^2} + \frac{q^2}{1-\gamma q^2} + \gamma^{-1}q^2.$$

**Overparameterized case** For instance removal, the squared terms become

$$B_{b,b} = \operatorname{tr}\left(\left(V_b(U_b^\top X V_b)^\dagger U_b^\top X - V_b(U_b'^\top X V_b)^\dagger U_b'^\top X\right)^\top \left(V_b(U_b^\top X V_b)^\dagger U_b^\top X - V_b(U_b'^\top X V_b)^\dagger U_b'^\top X\right)\right),$$

where $U_b$ and $U_b'$ are independent. We apply (29) in Lemma D.6 to have

$$\mathbb{E}_{X,S,T}\left[\operatorname{tr}\left(\left(V_b(U_b^\top X V_b)^\dagger U_b^\top X\right)^\top \left(V_b(U_b^\top X V_b)^\dagger U_b^\top X\right)\right)\right]$$
$$=\mathbb{E}_{X,S,T}\left[\operatorname{tr}\left(\left(V_b(U_b'^\top X V_b)^\dagger U_b'^\top X\right)^\top \left(V_b(U_b'^\top X V_b)^\dagger U_b'^\top X\right)\right)\right] = \frac{|\mu_b|(d-|\nu_b|)}{|\nu_b| - |\mu_b| - 1} + |\mu_b|.$$

For the other two cross terms, we treat them as only have $\nu_b$ columns, and apply (28) in Lemma D.6 to have

$$\mathbb{E}_{X,S,T}\left[\operatorname{tr}\left(\left(V_b(U_b^\top X V_b)^\dagger U_b^\top X\right)^\top \left(V_b(U_b'^\top X V_b)^\dagger U_b'^\top X\right)\right)\right]$$
$$=\frac{|\mu_b/\mu_b'||\mu_b'/\mu_b|}{qd - |\mu_b \cap \mu_b'| - 1} + |\mu_b \cap \mu_b'|\frac{d - |\mu_b \cap \mu_b'| - 1}{qd - |\mu_b \cap \mu_b'| - 1}.$$

Combined together, this is

$$\mathbb{E}_{X,S,T}[B_{b,b}] = 2\left(\frac{|\mu_b|(d-|\nu_b|)}{|\nu_b| - |\mu_b| - 1} + |\mu_b|\right) - 2\left(\frac{|\mu_b/\mu_b'||\mu_b'/\mu_b|}{qd - |\mu_b \cap \mu_b'| - 1} + |\mu_b \cap \mu_b'|\frac{d - |\mu_b \cap \mu_b'| - 1}{qd - |\mu_b \cap \mu_b'| - 1}\right).$$

Taking the limit, this is

$$\lim_{n,d\to\infty}\frac{1}{d}\mathbb{E}_{X,S,T}[B_{b,b}] = 2\frac{p(1-q)}{\gamma q - p} + 2\gamma^{-1}p - 2\frac{\gamma^{-1}p^2(1-p)^2}{\gamma q - p^2} - 2\gamma^{-1}p^2\frac{\gamma - p^2}{\gamma q - p^2}.$$

For feature removal, the squared term becomes

$$B_{b,b} = \operatorname{tr}\left(\left(V_b(U_b^\top X V_b)^\dagger U_b^\top X - V_b'(U_b^\top X V_b')^\dagger U_b^\top X\right)^\top \left(V_b(U_b^\top X V_b)^\dagger U_b^\top X - V_b'(U_b^\top X V_b')^\dagger U_b^\top X\right)\right),$$

where $V_b$ and $V_b'$ are independent. We apply (29) in Lemma D.6 to have

$$\mathbb{E}_{X,S,T}\left[\operatorname{tr}\left(\left(V_b(U_b^\top X V_b)^\dagger U_b^\top X\right)^\top \left(V_b(U_b^\top X V_b)^\dagger U_b^\top X\right)\right)\right]$$
$$=\mathbb{E}_{X,S,T}\left[\operatorname{tr}\left(\left(V_b'(U_b^\top X V_b')^\dagger U_b^\top X\right)^\top \left(V_b'(U_b^\top X V_b')^\dagger U_b^\top X\right)\right)\right] = \frac{|\mu_b|(d-|\nu_b|)}{|\nu_b| - |\mu_b| - 1} + |\mu_b|.$$

For the other two cross terms, we treat them as only have $\mu_b$ rows, and apply (28) in Lemma D.6 to have

$$\mathbb{E}_{X,S,T}\left[\operatorname{tr}\left(\left(V_b(U_b^\top X V_b)^\dagger U_b^\top X\right)^\top \left(V_b'(U_b^\top X V_b')^\dagger U_b^\top X\right)\right)\right] = |\mu_b|.$$

Combined together, this is

$$\mathbb{E}_{X,S,T}[B_{b,b}] = 2\left(\frac{|\mu_b|(d-|\nu_b|)}{|\nu_b| - |\mu_b| - 1} + |\mu_b|\right) - 2|\mu_b|.$$

Taking the limit, this is

$$\lim_{n,d\to\infty}\frac{1}{d}\mathbb{E}_{X,S,T}[B_{b,b}] = \frac{2p(1-q)}{\gamma q - p}.$$

As for the cross terms ($b_1 \neq b_2$) for instance removal, we have

$$\lim_{n,d \to \infty} \frac{n}{d} \cdot \mathbb{E}_{\boldsymbol{X},\boldsymbol{S},\boldsymbol{T}} \left[ B_{b_1,b_2} \right]$$

$$= \lim_{n,d \to \infty} \frac{n}{d} \cdot \mathbb{E}_{\boldsymbol{X},\boldsymbol{S},\boldsymbol{T}} \left[ \frac{|\boldsymbol{\mu}_1'/\boldsymbol{\mu}_2'||\boldsymbol{\mu}_2'/\boldsymbol{\mu}_1'|}{\mathrm{tr}(\Theta^\vee) - |\boldsymbol{\mu}_1' \cap \boldsymbol{\mu}_2'|} + |\boldsymbol{\mu}_1' \cap \boldsymbol{\mu}_2'| \frac{d - |\boldsymbol{\mu}_1' \cap \boldsymbol{\mu}_2'| - 1}{\mathrm{tr}(\Theta^\vee) - |\boldsymbol{\mu}_1' \cap \boldsymbol{\mu}_2'| - 1} \right]$$

$$- \lim_{n,d \to \infty} \frac{n}{d} \cdot \mathbb{E}_{\boldsymbol{X},\boldsymbol{S},\boldsymbol{T}} \left[ \frac{|\boldsymbol{\mu}_1/\boldsymbol{\mu}_2||\boldsymbol{\mu}_2/\boldsymbol{\mu}_1|}{\mathrm{tr}(\Theta^\vee) - |\boldsymbol{\mu}_1 \cap \boldsymbol{\mu}_2|} + |\boldsymbol{\mu}_1 \cap \boldsymbol{\mu}_2| \frac{d - |\boldsymbol{\mu}_1 \cap \boldsymbol{\mu}_2| - 1}{\mathrm{tr}(\Theta^\vee) - |\boldsymbol{\mu}_1 \cap \boldsymbol{\mu}_2| - 1} \right]$$

$$= \lim_{n,d \to \infty} \frac{n}{d} \cdot \left[ \frac{\frac{m^2}{n-1} \frac{(n-1-m)^2}{n-2}}{d - \frac{m^2}{n-1}} + \frac{m^2}{n-1} - \frac{\frac{m^2}{n} \frac{(n-m)^2}{n-1}}{d - \frac{m^2}{n}} - \frac{m^2}{n} \right]$$

$$= \frac{\gamma^{-1} p^4 (1-p^2) - 2p^3(1-p)}{(\gamma - p^2)^2} + \gamma^{-1} p^2.$$

As for the cross terms ($b_1 \neq b_2$) for feature removal, we have

$$\lim_{n,d \to \infty} \frac{n}{d} \cdot \mathbb{E}_{\boldsymbol{X},\boldsymbol{S},\boldsymbol{T}} \left[ B_{b_1,b_2} \right]$$

$$= \lim_{n,d \to \infty} \frac{n}{d} \cdot \mathbb{E}_{\boldsymbol{X},\boldsymbol{S},\boldsymbol{T}} \left[ \frac{|\boldsymbol{\mu}_1/\boldsymbol{\mu}_2||\boldsymbol{\mu}_2/\boldsymbol{\mu}_1|}{d - 1 - |\boldsymbol{\mu}_1 \cap \boldsymbol{\mu}_2|} + |\boldsymbol{\mu}_1 \cap \boldsymbol{\mu}_2| \frac{d - |\boldsymbol{\mu}_1 \cap \boldsymbol{\mu}_2| - 1}{d - 1 - |\boldsymbol{\mu}_1 \cap \boldsymbol{\mu}_2| - 1} \right]$$

$$- \lim_{n,d \to \infty} \frac{n}{d} \cdot \mathbb{E}_{\boldsymbol{X},\boldsymbol{S},\boldsymbol{T}} \left[ \frac{|\boldsymbol{\mu}_1/\boldsymbol{\mu}_2||\boldsymbol{\mu}_2/\boldsymbol{\mu}_1|}{d - |\boldsymbol{\mu}_1 \cap \boldsymbol{\mu}_2|} + |\boldsymbol{\mu}_1 \cap \boldsymbol{\mu}_2| \frac{d - |\boldsymbol{\mu}_1 \cap \boldsymbol{\mu}_2| - 1}{d - |\boldsymbol{\mu}_1 \cap \boldsymbol{\mu}_2| - 1} \right]$$

$$= \frac{\gamma^{-1} p^2 (1-p)^2}{(\gamma - p^2)^2} + \frac{\gamma^{-1} p^2}{\gamma - p^2}.$$

$\square$

# E. Contents Related to Model Free Stability Guarantees

This appendix proves stronger stability guarantees in which the expectation is taken only over the algorithmic randomness.

### E.1. Stability Analysis of Random Forest

We now turn to decision-tree models (Breiman et al., 2017) and their ensemble variant, random forests (Breiman, 2001). Figure 5 provides a schematic overview. A decision tree's partition can be encoded by two vectors: one that records the feature chosen at each internal node and another that stores the corresponding split threshold; see the `feature` and `threshold` attributes in Pedregosa et al. (2011). Both vectors have length $2^T - 1$, where $T$ is the `max_depth` parameter (Pedregosa et al., 2011), i.e., the maximum number of splits from the root node to any leaf node.

We consider a simplified model of dyadic trees, where each splitting takes place only at the median along the chosen feature. We further consider the max-edge version of the dyadic tree. The corresponding local-path procedure is summarized in Algorithm 2.

Rather than analysing the entire tree, we study a local property: the leaf node that contains a given query point $\boldsymbol{x}$. This node is uniquely determined by the sequence of features used to split the path from the root to that leaf; see Figure 5. To encode the node, we define the feature weight matrix

$$\boldsymbol{w}_t = (\boldsymbol{w}_t^1, \cdots, \boldsymbol{w}_t^d) \in \mathbb{R}^{T \times d}.$$

If, up to the $t$-th step, the leaf node that contains $\boldsymbol{x}$ has been split along the $j$-th feature for $t_j$ times, then the first $t_j$ entries of $\boldsymbol{w}_t^j$ are one, while the rest of the entries are zero. We write

$$N_t^k := \|\boldsymbol{w}_t^k\|_1, \qquad N_t^{-j,k} := \|(\boldsymbol{w}_t^{-j})^k\|_1,$$

for the corresponding split counts in the full and feature-removed runs.

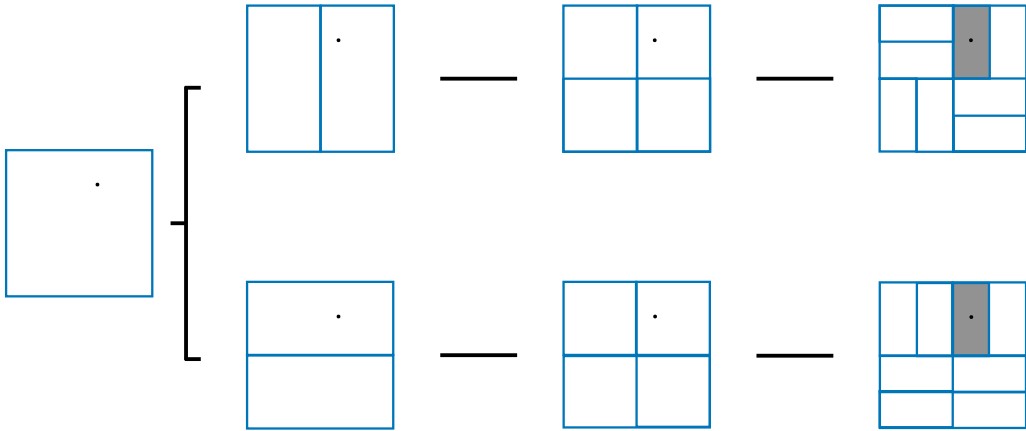

*Figure 5.* Two possibilities of dyadic tree partition. The black dot represents the target test sample $\boldsymbol{x}$. The two partitions, though different in other areas, have the same representation $\boldsymbol{w}$ that corresponds to the grey area.

At each local split $t$, draw a candidate feature set

$$\boldsymbol{\nu}_t \subset [d], \qquad |\boldsymbol{\nu}_t| = s, \qquad q = s/d,$$

uniformly without replacement. Define the candidate-set max-edge set

$$\mathcal{M}(\boldsymbol{w}_t, \boldsymbol{\nu}_t) := \operatorname*{argmin}_{k \in \boldsymbol{\nu}_t} N_t^k.$$

The tree then chooses

$$k_t \in \operatorname*{argmax}_{k \in \mathcal{M}(\boldsymbol{w}_t, \boldsymbol{\nu}_t)} \Delta_k(\boldsymbol{w}_t; \mathcal{D}),$$

with deterministic tie-breaking. For the feature-removed run, use the coupled candidate set

$$\boldsymbol{\nu}_t^{-j} := \boldsymbol{\nu}_t \setminus \{j\}.$$

If $\boldsymbol{\nu}_t^{-j} = \varnothing$, the removed update returns the current encoding. Otherwise define

$$\mathcal{M}^{-j}(\boldsymbol{w}_t^{-j}, \boldsymbol{\nu}_t^{-j}) := \operatorname*{argmin}_{k \in \boldsymbol{\nu}_t^{-j}} N_t^{-j,k},$$

and choose the feature in this set with the largest split decrease, with the same deterministic tie-breaking rule. See Figure 5 for an illustration. In this case, only one vector with length $2^T - 1$ is enough to store the whole tree, as the thresholds are uniquely defined. We argue that dyadic trees often serve as a simplified model of decision-tree partitions and are widely used in nonparametric statistical estimation (e.g. Blanchard et al., 2007; Perchet & Rigollet, 2013; Cai et al., 2023; Ma et al., 2023; Cai et al., 2024; Ma & Yang, 2024).

For instance, the $\boldsymbol{x}$ in Figure 5 has

$$\boldsymbol{w}_1 = \begin{pmatrix} 0 & 1 \\ 0 & 0 \\ 0 & 0 \end{pmatrix} \text{ or } \begin{pmatrix} 1 & 0 \\ 0 & 0 \\ 0 & 0 \end{pmatrix}, \qquad \boldsymbol{w}_2 = \begin{pmatrix} 1 & 1 \\ 0 & 0 \\ 0 & 0 \end{pmatrix}, \qquad \boldsymbol{w}_3 = \begin{pmatrix} 1 & 1 \\ 0 & 1 \\ 0 & 0 \end{pmatrix}.$$

Moreover, given a fixed $\boldsymbol{w}_t$, the grey region in Figure 5 is the unique leaf node compatible with that encoding. Denote this region as $L_{\boldsymbol{w}_t}$.

We now justify the choice of the encoding $\boldsymbol{w}$. First, the encoding scheme is closely related to `feature_importances_` (Pedregosa et al., 2011), which measures a feature's global importance by the proportion of splits that use that feature. Analogously, for a specific test point $\boldsymbol{x}$, we can quantify the local importance of feature $j$ by

$$\frac{\|\boldsymbol{w}_T^j\|_1}{\sum_{j'=1}^d \|\boldsymbol{w}_T^{j'}\|_1}.$$

This point-specific view is also aligned with recent work on individual variable importance, which moves beyond population-level summaries (Dai et al., 2025). Moreover, we demonstrate a direct relationship between the prediction at $\boldsymbol{x}$ and the matrix $\boldsymbol{w}$ through the following proposition. We consider $(\boldsymbol{X}, \boldsymbol{y})$ generated from an additive model, where

$$\mathbb{E}_{\boldsymbol{y}|\boldsymbol{x}}[y] = f^*(\boldsymbol{x}) = \sum_{j=1}^{d} f_j^*(\boldsymbol{x}^j), \qquad \boldsymbol{x} \sim \mathrm{Unif}([0,1]^d). \tag{34}$$

**Proposition E.1.** *Let $L_{\boldsymbol{w}}$ be the rectangle of the node associated to $\boldsymbol{x}$ under $\boldsymbol{w}$. Let $f^{\boldsymbol{w}}$ be the decision tree predictor associated to $\boldsymbol{w}$. Here, we slightly abuse the notation since $\boldsymbol{w}$ only contains prediction information about $\boldsymbol{x}$. Since we only care about $\boldsymbol{x}$, we view all decision trees with the same $\boldsymbol{w}$ as equivalent. Based on the above definitions, there exists a map $\mathcal{K}$ such that*

$$\mathcal{K}(\boldsymbol{w}) = \mathbb{E}_{\boldsymbol{y}|\boldsymbol{X}, \boldsymbol{w}}\left[f^{\boldsymbol{w}}(\boldsymbol{x})\right].$$

*Moreover, the map $\mathcal{K}$ is affine: there exist a constant $c_{\boldsymbol{x}}$ and a matrix $\Theta_{\boldsymbol{x}} \in \mathbb{R}^{T \times d}$ such that*

$$\mathcal{K}(\boldsymbol{w}) = c_{\boldsymbol{x}} + \langle \Theta_{\boldsymbol{x}}, \boldsymbol{w} \rangle.$$

*Consequently,*

$$\mathcal{K}(\boldsymbol{w}_1) - \mathcal{K}(\boldsymbol{w}_2) = \langle \Theta_{\boldsymbol{x}}, \boldsymbol{w}_1 - \boldsymbol{w}_2 \rangle \lesssim \|\boldsymbol{w}_1 - \boldsymbol{w}_2\|_F.$$

Given the proposition, the random forest predictor is fully determined by the ensemble feature-weight matrix $\mathbb{E}_{\xi}[\boldsymbol{w}]$, namely

$$\begin{aligned}
\mathbb{E}_{\boldsymbol{y}|\boldsymbol{X}}\left[f(\boldsymbol{x})\right] &= \mathbb{E}_{\xi}\left[\mathbb{E}_{\boldsymbol{y}|\boldsymbol{X}, \boldsymbol{w}}\left[f^{\boldsymbol{w}}(\boldsymbol{x})\right]\right] \\
&= \mathbb{E}_{\xi}\left[\mathcal{K}(\boldsymbol{w})\right] = \mathcal{K}\left(\mathbb{E}_{\xi}[\boldsymbol{w}]\right),
\end{aligned}$$

where the last equality follows from the affine representation of $\mathcal{K}$. Moreover, for the feature-removed forest, the affine representation gives

$$\begin{aligned}
\left(\mathbb{E}_{\boldsymbol{y}|\boldsymbol{X}}\left[f(\boldsymbol{x}) - f^{-j}(\boldsymbol{x})\right]\right)^2 \\
= \left(\mathcal{K}(\mathbb{E}_{\xi}[\boldsymbol{w}_T]) - \mathcal{K}(\mathbb{E}_{\xi}[\boldsymbol{w}_T^{-j}])\right)^2 \\
= \left\langle \Theta_{\boldsymbol{x}}, \mathbb{E}_{\xi}[\boldsymbol{w}_T] - \mathbb{E}_{\xi}[\boldsymbol{w}_T^{-j}] \right\rangle^2 \\
\lesssim \left\| \mathbb{E}_{\xi}[\boldsymbol{w}_T] - \mathbb{E}_{\xi}[\boldsymbol{w}_T^{-j}] \right\|_F^2.
\end{aligned}$$

Thus, it suffices to analyze $\mathbb{E}_{\xi}[\boldsymbol{w}_T]$ with respect to the Frobenius norm.

We next verify the two recursive-object conditions needed in Proposition 4.5: the one-step radius condition and the one-step contraction condition.

**Theorem E.2** (Recursive-object conditions for max-edge dyadic forests). *Let $\|\cdot\|$ be the Frobenius norm. Let $\mathcal{A}$ be the max-edge dyadic forest algorithm defined above, and assume $d \geq 2$. Then the one-step candidate set satisfies*

$$\mathrm{rad}(\mathcal{W}_t) \leq 1, \qquad t \in [T].$$

*Moreover, for every removed feature $j \in [d]$, the contraction condition in Proposition 4.5 holds with*

$$\delta_t = \sqrt{2}, \qquad t \in [T].$$

*Consequently, Proposition 4.5 gives*

$$\frac{1}{d} \sum_{j=1}^{d} \left\| \mathbb{E}_{\xi}[\boldsymbol{w}_T] - \mathbb{E}_{\xi}[\boldsymbol{w}_T^{-j}] \right\|_F^2 \leq \left( \sum_{t=1}^{T} \prod_{t'=t+1}^{T} (1 + \sqrt{2}) \right)^2 \frac{q}{(d-1)(1-q)}, \tag{35}$$

*where $q$ is the feature-subsampling parameter in Proposition 4.5. Therefore, by the affine representation of $\mathcal{K}$,*

$$\frac{1}{d} \sum_{j=1}^{d} \left(\mathbb{E}_{\boldsymbol{y}|\boldsymbol{X}}\left[f(\boldsymbol{x}) - f^{-j}(\boldsymbol{x})\right]\right)^2 \lesssim \left( \sum_{t=1}^{T} \prod_{t'=t+1}^{T} (1 + \sqrt{2}) \right)^2 \frac{q}{(d-1)(1-q)}. \tag{36}$$

The key point is that the proof below does not require any regularity of the empirical split decrease. The split score may choose any feature inside the max-edge set, as long as the algorithm uses deterministic tie-breaking. The max-edge restriction determines the candidate set, while the contraction argument only uses the fact that one transition changes at most one entry of the local feature-weight matrix.

### E.2. Additional Experiments

We adopt the MARSadd setting from Friedman (1991), now a standard benchmark for random forests (Mentch & Zhou, 2020; Curth et al., 2024). We generate $\boldsymbol{X}$ from $\text{Unif}([0,1]^d)$ and set

$$y = 0.1e^{4X^1} + \frac{4}{1 + e^{-20(X^2 - 0.5)}} + 3X^3 + 2X^4 + X^5 + \mathcal{N}(0, \sigma^2). \tag{37}$$

We choose $n = 250$, $d = 10$, and $\sigma = 1$. We set $B = 256$, $q \in \{0.1, 0.2, 0.3, 0.4, 0.5\}$, and $T \in [7]$. In Figure 7 and 8, we conduct a parallel set of experiments to those for RF, which brings similar conclusions.

We adopt the following simulation setup. An orthogonal design matrix $\boldsymbol{X} \in \mathbb{R}^{n \times d}$ is generated by applying QR decomposition to a matrix with i.i.d. standard normal entries. The true coefficient vector is set to $\boldsymbol{\beta}^* = d^{-1/2}(1, \dots, 1) \in \mathbb{R}^d$, and the noise terms are drawn independently as $\varepsilon_i \sim \mathcal{N}(0, \sigma^2)$. We fix the parameters as follows: $n = 250$, $d = 200$, and $\sigma = 1$. The number of bagging rounds is set to $B = 256$. We vary the feature subsampling ratio $q \in \{0.05, 0.1, 0.3, 0.5\}$ and the number of forward selection steps $T \in [20]$. Larger values of $T$ are not considered, as none of the chosen $q$ values satisfy the regime $qT \lesssim 1$ beyond this range.

We next explore the connection between feature instability and generalization. Define $\boldsymbol{\beta}^*_{\text{strong}} = (5^{-1/2}, \dots, 5^{-1/2}, 0, \dots, 0) \in \mathbb{R}^{200}$ to be a sparse coefficient with 5 nonzero positions, which represents a strong signal. Let $\boldsymbol{\beta}^*_{\text{weak}} = d^{-1/2}(1, \dots, 1)$ and $\boldsymbol{\beta}^*_{\text{mixed}} = (\boldsymbol{\beta}^*_{\text{weak}} + \boldsymbol{\beta}^*_{\text{strong}}) / \|\boldsymbol{\beta}^*_{\text{weak}} + \boldsymbol{\beta}^*_{\text{strong}}\|_2$ represent a weak and a mixed signal, respectively. Note that this sense of strong and weak depicts a different aspect of learning hardness opposed to the SNR (Zhou & Mentch, 2023; Liu & Mazumder, 2025). We compare the relationship between feature instability, generalization gap, generalization error, and empirical risk in Figure 6, respectively, in four rows.

- Although there is no theoretical result establishing a relationship between the generalization gap and feature instability, we observe a similarity in their trends, regardless of whether the signal is concentrated or spread across the features.

- The generalization-risk curves, however, have different patterns under different signal types. Observing the second last row of Figure 6, when the signal is weak, it is suggested to use a smaller subsampling ratio. In contrast, when the signal is strong, using a larger $q$ and a smaller $T$ is a better choice. This observation aligns with past experience that the bagging provides a regularization effect, which is more promising when the signal is weak (LeJeune et al., 2020; Mentch & Zhou, 2020).

- In practice, the generalization error is unobservable. Practitioners see only the empirical risk in the last row. In all signal types, the empirical risk decreases as $T$ grows, i.e., the model capacity becomes larger. Thus, in practice, hyperparameter tuning involves balancing between the empirical risk and the generalization gap, where the latter can take evidence from feature instability.

### E.3. Finite Bagging

We have been focusing on infinite bagging, which is impossible in practice. In this section, we show that the obtained theoretical results is also representative for a finite bagging round $B$, given that $B$ is moderately large.

**Proposition E.3** (Feature instability of finite bagging)**.** *Let Assumption 4.3 hold. Then for bagged algorithm $\mathcal{A}$ with $B$ bagging rounds and its outputs $\boldsymbol{w}^{(b)} \in \mathcal{W}$, $b \in [B]$, there holds*

$$\frac{1}{d}\sum_{j=1}^{d}\left\|\frac{1}{B}\sum_{b=1}^{B}\boldsymbol{w}^{(b)} - \frac{1}{B}\sum_{b=1}^{B}\boldsymbol{w}^{(b),-j}\right\|^2 \leq \frac{\sqrt{3}}{d}\sum_{j=1}^{d}\left\|\mathbb{E}_{\xi}\left[\boldsymbol{w}\right] - \mathbb{E}_{\xi}\left[\boldsymbol{w}^{-j}\right]\right\|^2 + \frac{6\sqrt{3}\,\text{rad}\left(\mathcal{W}\right)^2\log\left(\frac{d+1}{\delta}\right)}{B}$$

*with probability $1 - \delta$.*

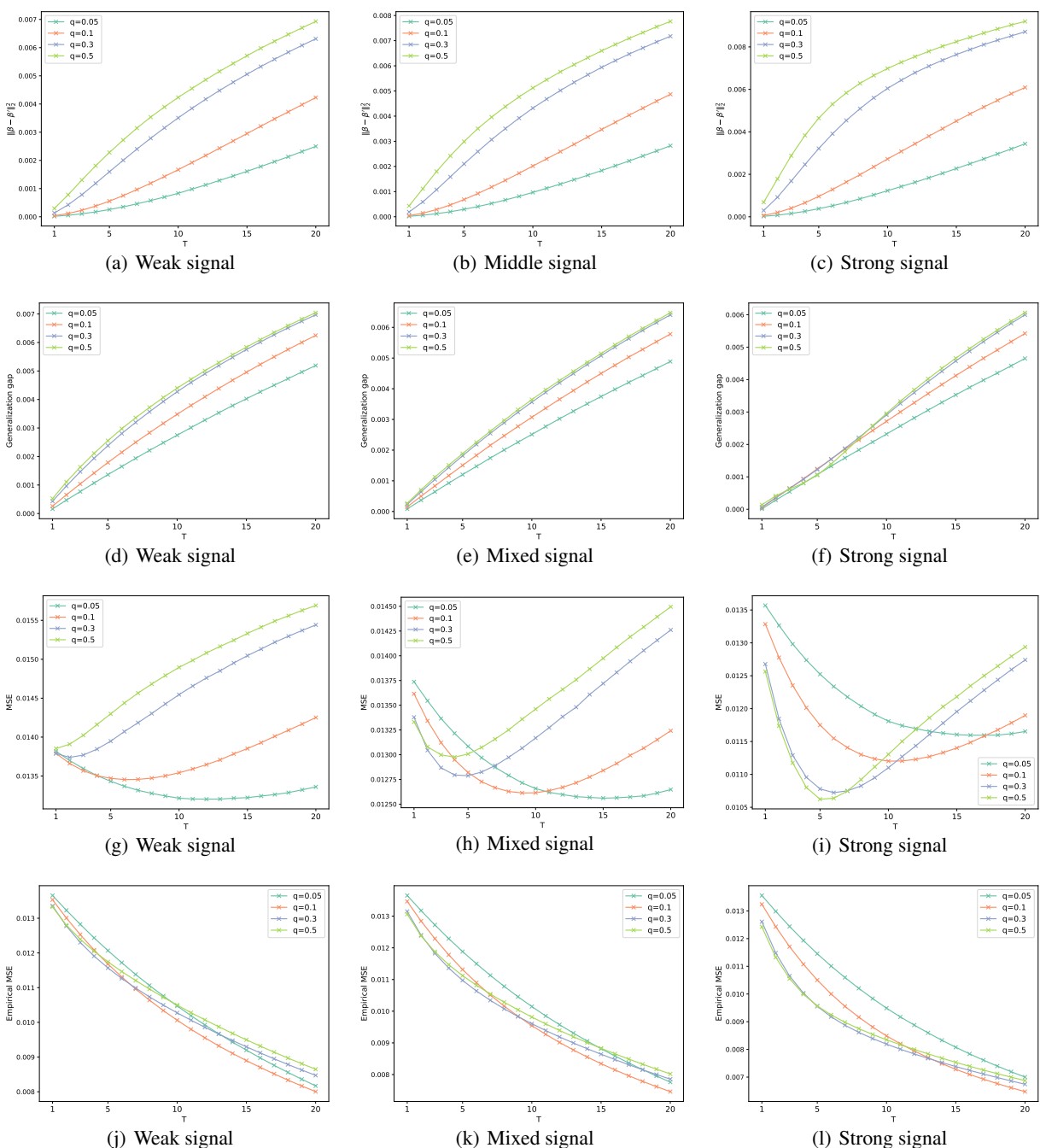

*Figure 6.* The feature instability, generalization gap, generalization error, and empirical risk of RFS under different signal types. Each cross corresponds to 100 repetitions. We set $\sigma = 0.1$.

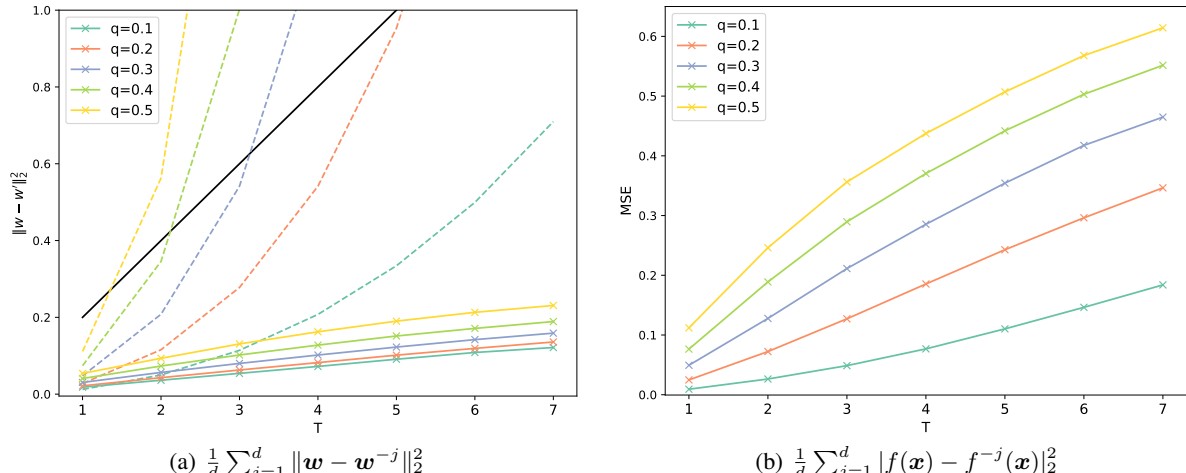

(a) $\frac{1}{d}\sum_{j=1}^{d}\|\boldsymbol{w}-\boldsymbol{w}^{-j}\|_2^2$      (b) $\frac{1}{d}\sum_{j=1}^{d}|f(\boldsymbol{x})-f^{-j}(\boldsymbol{x})|_2^2$

*Figure 7.* Feature-instability metrics for RF. Left: comparison of the feature importance between the feature-bagged RF and the non-bagged RF. The real lines with crosses correspond to the simulated values over 100 repetitions under different feature subsampling ratios. The dashed lines are the theoretical upper bound (35). The black solid line is the feature instability of non-bagged RF, i.e., a single tree. Right: comparison of the prediction value between the feature-bagged RF with different subsampling ratios. We compute over an independent test set of size $n$.

The proof of Proposition E.3 can be found in Appendix E.11, which is a direct application of Hoeffding's inequality and the union bound. The finite-bagging feature instability exceeds the infinite-bagging feature instability by a term that decays linearly with the number of bagging rounds $B$. On the constant, the radius of the overall algorithm $\mathcal{A}$ is no larger than the sum of superiors of $\mathrm{rad}(\mathcal{W}_t)$. Combining this finite-$B$ concentration term with Proposition 4.5, it suffices to set

$$B \gtrsim d \cdot \frac{1-q}{q}. \tag{38}$$

In Figure 9, empirical feature stability improves as $B$ increases. For larger $q$, fewer bagging rounds are needed to reach the theoretical stability regime predicted by (38).

As observed by Soloff et al. (2024a), the finite bagging is also harmless given $B \gtrsim n \cdot (1-p)/p$. Though the results are dual, we notice that in practice, achieving (38) is much more reasonable, as the dimension $d$ is usually much smaller than $n$. For instance, a general magnitude for tabular data is $(n,d) \sim (1e^5, 1e^2)$, which poses a requirement of $B \gtrsim 1e^2$ for feature stability, as opposed to $B \gtrsim 1e^5$ for instance stability.

### E.4. Algorithms of Random Forward Selection and Random Forest

The algorithm of RFS procedure proposed by Mentch & Zhou (2020) is detailed in Algorithm 1. At each step $t$ of the $b$-th bagging round, starting from the current selection vector $\boldsymbol{w}_{t-1}^{(b)}$, the algorithm first subsamples a fraction $q$ of the $d$ available features according to the randomness $\xi_t^{(b)}$, yielding a candidate set $\boldsymbol{\nu}_t^{(b)}$. Next, the algorithm selects the feature that yields the greatest reduction in residual sum of squares:

$$j_t^{(b)} = \arg \min_{j \in \boldsymbol{\nu}_t^{(b)}} \min_{\boldsymbol{\beta}} \left\| \boldsymbol{y} - \boldsymbol{X} \boldsymbol{E}_{\boldsymbol{w}_{t-1}^{(b)} + \boldsymbol{e}_j} \boldsymbol{\beta} \right\|_2^2,$$

where $\boldsymbol{e}_j$ denotes the $j$-th standard basis vector in $\mathbb{R}^d$ and $\boldsymbol{E}\boldsymbol{w}$ is the diagonal matrix that projects onto the coordinates selected by $\boldsymbol{w}$. The selected variable is then added to the active set, yielding the update:

$$\mathcal{A}(\boldsymbol{w}_{t-1}^{(b)}) = \boldsymbol{w}_t^{(b)} = \boldsymbol{w}_{t-1}^{(b)} + \boldsymbol{e}_{j_t^{(b)}}.$$

If all the indices in the candidate set $\boldsymbol{\nu}_t^{(b)}$ have already been selected (i.e., $\boldsymbol{\nu}_t^{(b)} \subseteq j : (\boldsymbol{w}_{t-1}^{(b)})j = 1$), then we define $j_t^{(b)} = \texttt{None}$ and set $\boldsymbol{e}_{\texttt{None}} = \boldsymbol{0}$ so that the selection vector remains unchanged.

---

**Algorithm 1** Randomized Forward Selection

---

**for** $b \in [B]$ **do**

    Draw bootstrap sample $\mathcal{D}^{(b)} = \{(\boldsymbol{x}_i^{(b)}, y_i^{(b)})\}_{i=1}^n$ from original data $\mathcal{D}$.

    Initialize empty active set $\boldsymbol{w}_0 = \boldsymbol{0}$.

    **for** $t \in [T]$ **do**

        Select subset of $s$ features uniformly at random, denoted $\boldsymbol{\nu}_t^{(b)}$.

        Select $j_t = \operatorname{argmin}_{j \in \boldsymbol{\nu}_t^{(b)}} \|\boldsymbol{y}^{(b)} - \boldsymbol{X}^{(b)} E_{\boldsymbol{w}_{t-1}+e_j} \left(\boldsymbol{X}^{(b)} E_{\boldsymbol{w}_{t-1}+e_j}\right)^\dagger \boldsymbol{y}^{(b)}\|_2^2$.

        Update $\boldsymbol{w}_t = \boldsymbol{w}_{t-1} + e_{j_t}$.

        Update coefficient estimates $\hat{\boldsymbol{\beta}}^{(b)}$ as

$$E_{\boldsymbol{w}_t}\hat{\boldsymbol{\beta}}^{(b)} = \operatorname*{argmin}_{\beta} \|\boldsymbol{y}^{(b)} - \boldsymbol{X}^{(b)} E_{\boldsymbol{w}_t}\beta\|^2, \quad (\boldsymbol{I} - E_{\boldsymbol{w}_t})\hat{\boldsymbol{\beta}}_{A_k^c}^{(b)} = \boldsymbol{0}.$$

    **end for**

**end for**

Compute final coefficient estimates $\hat{\boldsymbol{\beta}} = \frac{1}{B}\sum_{b=1}^B \hat{\boldsymbol{\beta}}^{(b)}$.

Compute predictions $\hat{\boldsymbol{y}} = \boldsymbol{X}\hat{\boldsymbol{\beta}}$.

---

---

**Algorithm 2** Max-edge dyadic randomized forest, local path at $\boldsymbol{x}$

---

Let $\boldsymbol{E}_{r,k} \in \mathbb{R}^{T \times d}$ be the matrix with one at entry $(r, k)$ and zero elsewhere.

**for** $b \in [B]$ **do**

    Draw bootstrap sample $\mathcal{D}^{(b)} = \{(\boldsymbol{x}_i^{(b)}, y_i^{(b)})\}_{i=1}^n$ from original data $\mathcal{D}$.

    Initialize local encoding $\boldsymbol{w}_0 = \boldsymbol{0}^{T \times d}$ and local cell $L_0 = [0, 1]^d$.

    **for** $t \in [T]$ **do**

        Select subset of $s$ features uniformly at random, denoted $\boldsymbol{\nu}_t^{(b)}$.

        Define $N_{t-1}^k = \|\boldsymbol{w}_{t-1}^k\|_1$ and $\mathcal{M}(\boldsymbol{w}_{t-1}, \boldsymbol{\nu}_t^{(b)}) = \operatorname{argmin}_{k \in \boldsymbol{\nu}_t^{(b)}} N_{t-1}^k$.

        Select $j_t \in \operatorname{argmax}_{k \in \mathcal{M}(\boldsymbol{w}_{t-1}, \boldsymbol{\nu}_t^{(b)})} \Delta_k(\boldsymbol{w}_{t-1}; \mathcal{D}^{(b)})$, with deterministic tie-breaking.

        Update $\boldsymbol{w}_t = \boldsymbol{w}_{t-1} + \boldsymbol{E}_{N_{t-1}^{j_t}+1, j_t}$.

        Split $L_{t-1}$ along feature $j_t$ at the midpoint and set $L_t$ to the child cell containing $\boldsymbol{x}$.

    **end for**

    Compute the prediction at $\boldsymbol{x}$ as the average of labels in $L_T$.

**end for**

Average the $B$ predictions.

---

## E.5. Proofs of Results in Section 4.1

***Proof of Proposition 4.1.*** The proposition is a special case of Proposition 4.4, where $\operatorname{rad}(\mathcal{W}) = \operatorname{rad}([-M, M]) = M$. □

***Proof of Proposition 4.4.*** The proof is a direct extension of that of Proposition 4.1, except that here we are considering the norm in general spaces. We use $\boldsymbol{\nu}$ to denote a subsampled set of features. Let event $E_{j,\boldsymbol{\nu}} = \{j \notin \boldsymbol{\nu}\}$. Denote the quantity $L_j = \widehat{\boldsymbol{w}} - \widehat{\boldsymbol{w}}^{-j}$, for which we similarly define $\widehat{\boldsymbol{w}}^{-j} = \mathbb{E}_{\boldsymbol{\nu}}\left[\mathcal{A}(\mathcal{D}^{-j}; \boldsymbol{\nu})\right]$. Recall that we have $\widehat{\boldsymbol{w}} = \mathbb{E}_{\boldsymbol{\nu}}\left[\mathcal{A}(\mathcal{D}; \boldsymbol{\nu})\right]$. We also denote $\boldsymbol{w}^{\boldsymbol{\nu}} = \mathcal{A}(\mathcal{D}; \boldsymbol{\nu})$. As a result, we can express $L_j$ as

$$
\begin{aligned}
L_j &= \mathbb{E}_{\boldsymbol{\nu}}\left[\mathcal{A}(\mathcal{D}; \xi) - \mathcal{A}(\mathcal{D}^{-j}; \xi)\right] \\
&= \mathbb{E}_{\boldsymbol{\nu}}\left[\mathbb{E}_{\boldsymbol{\nu}}\left[\boldsymbol{w}^{\boldsymbol{\nu}}\right] - \boldsymbol{w}^{\boldsymbol{\nu}} \mid E_{j,\boldsymbol{\nu}}\right] \\
&= \frac{1}{\mathbb{P}(E_{j,\boldsymbol{\nu}})}\mathbb{E}_{\boldsymbol{\nu}}\left[(\mathbb{E}_{\boldsymbol{\nu}}\left[\boldsymbol{w}^{\boldsymbol{\nu}}\right] - \boldsymbol{w}^{\boldsymbol{\nu}})\mathbf{1}\{E_{j,\boldsymbol{\nu}}\}\right].
\end{aligned}
$$

Note that we took uniform sampling without replacement. Thus, $\mathbb{P}(E_{j,\boldsymbol{\nu}}) = 1 - q$. Since $\mathbb{E}_{\boldsymbol{\nu}}\left[\mathbb{E}_{\boldsymbol{\nu}}\left[\boldsymbol{w}^{\boldsymbol{\nu}}\right] - \boldsymbol{w}^{\boldsymbol{\nu}}\right] = 0$, we have

$$
\begin{aligned}
L_j =& \frac{1}{\mathbb{P}(E_{j,\boldsymbol{\nu}})} \mathbb{E}_{\boldsymbol{\nu}}\left[\left(\mathbb{E}_{\boldsymbol{\nu}}\left[\boldsymbol{w}^{\boldsymbol{\nu}}\right] - \boldsymbol{w}^{\boldsymbol{\nu}}\right) \mathbf{1}\{E_{j,\boldsymbol{\nu}}\}\right] \\
=& \frac{1}{1-q} \mathbb{E}_{\boldsymbol{\nu}}\left[\left(\mathbb{E}_{\boldsymbol{\nu}}\left[\boldsymbol{w}^{\boldsymbol{\nu}}\right] - \boldsymbol{w}^{\boldsymbol{\nu}}\right)\left(\mathbf{1}\{E_{j,\boldsymbol{\nu}}\} - (1-q)\right)\right].
\end{aligned}
$$

Then, the sum of norms has

$$
\begin{aligned}
\sum_{j=1}^{d} \|\widehat{\boldsymbol{w}} - \widehat{\boldsymbol{w}}^{-j}\|_{\mathcal{H}}^2 &= \sum_{j=1}^{d} \|L_j\|_{\mathcal{H}}^2 \\
&= \sum_{j=1}^{d} \left\langle L_j, \frac{1}{1-q}\mathbb{E}_{\boldsymbol{\nu}}\left[\left(\mathbb{E}_{\boldsymbol{\nu}}\left[\boldsymbol{w}^{\boldsymbol{\nu}}\right] - \boldsymbol{w}^{\boldsymbol{\nu}}\right)\left(\mathbf{1}\{E_{j,\boldsymbol{\nu}}\} - (1-q)\right)\right]\right\rangle_{\mathcal{H}} \\
&\overset{(i)}{=} \sum_{j=1}^{d} \frac{1}{1-q}\mathbb{E}_{\boldsymbol{\nu}}\left[\left\langle L_j, \left(\mathbb{E}_{\boldsymbol{\nu}}\left[\boldsymbol{w}^{\boldsymbol{\nu}}\right] - \boldsymbol{w}^{\boldsymbol{\nu}}\right)\left(\mathbf{1}\{E_{j,\boldsymbol{\nu}}\} - (1-q)\right)\right\rangle_{\mathcal{H}}\right] \\
&= \sum_{j=1}^{d} \frac{1}{1-q}\mathbb{E}_{\boldsymbol{\nu}}\left[\left\langle L_j\left(\mathbf{1}\{E_{j,\boldsymbol{\nu}}\} - (1-q)\right), \mathbb{E}_{\boldsymbol{\nu}}\left[\boldsymbol{w}^{\boldsymbol{\nu}}\right] - \boldsymbol{w}^{\boldsymbol{\nu}}\right\rangle_{\mathcal{H}}\right] \\
&= \frac{1}{1-q}\mathbb{E}_{\boldsymbol{\nu}}\left[\left\langle \sum_{j=1}^{d} L_j\left(\mathbf{1}\{E_{j,\boldsymbol{\nu}}\} - (1-q)\right), \mathbb{E}_{\boldsymbol{\nu}}\left[\boldsymbol{w}^{\boldsymbol{\nu}}\right] - \boldsymbol{w}^{\boldsymbol{\nu}}\right\rangle_{\mathcal{H}}\right]
\end{aligned}
$$

Here, $(i)$ holds since $L_j$ is a constant and thus is independent of $\boldsymbol{\nu}$. Applying the Cauchy-Schwarz inequality, we have

$$
\begin{aligned}
\sum_{j=1}^{d} \|\widehat{\boldsymbol{w}} - \widehat{\boldsymbol{w}}^{-j}\|_{\mathcal{H}}^2 \leq& \frac{1}{1-q}\sqrt{\mathbb{E}_{\boldsymbol{\nu}}\left[\left\|\sum_{j=1}^{d} L_j\left(\mathbf{1}\{E_{j,\boldsymbol{\nu}}\} - (1-q)\right)\right\|_{\mathcal{H}}^2\right] \cdot \mathbb{E}_{\boldsymbol{\nu}}\left[\left\|\mathbb{E}_{\boldsymbol{\nu}}\left[\boldsymbol{w}^{\boldsymbol{\nu}}\right] - \boldsymbol{w}^{\boldsymbol{\nu}}\right\|_{\mathcal{H}}^2\right]} \\
\leq& \frac{\mathrm{rad}(\mathcal{W})}{1-q}\sqrt{\mathbb{E}_{\boldsymbol{\nu}}\left[\left\|\sum_{j=1}^{d} L_j\left(\mathbf{1}\{E_{j,\boldsymbol{\nu}}\} - (1-q)\right)\right\|_{\mathcal{H}}^2\right]}.
\end{aligned}
\tag{39}
$$

Here, the last step is true because

$$
\mathbb{E}_{\boldsymbol{\nu}}\left[\left\|\mathbb{E}_{\boldsymbol{\nu}}\left[\boldsymbol{w}^{\boldsymbol{\nu}}\right] - \boldsymbol{w}^{\boldsymbol{\nu}}\right\|_{\mathcal{H}}^2\right] = \inf_{\boldsymbol{w}\in\mathcal{W}}\mathbb{E}_{\boldsymbol{\nu}}\left[\|\boldsymbol{w} - \boldsymbol{w}^{\boldsymbol{\nu}}\|_{\mathcal{H}}^2\right] \leq \inf_{\boldsymbol{w}\in\mathcal{W}}\sup_{\boldsymbol{w}'\in\mathcal{W}}\left[\|\boldsymbol{w} - \boldsymbol{w}'\|_{\mathcal{H}}^2\right] \leq \mathrm{rad}^2(\mathcal{W}),
$$

where the first equality is due to the fact that expectation is the minimizer of squared loss. We calculate the remaining term by noticing

$$
\mathbb{E}_{\boldsymbol{\nu}}\left[\mathbf{1}\{E_{j_1,\boldsymbol{\nu}}^c\}\mathbf{1}\{E_{j_2,\boldsymbol{\nu}}^c\}\right] = \begin{cases} \frac{s(s-1)}{d(d-1)} & \text{if } j_1 \neq j_2, \\ \frac{s}{d} & \text{if } j_1 = j_2 \end{cases}
$$

and thus

$$
\mathrm{Cov}\left[\mathbf{1}\{E_{j_1,\boldsymbol{\nu}}^c\}, \mathbf{1}\{E_{j_2,\boldsymbol{\nu}}^c\}\right] = \mathbb{E}_{\boldsymbol{\nu}}\left[\mathbf{1}\{E_{j_1,\boldsymbol{\nu}}^c\}\mathbf{1}\{E_{j_2,\boldsymbol{\nu}}^c\}\right] - q^2 = \begin{cases} \frac{s(s-1)}{d(d-1)} - \frac{s^2}{d^2} & \text{if } j_1 \neq j_2, \\ \frac{s}{d} - \frac{s^2}{d^2} & \text{if } j_1 = j_2. \end{cases}
$$

Then we expand (39) by

$$\mathbb{E}_{\boldsymbol{\nu}}\left[\left\|\sum_{j=1}^{d} L_j \left(\mathbf{1}\{E_{j,\boldsymbol{\nu}}\} - (1-q)\right)\right\|_{\mathcal{H}}^2\right] = \sum_{j_1, j_2}^{d} \langle L_{j_1}, L_{j_2} \rangle \operatorname{Cov}\left[\mathbf{1}\{E_{j_1,\boldsymbol{\nu}}^c\}, \mathbf{1}\{E_{j_2,\boldsymbol{\nu}}^c\}\right]$$

$$= \sum_{j_1 \neq j_2} \langle L_{j_1}, L_{j_2} \rangle \left(\frac{s(s-1)}{d(d-1)} - \frac{s^2}{d^2}\right) + \sum_{j=1}^{d} \langle L_j, L_j \rangle \left(\frac{s}{d} - \frac{s^2}{d^2}\right)$$

$$= \left\|\sum_{j=1}^{d} L_j\right\|_{\mathcal{H}}^2 \left(\frac{s(s-1)}{d(d-1)} - \frac{s^2}{d^2}\right) + \sum_{j=1}^{d} \langle L_j, L_j \rangle \left(\frac{s}{d} - \frac{s(s-1)}{d(d-1)}\right)$$

$$\leq \frac{d}{d-1} q(1-q) \sum_{j=1}^{d} \|L_j\|_{\mathcal{H}}^2.$$

Bringing back this into (39), we have

$$\sum_{j=1}^{d} \|L_j\|_{\mathcal{H}}^2 \leq \frac{\operatorname{rad}(\mathcal{W})}{1-q} \sqrt{\frac{d}{d-1} q(1-q) \sum_{j=1}^{d} \|L_j\|_{\mathcal{H}}^2},$$

which leads to

$$\frac{1}{d} \sum_{j=1}^{d} \|L_j\|_{\mathcal{H}}^2 \leq \frac{\operatorname{rad}^2(\mathcal{W})}{d-1} \frac{q}{1-q}.$$

$\square$

## E.6. Related Contexts and Proofs of Results in Section 4.2

*Proof of Proposition 4.5.* The proof is done by recursively applying equation (44) and Proposition 4.4. $\square$

Our final goal is to bound the feature instability measure:

$$\frac{1}{d} \sum_{j=1}^{d} \left\|\mathbb{E}_{\xi}\left[\boldsymbol{w}_T\right] - \mathbb{E}_{\xi}\left[\boldsymbol{w}_T^{-j}\right]\right\|_{\mathcal{H}}^2,$$

where $T$ is some preset horizon length. We first consider a fixed $j \in [d]$ for the convenience of derivation. Let $\Delta \boldsymbol{w}_t^{-j} := \boldsymbol{w}_T - \boldsymbol{w}_T^{-j}$ and write

$$\left\|\mathbb{E}_{\xi}\left[\Delta \boldsymbol{w}_t^{-j}\right]\right\|_{\mathcal{H}}^2 = \left\|\mathbb{E}_{\xi}\left[S_{\xi_t} \circ \cdots \circ S_{\xi_1}(\boldsymbol{w}_0)\right] - \mathbb{E}_{\xi}\left[S_{\xi_t}^{-j} \circ \cdots \circ S_{\xi_1}^{-j}(\boldsymbol{w}_0)\right]\right\|_{\mathcal{H}}^2. \tag{40}$$

In the above quantity, the discrepancy between $\boldsymbol{w}_t$ and $\boldsymbol{w}_t^{-j}$ accumulates as $t$ grows as each step introduces additional discrepancy due to the difference between $S_{\xi_t}$ and $S_{\xi_t}^{-j}$.

The recursive nature of the procedure indicates that we should analyze the instability using a peeling strategy. Specifically, at the $t$-th recursive step, we decompose the instability into two terms. The first term, referred to as the peel at step $t$, deals with the heterogeneity caused by the removal in the $t$-th step, formally defined as

$$\operatorname{peel}_t^j = \mathbb{E}_{\xi}\left[S_{\xi_t} \circ S_{\xi_{t-1}} \circ \cdots \circ S_{\xi_1}(\boldsymbol{w}_0)\right] - \mathbb{E}_{\xi}\left[S_{\xi_t}^{-j} \circ S_{\xi_{t-1}} \circ \cdots \circ S_{\xi_1}(\boldsymbol{w}_0)\right]. \tag{41}$$

The second term is the error accumulated along $1, \ldots, t-1$, referred to as the pith at step $t$, is defined as

$$\operatorname{pith}_t^j = \mathbb{E}_{\xi}\left[S_{\xi_t}^{-j} \circ S_{\xi_{t-1}} \circ \cdots \circ S_{\xi_1}(\boldsymbol{w}_0)\right] - \mathbb{E}_{\xi}\left[S_{\xi_t}^{-j} \circ S_{\xi_{t-1}}^{-j} \circ \cdots \circ S_{\xi_1}^{-j}(\boldsymbol{w}_0)\right]. \tag{42}$$

Compared to $\Delta \boldsymbol{w}_{t-1}^{-j}$, the pith at step $t$ can be expressed as

$$\mathrm{pith}_t^j := \mathbb{E}_\xi \left[ S_{\xi_t}^{-j} \circ \Delta \boldsymbol{w}_{t-1}^{-j} \right],$$

which applies the random operator $S_{\xi_t}^{-j}$ to the change $\Delta \boldsymbol{w}_{t-1}^{-j}$ from the previous step. We will then show the one-step condition (16):

$$\left\| \mathrm{pith}_t^j \right\|_{\mathcal{H}} \le (1 + \delta_t) \left\| \mathbb{E}_\xi \left[ \Delta \boldsymbol{w}_{t-1}^{-j} \right] \right\|_{\mathcal{H}}, \tag{43}$$

where the inflation factor $1 + \delta_t$ depends on the algorithm $\mathcal{A}$ and the geometry of the space $\mathcal{H}$.

Applying the triangle inequality, we bound the recursive quantity from (40) as

$$
\begin{aligned}
\frac{1}{d} \sum_{j=1}^d \left\| \mathbb{E}_\xi \left[ \Delta \boldsymbol{w}_t^{-j} \right] \right\|_{\mathcal{H}}^2 &= \frac{1}{d} \sum_{j=1}^d \left\| \mathrm{peel}_t^j + \mathrm{pith}_t^j \right\|_{\mathcal{H}}^2 \\
&\le \frac{1}{d} \sum_{j=1}^d \left( \left\| \mathrm{peel}_t^j \right\|_{\mathcal{H}} + \left\| \mathrm{pith}_t^j \right\|_{\mathcal{H}} \right)^2 \\
&\le \frac{1}{d} \sum_{j=1}^d \left( \left\| \mathrm{peel}_t^j \right\|_{\mathcal{H}} + (1 + \delta_t) \left\| \mathbb{E}_\xi \left[ \Delta \boldsymbol{w}_{t-1}^{-j} \right] \right\|_{\mathcal{H}} \right)^2 .
\end{aligned}
$$

Then, by Minkowski's inequality, we obtain the recurrence bound:

$$\sqrt{\frac{1}{d} \sum_{j=1}^d \left\| \mathbb{E}_\xi \left[ \Delta \boldsymbol{w}_t^{-j} \right] \right\|_{\mathcal{H}}^2} \le \underbrace{\sqrt{\frac{1}{d} \sum_{j=1}^d \left\| \mathrm{peel}_j \right\|_{\mathcal{H}}^2}}_{\text{Proposition 4.4}} + (1 + \delta_t) \underbrace{\sqrt{\frac{1}{d} \sum_{j=1}^d \left\| \mathbb{E}_\xi \left[ \Delta \boldsymbol{w}_{t-1}^{-j} \right] \right\|_{\mathcal{H}}^2}}_{\text{step } t-1}. \tag{44}$$

The first term on the right-hand side corresponds to the instability due to one-step fitting, which can be bounded via Proposition 4.4. The inequality (44) forms an inhomogeneous linear recurrence, which admits a closed-form solution. This is formalized in the following proposition.

We conclude with a brief remark on the peeling strategy. For instance stability, a closely related analysis exists for stochastic gradient descent (SGD) (Hardt et al., 2016; Lei & Ying, 2020), which was noted earlier in this section as a key motivating example. That analysis relies on assumptions analogous to Assumption 4.3 and the inflation condition (16). As a result, the corresponding instability bounds for SGD also exhibit cumulative error growth with the number of update steps.

### E.7. Derivation of Feature Instability for Non-bagged RFS and RF

We derive the feature instability without bagging.

We first consider RFS. Notice that without bagging, every $\boldsymbol{w}_T$ includes $T$ features without doubt. Thus, $\mathbb{E}_\xi [\boldsymbol{w}_T]$ has $T$ positions being one and others being zero. Moreover, the selected features are the $T$ features with the largest $\boldsymbol{y}^\top \boldsymbol{X}^j \boldsymbol{X}^{j\top} \boldsymbol{y}$, as suggested by (45). Thus, as long as $j$ is in the group of $T$ features with largest $\boldsymbol{y}^\top \boldsymbol{X}^j \boldsymbol{X}^{j\top} \boldsymbol{y}$, the term $\mathbb{E}_\xi [\boldsymbol{w}_T] - \mathbb{E}_\xi \left[ \boldsymbol{w}_T^{-j} \right]$ has one position being 1 and the other being -1. Thus, $\left\| \mathbb{E}_\xi [\boldsymbol{w}_T] - \mathbb{E}_\xi \left[ \boldsymbol{w}_T^{-j} \right] \right\|^2 = 2$. There are in total $T$ out of $d$ such terms, and thus

$$\frac{1}{d} \sum_{j=1}^d \left\| \mathbb{E}_\xi [\boldsymbol{w}_T] - \mathbb{E}_\xi \left[ \boldsymbol{w}_T^{-j} \right] \right\|^2 = \frac{2T}{d}.$$

For random forest, things are the same. $\mathbb{E}_\xi [\boldsymbol{w}_T]$ has in total $T$ positions being one and others being zero. Removing $j$-th feature results in a $\mathbb{E}_\xi [\boldsymbol{w}_T]$ with the positions in $j$-th column being zero, and equally many entries in other columns being

one. Thus, $\left\|\mathbb{E}_\xi\left[\boldsymbol{w}_T\right] - \mathbb{E}_\xi\left[\boldsymbol{w}_T^{-j}\right]\right\|^2 = 2 \cdot \sum_{i=1}^T (\mathbb{E}_\xi\left[\boldsymbol{w}_T\right])_i^j$, which leads to

$$\frac{1}{d}\sum_{j=1}^d \left\|\mathbb{E}_\xi\left[\boldsymbol{w}_T\right] - \mathbb{E}_\xi\left[\boldsymbol{w}_T^{-j}\right]\right\|^2 = \frac{1}{d}\sum_{j=1}^d 2 \cdot \sum_{i=1}^T (\mathbb{E}_\xi\left[\boldsymbol{w}_T\right])_i^j = \frac{2T}{d}.$$

## E.8. Derivation of the Affine Random Forest Encoding Map

***Proof of Proposition E.1.*** Fix the query point $\boldsymbol{x}$. Let $I_{k,r}(\boldsymbol{x})$ be the dyadic interval in coordinate $k$ that contains $x^k$ after $r$ midpoint splits along coordinate $k$. Define

$$A_{k,r}(\boldsymbol{x}) := 2^r \int_{I_{k,r}(\boldsymbol{x})} f_k^*(u)\,du, \qquad r = 0, \ldots, T.$$

If $r_k(\boldsymbol{w}) = \|\boldsymbol{w}^k\|_1$, then under the additive model and the uniform design,

$$\mathbb{E}_{\boldsymbol{y}|\boldsymbol{X},\boldsymbol{w}}[f^{\boldsymbol{w}}(\boldsymbol{x})] = \sum_{k=1}^d A_{k,r_k(\boldsymbol{w})}(\boldsymbol{x}).$$

Because the encoding is column-filled,

$$A_{k,r_k(\boldsymbol{w})} = A_{k,0} + \sum_{r=1}^T w_{r,k}\{A_{k,r} - A_{k,r-1}\}.$$

Therefore

$$\mathcal{K}(\boldsymbol{w}) = \sum_{k=1}^d A_{k,0} + \sum_{k=1}^d\sum_{r=1}^T w_{r,k}\{A_{k,r} - A_{k,r-1}\}.$$

Thus $\mathcal{K}$ is affine in $\boldsymbol{w}$. In particular,

$$\mathcal{K}(\boldsymbol{w}_1) - \mathcal{K}(\boldsymbol{w}_2) = \sum_{k=1}^d\sum_{r=1}^T (w_{1,r,k} - w_{2,r,k})\{A_{k,r} - A_{k,r-1}\},$$

which is linear in $\boldsymbol{w}_1 - \boldsymbol{w}_2$. Since this is a fixed finite-dimensional linear functional, it is bounded by a constant times $\|\boldsymbol{w}_1 - \boldsymbol{w}_2\|_F$. $\qquad\square$

## E.9. Proof of the Random Forward Selection Instability Bound

***Proof of Theorem 4.6.*** (i) We first bound the radius of $\mathcal{W}_t$. Note that the space of $\mathcal{W}_t$ depends on the input $\boldsymbol{w}_{t-1}$. Recall that $\boldsymbol{w}_{t-1} \in [0,1]^d$, and suppose $\boldsymbol{w}_{t-1}$ has $\tilde{t} - 1$ positive positions, where the collection of positive positions is denoted as $W$. Then there is a natural choice of $\tilde{\boldsymbol{w}}$ where

$$\tilde{\boldsymbol{w}}^j = \frac{1}{d - \tilde{t} + 1}\mathbf{1}(j \notin W) + \mathbf{1}(j \in W).$$

For any $\boldsymbol{w}_t$, there holds

$$\|\tilde{\boldsymbol{w}} - \boldsymbol{w}_t\|_{\mathcal{H}}^2 = \sum_{j \notin W} \frac{1}{(d - \tilde{t} + 1)^2} - \frac{2}{d - \tilde{t} + 1} + 1 = 1 - \frac{1}{d - \tilde{t} + 1} \leq 1.$$

Thus,

$$\inf_{\boldsymbol{w} \in \mathcal{W}_t} \sup_{\boldsymbol{w}' \in \mathcal{W}_t} \|\boldsymbol{w} - \boldsymbol{w}'\|_{\mathcal{H}} \leq \sup_{\boldsymbol{w}' \in \mathcal{W}_t} \|\tilde{\boldsymbol{w}} - \boldsymbol{w}'\|_{\mathcal{H}} \leq 1$$

holds uniformly for all $t$ and $\xi$.

(ii) We next prove the contraction bound. Throughout this part, without loss of generality, we assume that the feature scores are ordered as

$$\boldsymbol{y}^\top \boldsymbol{X}^{j_1} \boldsymbol{X}^{j_1\top} \boldsymbol{y} \geq \boldsymbol{y}^\top \boldsymbol{X}^{j_2} \boldsymbol{X}^{j_2\top} \boldsymbol{y} \quad \text{if } j_1 \geq j_2.$$

Under the orthogonal-design assumption $\boldsymbol{X}^\top \boldsymbol{X} = n \boldsymbol{I}_d$, let

$$r_{t-1} = \boldsymbol{y} - \boldsymbol{X} \boldsymbol{E}_{\boldsymbol{w}_{t-1}} \hat{\boldsymbol{\beta}}_{\boldsymbol{w}_{t-1}}$$

be the residual after fitting the currently active set. For every unselected feature $j \notin \operatorname{supp}(\boldsymbol{w}_{t-1})$, adding feature $j$ decreases the residual sum of squares by

$$\mathrm{RSS}(\boldsymbol{w}_{t-1}) - \mathrm{RSS}(\boldsymbol{w}_{t-1} + \boldsymbol{e}_j) = \frac{1}{n} \left( \boldsymbol{X}^{j\top} r_{t-1} \right)^2 = \frac{1}{n} \boldsymbol{y}^\top \boldsymbol{X}^j \boldsymbol{X}^{j\top} \boldsymbol{y}. \tag{45}$$

Therefore, conditional on the candidate set, RFS selects

$$j_t \in \operatorname*{argmax}_{j \in \boldsymbol{\nu}_t \setminus \operatorname{supp}(\boldsymbol{w}_{t-1})} \boldsymbol{y}^\top \boldsymbol{X}^j \boldsymbol{X}^{j\top} \boldsymbol{y},$$

with deterministic tie-breaking.

We first record the only structural fact about the pathwise difference that is used below.

*Auxiliary fact.* For every fixed realization of $\xi_{1:t}$ and every $j \in [d]$, define

$$D_t^j := S_{\xi_t} \circ \cdots \circ S_{\xi_1}(\boldsymbol{w}_0) - S_{\xi_t}^{-j} \circ \cdots \circ S_{\xi_1}^{-j}(\boldsymbol{w}_0).$$

Then

$$D_t^j \in \{\boldsymbol{0}, \boldsymbol{e}_j\} \cup \{\boldsymbol{e}_j - \boldsymbol{e}_a : a \in [d] \setminus \{j\}\}.$$

Equivalently, $D_t^j$ is nonnegative on the $j$-th coordinate, nonpositive on all other coordinates, and has at most one negative coordinate.

To prove the auxiliary fact, use induction on $t$. The statement is trivial at $t = 0$. Suppose it holds at step $t - 1$. If $D_{t-1}^j = \boldsymbol{0}$, then the two paths have the same active set except that the reduced procedure cannot select feature $j$. Therefore, either both paths select the same feature, or the full path selects $j$ while the reduced path selects the best available alternative. The new difference is thus either $\boldsymbol{0}$, $\boldsymbol{e}_j$, or $\boldsymbol{e}_j - \boldsymbol{e}_a$. If $D_{t-1}^j = \boldsymbol{e}_j$, then the two paths have the same active set outside coordinate $j$, and the reduced update $S_{\xi_t}^{-j}$ never operates along $j$; hence the non-$j$ selection is the same and the difference remains $\boldsymbol{e}_j$. Finally, if $D_{t-1}^j = \boldsymbol{e}_j - \boldsymbol{e}_a$, then the full path contains $j$ whereas the reduced path contains $a$, with all other active coordinates identical. At the next step, the only possible negative discrepancy is either still attached to $a$, transferred to a lower-ranked feature $b < a$, or removed if the full path selects $a$. No second negative coordinate can be created, because both paths select at most one new feature at the current step. This proves the auxiliary fact.

Recall that the pith term in the proof of Proposition 4.5 is

$$
\begin{aligned}
(@) :=& \mathbb{E}_\xi \left[ S_{\xi_t}^{-j} \circ S_{\xi_{t-1}} \circ \cdots \circ S_{\xi_1}(\boldsymbol{w}_0) \right] \\
& - \mathbb{E}_\xi \left[ S_{\xi_t}^{-j} \circ S_{\xi_{t-1}}^{-j} \circ \cdots \circ S_{\xi_1}^{-j}(\boldsymbol{w}_0) \right].
\end{aligned}
$$

We need to show

$$\|(@)\|_2 \leq (1 + \delta_t) \left\| \mathbb{E}_\xi \left[ S_{\xi_{t-1}} \circ \cdots \circ S_{\xi_1}(\boldsymbol{w}_0) - S_{\xi_{t-1}}^{-j} \circ \cdots \circ S_{\xi_1}^{-j}(\boldsymbol{w}_0) \right] \right\|_2. \tag{46}$$

Define the pathwise previous-step difference

$$D_{t-1}^j := S_{\xi_{t-1}} \circ \cdots \circ S_{\xi_1}(\boldsymbol{w}_0) - S_{\xi_{t-1}}^{-j} \circ \cdots \circ S_{\xi_1}^{-j}(\boldsymbol{w}_0),$$

and define the pathwise pith difference

$$D_t^{j,\mathrm{pith}} := S_{\xi_t}^{-j} \circ S_{\xi_{t-1}} \circ \cdots \circ S_{\xi_1}(\boldsymbol{w}_0) - S_{\xi_t}^{-j} \circ S_{\xi_{t-1}}^{-j} \circ \cdots \circ S_{\xi_1}^{-j}(\boldsymbol{w}_0).$$

Then
$$(@) = \mathbb{E}_\xi \left[ D_t^{j,\text{pith}} \right].$$

By the auxiliary fact, on the event $D_{t-1}^j \neq \mathbf{0}$, the difference is either $\boldsymbol{e}_j$ or $\boldsymbol{e}_j - \boldsymbol{e}_a$ for a unique $a \neq j$. Let
$$\mathcal{I}_j = [d] \setminus \{j\}.$$

For $a \in \mathcal{I}_j$, define
$$u_a := \mathbb{P}_{\xi_{1:(t-1)}}(D_{t-1}^j = \boldsymbol{e}_j - \boldsymbol{e}_a),$$
and define
$$\rho := \mathbb{P}_{\xi_{1:(t-1)}}(D_{t-1}^j = \boldsymbol{e}_j), \qquad \alpha := \rho + \sum_{a \in \mathcal{I}_j} u_a.$$

Let $u = (u_a)_{a \in \mathcal{I}_j}$, viewed as a row vector indexed by $\mathcal{I}_j$, and embedded in $\mathbb{R}^d$ by putting zero on the $j$-th coordinate. Then
$$\mathbb{E}_\xi \left[ D_{t-1}^j \right] = \alpha \boldsymbol{e}_j - u. \tag{47}$$

If $\mathbb{E}_\xi[D_{t-1}^j] = \mathbf{0}$, then $\alpha = 0$ and $u = \mathbf{0}$, and the desired contraction is trivial. Hence we assume below that $\mathbb{E}_\xi[D_{t-1}^j] \neq \mathbf{0}$.

For $a, b \in \mathcal{I}_j$, define the conditional transition kernel
$$M_{ab}^{(t,j)} := \mathbb{P} \left( D_t^{j,\text{pith}} = \boldsymbol{e}_j - \boldsymbol{e}_b \mid D_{t-1}^j = \boldsymbol{e}_j - \boldsymbol{e}_a \right), \tag{48}$$

with $M_{ab}^{(t,j)} = 0$ if the conditioning event has probability zero. Write $M = M^{(t,j)}$ for short.

This is a conditional kernel. It is not the unconditional matrix $\mathbb{E}[\mathbf{1}(E_{a,b})]$. Therefore, the following representation follows from the law of total probability and does not require any independence between the previous discrepancy event and the current transition event:
$$\mathbb{E}_\xi \left[ D_t^{j,\text{pith}} \right] = \alpha \boldsymbol{e}_j - uM. \tag{49}$$

Indeed, if $D_{t-1}^j = \boldsymbol{e}_j$, the two inputs to $S_{\xi_t}^{-j}$ differ only on coordinate $j$, and $S_{\xi_t}^{-j}$ does not operate along $j$, so the pith difference remains $\boldsymbol{e}_j$. If $D_{t-1}^j = \boldsymbol{e}_j - \boldsymbol{e}_a$, then after applying $S_{\xi_t}^{-j}$, the negative coordinate is either transferred to some $b \in \mathcal{I}_j$, or disappears. This gives exactly (49).

We now bound the matrix norms of $M$. First, for every $a \in \mathcal{I}_j$,
$$\sum_{b \in \mathcal{I}_j} M_{ab} = \mathbb{P} \left( D_t^{j,\text{pith}} \neq \boldsymbol{e}_j \mid D_{t-1}^j = \boldsymbol{e}_j - \boldsymbol{e}_a \right) \leq 1. \tag{50}$$

Hence
$$\|M\|_\infty \leq 1.$$

Second, $M_{ab} = 0$ whenever $b > a$. To see this, suppose the previous negative coordinate is $a$. A transition from $a$ to $b$ means that, under the same candidate set at step $t$, the mixed path selects $a$, while the reduced path selects $b$. If $b > a$, then $b$ has score no smaller than $a$. Since $b$ is available to the mixed path whenever such a transition is considered, the mixed path would not select $a$ before $b$. Thus the negative coordinate can only stay at $a$, move to some $b < a$, or disappear.

It remains to control the column leakage. Define
$$\eta_t := \max\{q(1+q), q^2 t\}.$$

We claim that for every $b \in \mathcal{I}_j$,
$$\sum_{a=b+1}^d M_{ab} \leq \eta_t. \tag{51}$$

Fix $a > b$. Condition on both the event $D_{t-1}^j = e_j - e_a$ and the past $\sigma$-field generated by $\xi_{1:(t-1)}$. Let $A$ and $A^{-j}$ be the active sets of the two inputs to $S_{\xi_t}^{-j}$. Since the two paths differ only by replacing $a$ with $j$, their union has cardinality at most $t$. Moreover, $a \in A^{-j} \cup A$, and the transition $a \mapsto b$ requires $a$ and $b$ to be included in the fresh candidate set. After fixing $a$ and $b$, the remaining $s - 2$ sampled features must avoid every unselected feature $r > b$. Otherwise, the reduced path would select a feature with index larger than $b$, or the mixed path would not select $a$. Therefore, the remaining $s - 2$ sampled features must lie among the $b - 1$ lower-ranked features and the at most $t - 1$ active features in $(A \cup A^{-j}) \setminus \{a\}$. Thus,

$$M_{ab} \le \frac{\binom{b+t-2}{s-2}}{\binom{d}{s}}, \tag{52}$$

with the convention that the numerator is zero if $b + t - 2 < s - 2$. For $s = 1$, a transition $a \mapsto b$ with $a \ne b$ is impossible, so the column-leakage bound is trivial. Hence assume $s \ge 2$.

Let $k = d - b$. Summing (52) over $a > b$ gives

$$\sum_{a=b+1}^{d} M_{ab} \le k \frac{\binom{d-k+t-2}{s-2}}{\binom{d}{s}}. \tag{53}$$

If $k \le t$, then

$$k \frac{\binom{d-k+t-2}{s-2}}{\binom{d}{s}} \le k \frac{\binom{d-2}{s-2}}{\binom{d}{s}} = k \frac{s(s-1)}{d(d-1)} \le q^2 t.$$

If $k > t$, then

$$k \frac{\binom{d-k+t-2}{s-2}}{\binom{d}{s}} = \frac{s(s-1)}{d(d-1)} k \frac{\binom{d-k+t-2}{s-2}}{\binom{d-2}{s-2}}$$

$$\le \frac{s(s-1)}{d(d-1)} k \left(1 - \frac{k-t}{d-2}\right)^{s-2}.$$

The one-dimensional function

$$h(k) := k \left(1 - \frac{k-t}{d-2}\right)^{s-2}, \qquad k > t,$$

is maximized either at the boundary $k = t$, which yields the already bounded term $q^2 t$, or at the stationary point

$$k_* = \frac{d+t-2}{s-1}.$$

Substituting $k_*$ and using $k_* > t$ gives the elementary bound

$$\frac{s(s-1)}{d(d-1)} k_* \left(1 - \frac{k_* - t}{d-2}\right)^{s-2} \le q(1+q).$$

Consequently, (51) holds.

Combining (50) and (51), we have

$$\|M\|_\infty \le 1, \qquad \|M\|_1 \le 1 + \eta_t.$$

Indeed, for a fixed column $b$,

$$\sum_{a \in \mathcal{I}_j} M_{ab} = M_{bb} + \sum_{a>b} M_{ab} \le \sum_{c \in \mathcal{I}_j} M_{bc} + \eta_t \le 1 + \eta_t.$$

Therefore,

$$\|M\|_2^2 \le \|M\|_1 \|M\|_\infty \le 1 + \eta_t. \tag{54}$$

We now complete the contraction argument. From (47) and (49),

$$\left\|\mathbb{E}_\xi[D_{t-1}^j]\right\|_2^2 = \alpha^2 + \|u\|_2^2,$$

and

$$\left\|\mathbb{E}_\xi[D_t^{j,\text{pith}}]\right\|_2^2 = \alpha^2 + \|uM\|_2^2.$$

By (54),

$$\|uM\|_2^2 \le \|M\|_2^2 \|u\|_2^2 \le (1+\eta_t)\|u\|_2^2.$$

Hence

$$\left\|\mathbb{E}_\xi[D_t^{j,\text{pith}}]\right\|_2^2 \le \alpha^2 + (1+\eta_t)\|u\|_2^2$$
$$= \left\|\mathbb{E}_\xi[D_{t-1}^j]\right\|_2^2 + \eta_t\|u\|_2^2.$$

Since

$$\|u\|_2 \le \|u\|_1 = \sum_{a\in\mathcal{I}_j} u_a \le \alpha,$$

we have

$$\frac{\|u\|_2^2}{\alpha^2 + \|u\|_2^2} \le \frac{1}{2}.$$

Therefore,

$$\frac{\left\|\mathbb{E}_\xi[D_t^{j,\text{pith}}]\right\|_2^2}{\left\|\mathbb{E}_\xi[D_{t-1}^j]\right\|_2^2} \le 1 + \frac{\eta_t}{2}.$$

Taking square roots and using $\sqrt{1+x} \le 1 + x/2$ for $x \ge 0$, we obtain

$$\left\|\mathbb{E}_\xi[D_t^{j,\text{pith}}]\right\|_2 \le \left(1 + \frac{\eta_t}{4}\right)\left\|\mathbb{E}_\xi[D_{t-1}^j]\right\|_2.$$

Since $(@) = \mathbb{E}_\xi[D_t^{j,\text{pith}}]$, this proves (46) with

$$\delta_t = \frac{1}{4}\eta_t = \frac{1}{4}\max\{q(1+q), q^2 t\}.$$

Finally, we apply Proposition 4.5. Since $\text{rad}(\mathcal{W}_t) \le 1$, when $T \lesssim q^{-1}$, we have for every $t \le T$,

$$q^2 t \le q(1+q),$$

and thus

$$\delta_t = \frac{q(1+q)}{4}.$$

Therefore, (17) gives

$$\frac{1}{d}\sum_{j=1}^d \left\|\mathbb{E}_\xi[\boldsymbol{w}_T] - \mathbb{E}_\xi[\boldsymbol{w}_T^{-j}]\right\|_2^2 \le \left(\sum_{t=1}^T \left(1 + \frac{q(1+q)}{4}\right)^{T-t}\right)^2 \frac{q}{(d-1)(1-q)}$$

$$= \left(\frac{\left(1 + \frac{q(1+q)}{4}\right)^T - 1}{\frac{q(1+q)}{4}}\right)^2 \frac{q}{(d-1)(1-q)}$$

$$= \left(\left(1 + \frac{q(1+q)}{4}\right)^T - 1\right)^2 \frac{16}{q(1+q)^2(d-1)(1-q)}.$$

When $T \lesssim q^{-1}$, this simplifies to

$$\frac{1}{d} \sum_{j=1}^{d} \left\| \mathbb{E}_\xi[\boldsymbol{w}_T] - \mathbb{E}_\xi[\boldsymbol{w}_T^{-j}] \right\|_2^2 \lesssim \frac{T^2}{d-1} \cdot \frac{q}{1-q}.$$

This proves the theorem.

$\square$

### E.10. Proof of the Random Forest Instability Bound

*Proof of Theorem E.2.* Let $\boldsymbol{E}_{r,k} \in \mathbb{R}^{T \times d}$ denote the matrix with one at entry $(r, k)$ and zero elsewhere.

We first prove the radius condition. Fix a feasible state $\boldsymbol{w}_{t-1}$ and define the one-step candidate set

$$\mathcal{W}_t(\boldsymbol{w}_{t-1}) := \{S_{\xi_t}(\boldsymbol{w}_{t-1}) : \xi_t \text{ is an admissible realization of the tree randomness}\}.$$

Under the max-edge dyadic rule, every possible next state has the form

$$\boldsymbol{w}_{t-1} + \boldsymbol{E}_{N_{t-1}^k+1,k}$$

for some $k \in \mathcal{M}(\boldsymbol{w}_{t-1})$. If the implementation allows no split for a particular realization of $\xi_t$, the next state is simply $\boldsymbol{w}_{t-1}$, which only makes the following bound easier. Therefore, for every $\boldsymbol{w}_t \in \mathcal{W}_t(\boldsymbol{w}_{t-1})$,

$$\|\boldsymbol{w}_t - \boldsymbol{w}_{t-1}\|_F \leq 1.$$

Since the center in the definition of radius may be any point in the ambient space, choosing $\boldsymbol{w}_{t-1}$ as the center gives

$$\mathrm{rad}\big(\mathcal{W}_t(\boldsymbol{w}_{t-1})\big) \leq 1.$$

Taking the supremum over feasible $\boldsymbol{w}_{t-1}$ yields

$$\mathrm{rad}(\mathcal{W}_t) \leq 1.$$

The same argument applies to the feature-removed transition $S_{\xi_t}^{-j}$ after replacing $\mathcal{M}(\boldsymbol{w}_{t-1})$ by $\mathcal{M}^{-j}(\boldsymbol{w}_{t-1}^{-j})$.

We next prove the contraction condition. Fix a removed feature $j \in [d]$. Let

$$\boldsymbol{w}_{t-1} = S_{\xi_{t-1}} \circ \cdots \circ S_{\xi_1}(\boldsymbol{w}_0)$$

be the local encoding after running the full tree for $t-1$ steps, and let

$$\boldsymbol{w}_{t-1}^{-j} = S_{\xi_{t-1}}^{-j} \circ \cdots \circ S_{\xi_1}^{-j}(\boldsymbol{w}_0)$$

be the corresponding local encoding after running the tree with feature $j$ removed. The two runs are coupled using the same randomness $\xi = (\xi_1, \ldots, \xi_t)$.

Define the one-step increment of the feature-removed transition by

$$\boldsymbol{U}_{\xi_t}^{-j}(\boldsymbol{w}) := S_{\xi_t}^{-j}(\boldsymbol{w}) - \boldsymbol{w}.$$

For any feasible $\boldsymbol{w}$, the increment $\boldsymbol{U}_{\xi_t}^{-j}(\boldsymbol{w})$ is either zero or a standard basis matrix $\boldsymbol{E}_{r,k}$ with $k \neq j$. Hence, for any two feasible states $\boldsymbol{w}$ and $\boldsymbol{w}'$,

$$\left\| \boldsymbol{U}_{\xi_t}^{-j}(\boldsymbol{w}) - \boldsymbol{U}_{\xi_t}^{-j}(\boldsymbol{w}') \right\|_F \leq \sqrt{2}\,\mathbf{1}\{\boldsymbol{w} \neq \boldsymbol{w}'\}.$$

If $\boldsymbol{w} = \boldsymbol{w}'$, the two increments are identical because the split rule and the tie-breaking rule are deterministic.

Now let

$$\Delta_{t-1}^{-j} := \mathbb{E}_\xi \left[ \boldsymbol{w}_{t-1} - \boldsymbol{w}_{t-1}^{-j} \right].$$

We claim that

$$\mathbb{P}_\xi \left( \boldsymbol{w}_{t-1} \neq \boldsymbol{w}_{t-1}^{-j} \right) \leq \|\Delta_{t-1}^{-j}\|_F.$$

Indeed, if the full run has never selected feature $j$ during the first $t-1$ steps, then the full run and the feature-removed run have the same local cell, the same max-edge candidates after excluding $j$, and the same split scores on all features in $[d] \setminus \{j\}$. By deterministic tie-breaking, the two local paths are identical. Therefore,

$$\left\{ \boldsymbol{w}_{t-1} \neq \boldsymbol{w}_{t-1}^{-j} \right\} \subseteq \left\{ N_{t-1}^{j} \geq 1 \right\}.$$

Since the encoding is column-filled and the feature-removed tree has zero in the $j$-th column, the $(1, j)$ entry of $\Delta_{t-1}^{-j}$ is

$$\left( \Delta_{t-1}^{-j} \right)_{1,j} = \mathbb{P}_{\xi} \left( N_{t-1}^{j} \geq 1 \right).$$

Thus,

$$\mathbb{P}_{\xi} \left( \boldsymbol{w}_{t-1} \neq \boldsymbol{w}_{t-1}^{-j} \right) \leq \mathbb{P}_{\xi} \left( N_{t-1}^{j} \geq 1 \right) = \left| \left( \Delta_{t-1}^{-j} \right)_{1,j} \right| \leq \| \Delta_{t-1}^{-j} \|_{F}.$$

Using the decomposition

$$S_{\xi_t}^{-j}(\boldsymbol{w}) = \boldsymbol{w} + \boldsymbol{U}_{\xi_t}^{-j}(\boldsymbol{w}),$$

we obtain

$$\begin{aligned}
&\left\| \mathbb{E}_{\xi} \left[ S_{\xi_t}^{-j}(\boldsymbol{w}_{t-1}) - S_{\xi_t}^{-j}(\boldsymbol{w}_{t-1}^{-j}) \right] \right\|_{F} \\
&= \left\| \Delta_{t-1}^{-j} + \mathbb{E}_{\xi} \left[ \boldsymbol{U}_{\xi_t}^{-j}(\boldsymbol{w}_{t-1}) - \boldsymbol{U}_{\xi_t}^{-j}(\boldsymbol{w}_{t-1}^{-j}) \right] \right\|_{F} \\
&\leq \| \Delta_{t-1}^{-j} \|_{F} + \mathbb{E}_{\xi} \left[ \left\| \boldsymbol{U}_{\xi_t}^{-j}(\boldsymbol{w}_{t-1}) - \boldsymbol{U}_{\xi_t}^{-j}(\boldsymbol{w}_{t-1}^{-j}) \right\|_{F} \right] \\
&\leq \| \Delta_{t-1}^{-j} \|_{F} + \sqrt{2} \, \mathbb{P}_{\xi} \left( \boldsymbol{w}_{t-1} \neq \boldsymbol{w}_{t-1}^{-j} \right) \\
&\leq (1 + \sqrt{2}) \| \Delta_{t-1}^{-j} \|_{F}.
\end{aligned}$$

Equivalently,

$$\left\| \mathbb{E}_{\xi} \left[ S_{\xi_t}^{-j}(\boldsymbol{w}_{t-1}) - S_{\xi_t}^{-j}(\boldsymbol{w}_{t-1}^{-j}) \right] \right\|_{F} \leq (1 + \delta_t) \left\| \mathbb{E}_{\xi} \left[ \boldsymbol{w}_{t-1} - \boldsymbol{w}_{t-1}^{-j} \right] \right\|_{F}$$

with

$$\delta_t = \sqrt{2}.$$

This proves the contraction condition required by Proposition 4.5. Combining this contraction condition with the radius bound $\mathrm{rad}(\mathcal{W}_t) \leq 1$ and applying Proposition 4.5 yields (35). The prediction bound follows from Proposition E.1. $\qquad\square$

### E.11. Proof of the Finite-Bagging Instability Bound

***Proof of Proposition E.3.*** Note that Hoeffding's inequality in Hilbert space (e.g., Boucheron et al. (2003)) yields

$$\left\| \frac{1}{B} \sum_{b=1}^{B} \boldsymbol{w}^{(b)} - \mathbb{E}_{\xi} [\boldsymbol{w}] \right\| \leq \sqrt{\frac{3 \, \mathrm{rad} \, (\mathcal{W})^2 \log \left( (d+1)/\delta \right)}{B}} \tag{55}$$

with probability $1 - \delta/(d+1)$, and

$$\left\| \frac{1}{B} \sum_{b=1}^{B} \boldsymbol{w}^{(b),-j} - \mathbb{E}_{\xi} [\boldsymbol{w}^{-j}] \right\| \leq \sqrt{\frac{3 \, \mathrm{rad} \, (\mathcal{W})^2 \log \left( (d+1)/\delta \right)}{B}} \tag{56}$$

with probability $1 - \delta/(d+1)$ for each $j \in [d]$. Applying union bound, we have (55) and (56) hold simultaneously for all $j \in [d]$ with probability at least $1 - \delta$. Then, with Minkowski's inequality, we have

$$
\frac{1}{d} \sum_{j=1}^{d} \left\| \frac{1}{B} \sum_{b=1}^{B} \boldsymbol{w}^{(b)} - \frac{1}{B} \sum_{b=1}^{B} \boldsymbol{w}^{(b),-j} \right\|^2 \leq \frac{\sqrt{3}}{d} \sum_{j=1}^{d} \left\| \frac{1}{B} \sum_{b=1}^{B} \boldsymbol{w}^{(b)} - \mathbb{E}_\xi [\boldsymbol{w}] \right\|^2
$$

$$
+ \frac{\sqrt{3}}{d} \sum_{j=1}^{d} \left\| \mathbb{E}_\xi [\boldsymbol{w}] - \mathbb{E}_\xi [\boldsymbol{w}^{-j}] \right\|^2
$$

$$
+ \frac{\sqrt{3}}{d} \sum_{j=1}^{d} \left\| \frac{1}{B} \sum_{b=1}^{B} \boldsymbol{w}^{(b),-j} - \mathbb{E}_\xi [\boldsymbol{w}^{-j}] \right\|^2
$$

$$
\leq \frac{\sqrt{3}}{d} \sum_{j=1}^{d} \left\| \frac{1}{B} \sum_{b=1}^{B} \boldsymbol{w}^{(b)} - \mathbb{E}_\xi [\boldsymbol{w}] \right\|^2
$$

$$
+ \frac{6\sqrt{3} \operatorname{rad}(\mathcal{W})^2 \log((d+1)/\delta)}{B}.
$$

$\square$

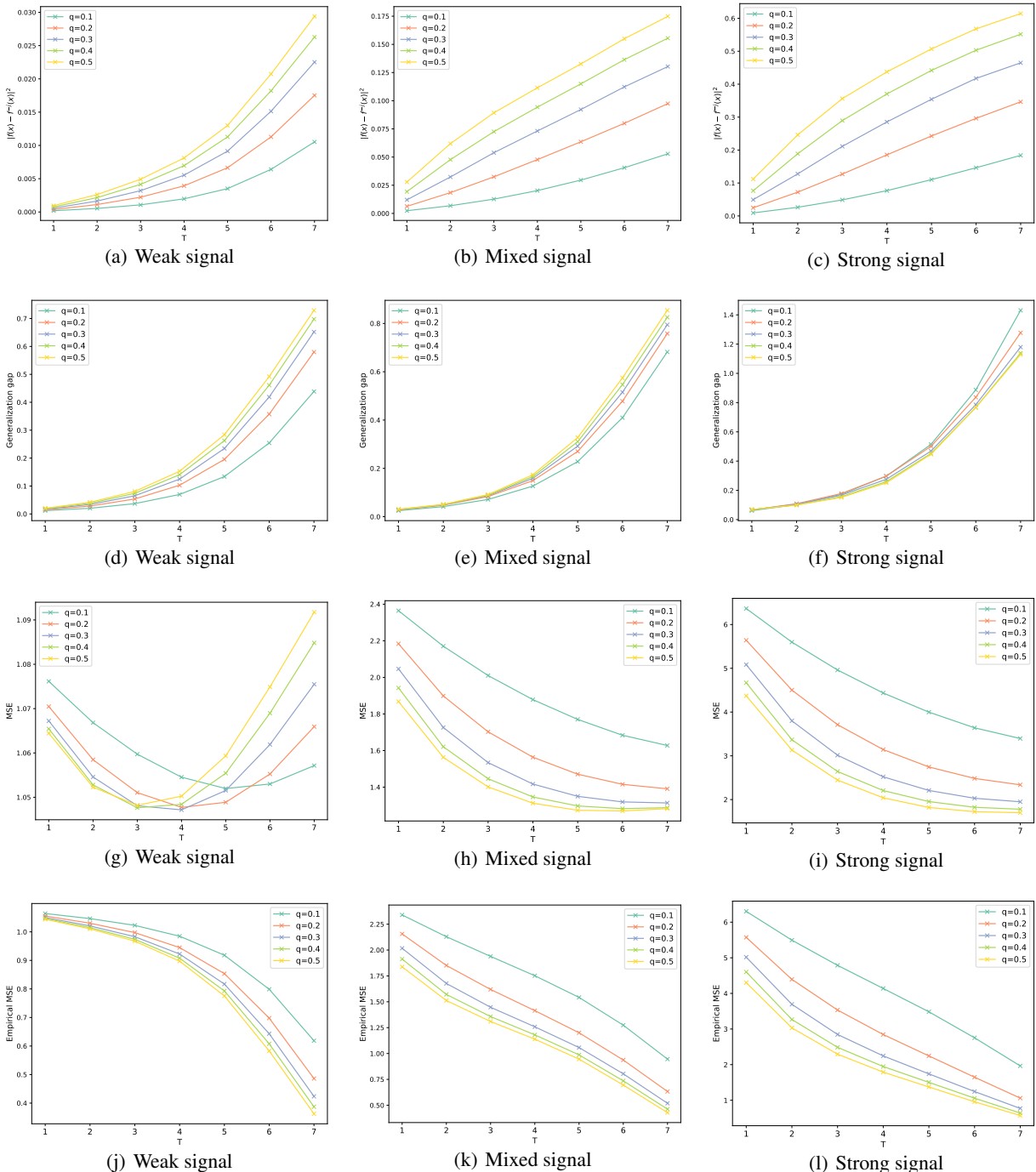

*Figure 8.* The feature instability, generalization gap, generalization error, and empirical risk of RF under different signal types. Each cross corresponds to 100 repetitions.

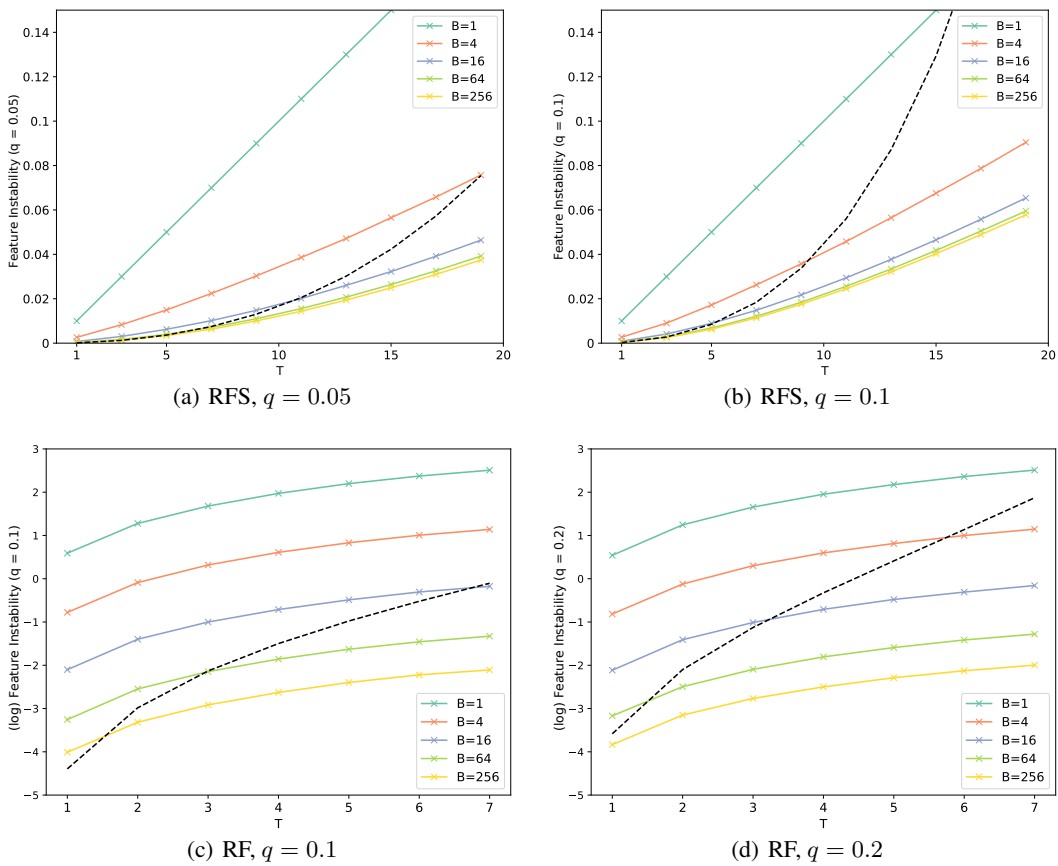

(a) RFS, $q = 0.05$

(b) RFS, $q = 0.1$

(c) RF, $q = 0.1$

(d) RF, $q = 0.2$

*Figure 9.* Comparison of feature instability across different numbers of bagging rounds $B$. Each cross corresponds to 100 repetitions. The distributions used by RFS and RF are the same as those in Figure 3 and 7. The black lines are instability bounds from Theorems 4.6 and 4.7.

