# OpenReview forum: "Feature Bagging Provides Stability"
_ICML.cc/2026/Conference — ICML 2026 regular_

### Official Review · Reviewer_ExYY · 2026-02-14

**Soundness:** 3
**Presentation:** 3
**Significance:** 3
**Originality:** 3
**Overall Recommendation:** 4
**Confidence:** 4

**Summary:**

The paper studies whether feature bagging improves algorithmic stability, and proposes a new stability notion called feature stability (FS), defined as the average prediction change when removing a single feature. The paper claims (i) FS complements IS and carries non-redundant information about generalization, (ii) in a Gaussian linear regression model, feature bagging improves both FS and IS in a proportional-asymptotic sense, with stability improving as the number of bags increases, and (iii) in an model-free framework, feature bagging yields general high-probability FS guarantees, and only O(d) bags suffice to approach the infinite-bagging stability regime (with d the dimension).

**Compliance With Llm Reviewing Policy:**

Affirmed.

**Ethical Review Concerns:**

This is a theoretical work with no ethics review needed.

**Final Justification:**

Thank you for the clarification. The generalized coupling view makes the Sec. 3 vs. Sec. 4 distinction much clearer. I think the revision should explicitly state in the main text which coupling is used in each section, and should clearly specify the feature-removal convention for general learners. With those clarifications, I consider the main concerns largely resolved, and I lean positively towards acceptance, as already mentioned in my review.

**Key Questions For Authors:**

1. What is the exact stability notion used for the randomized (bagging/subsampling) algorithms in Sec. 3?
In Defs. 2.1–2.2, the same randomness symbol ξ appears in both the full and perturbed datasets, but in Sec. 3.1 instance/feature removal is modeled via independently resampled selection matrices from reduced index sets (Eqs. (9)–(10)). Please clarify the coupling and justify why the analyzed quantity corresponds to your stated IS/FS definitions. If the current analysis matches only a different stability notion, the main claims about “stability” may need reframing.

2. How should FS be computed for general learners when removing a feature changes the input dimension? A clear convention is important for reproducibility and for interpreting FS as “robustness to missing/perturbed features.”

3. How sensitive are the FS–generalization conclusions to the chosen dependence estimators and hyperparameter grids? If the FS–generalization link is unstable to estimator choices, the motivation for FS as a general diagnostic weakens.

4. Can you add experiments beyond synthetic random-forest regression to demonstrate FS’s usefulness more broadly? For example, real tabular benchmarks, comparing FS/IS across different ensemble methods, which will provide stronger empirical breadth.

**Limitations:**

Yes, the Impact Statement is sufficient.

**Strengths And Weaknesses:**

Soundness

A major strength is that the paper provides formal guarantees in two quite different regimes: a parametric proportional-asymptotics analysis for linear regression with subsampling, and an assumption-free framework for recursive feature subsampling inspired by random forests. In the linear setting, the main decomposition separates instability into variance vs. bias-driven terms and also into “same subsample” vs “different subsample” contributions that scale differently in the number of bags and sample size. These are consistent with known results that ensembling stabilizes variance spikes in overparameterized regimes. In the model-free part, the work effectively “dualizes” existing instance-bagging stability bounds to obtain feature-bagging FS bounds under bounded-range assumptions, and then extends to a Hilbert-space-valued recursion to treat recursive feature subsampling.

I am finding a potential conceptual mismatch between (a) the stability definition stated earlier (which conditions on a fixed random seed of algorithmic randomness) and (b) how the linear-regression section defines the “removed-instance/removed-feature” estimator for subsampled bagging. Removing a feature is modeled by independently resampling the selection matrices from the reduced index set, which corresponds to comparing two ensembles built with different subsampling realizations rather than coupling them by the same randomness. A second caveat is the random forest modeling abstraction. The analysis in the Appendix studies a simplified dyadic/median-split tree model and relates prediction differences to a “feature-weight matrix” encoding of splits along the path relevant to a test point, then bounds stability in Frobenius norm. This is a reasonable theoretical simplification, but it departs from the greedy CART splitting of practical random forests, and the paper’s main text should do more to delimit what is and is not captured about modern implementations.

Presentation

The narrative is clear and well-structured. The definition of FS compares predictions from models trained with a feature removed; for general learners, this can imply a change in input dimensionality. The paper should state more explicitly how prediction inputs are aligned (e.g., whether one views the removed feature as absent at test time, imputed, or the predictor embedded back into the full space) to avoid ambiguity.

Significance

Understanding why feature bagging helps and how it relates to stability is relevant. Bagging as a stability mechanism is already well-grounded theoretically in the instance-sub-bagging setting. This paper’s attempt to extend the stability story to feature bagging and to recursive feature subsampling is a meaningful contribution. The paper also proposes FS as an additional stability dimension and provides empirical evidence that FS contains information about generalization not captured by IS alone. If the community adopts FS as a standard diagnostic (analogous to how IS is used in some theoretical and applied contexts), that would be impactful. On the other hand, the empirical scope is currently narrow. The FS–generalization link is demonstrated on a synthetic-additive benchmark with random forests and a specific hyperparameter sweep; the paper does not include evidence that FS is similarly explanatory for other model classes where feature bagging/feature subsampling arises. This limits practical significance, and the paper is positioned as theoretical.

Originality

The paper's originality is strongest in its framing: introducing “feature stability” as a dual to instance stability and then building a unified story about how different bagging axes influence both stability notions. Older random subspace ensemble literature explicitly discusses stability improvements from random feature subsets. The paper should cite and contrast with this line to clarify what is new: formal FS definitions, modern stability-theory framing, sharper asymptotic characterizations, etc. There is a recent theory on feature-subsampled linear ensembles that studies learning curves and double-descent under feature subsampling and ensembling. The paper does cite key modern “randomization as regularization” work for random forests and related explanations.

---

> ### Author Rebuttal · Authors · 2026-03-29
>
> We thank the reviewer for the constructive comments. We sincerely hope that the rebuttal has addressed the concerns and would greatly appreciate it if the reviewer could consider updating the score accordingly.
> - **Weakness and question 1: stability notion mismatch:**  We agree that the current presentation does not distinguish these two notions clearly enough. The analysis in Sec. 3 and Sec. 4 are intended to serve different purposes. In Sec. 3, under a parametric model, we analyze a *distributional perturbation* notion: under distributional assumptions, we study precisely how stability behaves. In Sec. 4, by contrast, FS is a *conditional-on-the-dataset algorithmic* property defined with respect to the algorithm's randomness, and our goal is to provide a uniform guarantee purely at the algorithmic level. These two notions are not contradictory. The quantity $\mathcal{A}(\mathcal{D},\xi)$ only fixes the random realization under $\mathcal{D}$, while the randomness in $\mathcal{A}(\mathcal{D}^{-j}, \xi)$ may still be completely different from that in $\mathcal{A}(\mathcal{D}, \xi)$. We will revise the outline to explicitly clarify their purposes.
> - **Weakness and question 2: FS when dimension changes:** We thank the reviewer for this intriguing question. For most ML models, working with $\mathcal{D}$ and $\mathcal{D}^{:,-j}$ is not substantially different: the training procedure simply uses the reduced data matrix; for neural networks, one changes only the input dimension while keeping the hidden dimensions fixed. At test time, one simply ignores the removed feature. If one must work with models with a fixed input dimension, we suggest using $\frac{1}{d^2}\sum_{i,j}(f^{:,-i}(x)-f^{:,-j}(x))^2$, which serves as a lower-bound proxy for $4\times\mathrm{FS}$.
> - **Weakness and questions 3/4: robustness of the FS-generalization/real-data:** We thank the reviewer for the helpful suggestions and have conducted a substantially broader empirical study. Along with the discussion below, please see the corresponding tables in the anonymous PDF (https://anonymous.4open.science/r/icml26-2EB3/tables.pdf). Our added experiments suggest that the FS-generalization conclusions are not sensitive to the tested dependence estimators or hyperparameter grids, and persists on real datasets:
>     - For **real-data**, we extended the evaluation to four real tabular regression benchmarks: `diabetes`, `abalone`, `cpu`, and `housing`.
>     - For **robustness to dependence estimators and hyperparameter grids**, we evaluated four base estimator families (RF, DT, AdaBoost, GBRT), sweeping 105/35/63/63 hyperparameter settings, respectively, and ran 100 bootstrap replicates, each subsampling one-third of the hyperparameter grid. For the conditional-dependence analysis, we also varied the PMI residualizer across RF/DT/AdaBoost/GBRT.
>     - **Results:** Table 1-4 directly address both concerns. First, the FS-generalization conclusions are not sensitive to the tested dependence estimators or hyperparameter grids: in the synthetic summaries, every displayed entry is positive in both PMI and $R^2$ across all $\nu$ settings, four base estimator families, and four PMI residualizers; more globally, across all 160 groups, both metrics remain positive and the bootstrap positive rate is 1.0 for every group. Second, the same qualitative pattern persists on the four real tabular benchmarks, where every displayed dataset-method pair remains positive in both metrics.
>     - To show that the gain is valid, we also conducted a complementary noise-control check by replacing FS with a Gaussian-noise baseline and then rerunning the same conditional-dependence analysis. Tables 5-9 report the differences; $\Delta R^2(FS)-\Delta R^2(noise)$ is positive in all noise-control entries, and $PMI(FS;gap\mid IS)-PMI(noise; gap\mid IS)$ is positive in 319/320 entries, with the exception numerically negligible. Thus, FS appears empirically more informative than a random covariate.
> - **Weakness: simplified tree model:** We agree that the current study is limited to simplified tree models. Yet, we expect a rigorous analysis of CART to require more refined tools from tree theory, in particular of partitions (works like [1,2]), which is known to be challenging.
>     1. Eliza O'Reilly, Statistical Advantages of Oblique Randomized Decision Trees and Forests.
>     2. Mehdi Dagdoug, Cl\'ement Dombry, and Jean-Jil Duchamps, An RKHS Perspective on Tree Ensembles.
> - **Weakness: relation to literature:** We will position our work more explicitly, especially to the literature on feature-subsampled linear ensembles. Much of this literature focuses on prediction risk, whose characterization typically requires delicate random-matrix or related asymptotic calculations. Our focus here is on stability, which is a complementary and often more tractable quantity that admits both exact analysis in the parametric setting and general guarantees in the model-free setting.

---

> > ### Author Rebuttal · Reviewer_ExYY · 2026-04-01
> >
> > Thank you for the detailed rebuttal. The clarification on prediction under feature bagging is helpful. So, at test time, each base learner uses its own sampled feature subset, and the ensemble prediction is the average across base learners. That should be stated explicitly in the paper.
> >
> > The added empirical evidence is also useful. The FS–generalization observations are robust across multiple estimator families, hyperparameter subsamples, PMI residualizers, and several real tabular datasets.
> >
> > That said, I still view one conceptual issue as only partially resolved: the paper’s main definitions present FS/IS using a shared-randomness coupling, while the Sec. 3 linear-regression analysis compares ensembles built from independently resampled subsampling realizations. You mentioned that Sec. 3 and Sec. 4 are serving different purposes and are not contradictory, but I still think the paper should make this distinction explicit in the main text and be careful about which stability notion is being analyzed in each section. The convention for handling feature removal when the input dimension changes should be stated clearly in the paper for reproducibility.
> >
> > Your answer addresses the empirical breadth concern well enough for me to remain positive on the submission.

---

> > > ### Author Response · Authors · 2026-04-02
> > >
> > > We sincerely thank the reviewer for these insightful comments, which have helped improve both the clarity and the overall quality of the paper. We are also grateful that the added experiments have addressed the empirical-breadth concern. On the remaining conceptual point, we would like to make the distinction between Sec. 3 and Sec. 4 fully explicit.
> > >
> > > A clean way to state this is to generalize the FS notion through a coupling family. Specifically, for each feature index $j$, let $\Gamma_j(\mathcal{D})$ be a coupling on the seed spaces used for the full-data and feature-removed runs, respectively. Then one may define feature stability under $\Gamma$ by
> > > $$
> > > \frac{1}{d}\sum_{j=1}^d
> > > \mathbb{E}_{(\xi,\xi^j)\sim \Gamma_j(\mathcal{D})}[(\mathcal{A}(\mathcal{D};\xi)(x) - \mathcal{A}(\mathcal{D}^{:,-j};\xi^j)(x))^2].
> > > $$
> > > Under this generalized view, the shared-randomness notion in Defs. 2.1-2.2 and Sec. 4 corresponds to the diagonal coupling $\xi^j=\xi$, while the linear analysis in Sec. 3 corresponds to an independent-resampling coupling. Thus, Secs. 3 and 4 are not intended to analyze literally the same coupling, but rather two parallel special cases of a common generalized stability notion.
> > > The definition also separates two ingredients: the randomized algorithm $\mathcal{A}$ itself, and the coupling rule $\Gamma_j(\mathcal{D})$ used to compare the full-data and feature-removed runs.
> > >
> > > This also clarifies the role of $\xi$. Our intended interpretation is that $\xi$ denotes the algorithm's random seed, rather than a realized shared subsample itself. A shared-randomness coupling can be defined by letting the random seed generate a primitive random ordering of the features. For example, let $\xi = (r_1,\dots,r_d)$ be i.i.d. continuous random priorities assigned to the $d$ features. For any admissible feature set $A \subseteq [d]$, define $\nu(\xi, A)$ to be the set of the $s$ indices in $A$ with the smallest priorities. Then the feature subset used on the full dataset is $\nu_{\mathrm{full}} = \nu(\xi, [d])$, while the feature subset used after removing feature $j$ is $\nu_{\mathrm{rem}} = \nu(\xi, [d]\setminus\{j\})$. These two subsets are generally not identical, but they are still coupled through the same underlying seed $\xi$.
> > >
> > > Accordingly, in the revision we will explicitly state, for each algorithm studied in the paper, how the coupling/decoupling is handled, together with the general underlying principle. In particular, we will state clearly that Sec. 3 analyzes the independent-resampling coupling for subsampled linear ensembles, whereas Defs. 2.1--2.2 and Sec. 4 use the shared-randomness coupling. We will also bring forward the generalized FS notion above to unify these settings.
> > >
> > > We hope that these clarifications fully resolve the remaining concern and further improve the presentation. We would be grateful if you could take them into account in your final evaluation.

---

### Official Review · Reviewer_fqR9 · 2026-02-25

**Soundness:** 4
**Presentation:** 4
**Significance:** 3
**Originality:** 3
**Overall Recommendation:** 5
**Confidence:** 2

**Summary:**

This paper, introduces a new notion of stability called feature stability. This notion of stability is inspired by another stability notion which they call instance stability (also known as uniform stability). An algorithm that is feature stable will produce a classifier with similar loss as if we gave it the same dataset, but removed one of the features. The main point of the paper is to show that doing bagging on the features improves feature stability.

After defining feature stability, the paper uses linear regression as an example, where they show asymptotic bounds on both feature stability and instance stability when utilizing the corresponding bagging approach.
The paper then moves on to model-free results where they manage to show that for bounded loss functions, feature bagging will make any base algorithm feature stable in the limit (that is, when doing infinitely many subsamples). They also prove similar bounds in a recursive framework that they introduce which is then applied to random forward selection.

Furthermore, their theoretical results are backed up with experiments which empirically estimate the feature stability and instance stability when subsampling at different rates.

**Compliance With Llm Reviewing Policy:**

Affirmed.

**Final Justification:**

Most of the concerns in my review was very minor, and all of these were properly addressed in the rebuttal of the authors. My major question was about the connection between feature stability and generalization bounds, as I believe such a connection would strengthen the motivation for looking at feature stability in the first place. The authors explain what difficulties they see in proving such connections and provide a guess for what kind of connection they think one might be able to prove (please see their rebuttal for details on this).
I think their analysis of how feature bagging affects feature stability is still sufficient for an ICML paper which is why I keep my recommendation.

**Key Questions For Authors:**

1. What are your thoughts on proving a connection between generalization and feature stability?\
2. Do you believe that it is possible to show similar generalization bounds for feature stability as those done by Bousquet & Elisseeff, (2002) for instance stability?

**Limitations:**

yes

**Strengths And Weaknesses:**

**Strengths:**\
The paper is well-structured and all the necessary notation/definitions are lined out nicely before/when used. The authors also do a good job explaining the different kinds of bagging in detail before moving on which could otherwise have been very confusing.\
The authors provide formal proofs to all the claims of the paper (although I have not read them), and they even provide experiments backing up the theoretical claims.
Lastly, the definition of feature stability seems very natural and therefore worth investigating.

**Weaknesses:**\
I think the main weakness of the paper is in the theoretical motivation of achieving feature stability for an algorithm. For instance stability, you can formally prove that this property implies generalization of the algorithm. A similar result (either positive or negative) for feature stability would significantly improve the impact of the paper.
With that said, the paper does motivate feature stability in section 2.2 and they do show empirical evidence which suggest that a similar connection exists for feature stability.

Beyond the above, I have a few minor comments, which should be easy to address:
- 034(right): I don't understand the point of the toy example. How does it make sense to relate to equation (1)? And how can you say that (1) is constant, when changing features? It doesn't say anything about what happens in that case right?
- 080(left): When you say "either instance bagging and feature bagging" it confuses me. Should it be either-or or both-and?
- Could you make it more clear where one can find the proof of Theorem 3.1 in the appendix? I wasn't immediately able to find it.
- In connection to the above point, can you also make it clear where to find the precise definition of $\Delta_\ell^{V=},\Delta_\ell^{V\neq},\Delta_\ell^{B=},\Delta_\ell^{B\neq}$? (I couldn't find it in the appendix)
- In Figure 2 you swapped instance subsampling and feature subsampling.
- 373(left): It strictly generalizes Proposition 4.2 (not 4.5).
- The formulation of Proposition 4.6 is confusing. I guess you mean to say Assumption 4.4. implies (16). However, it sounds like you are assuming both assumption 4.4 and equation (16).

---

> ### Author Rebuttal · Authors · 2026-03-29
>
> We thank the reviewer for the positive assessment, carefully checking of the manuscript, and the helpful comments.
> We sincerely hope that the rebuttal has addressed the concerns and would greatly appreciate it if the reviewer could consider updating the score accordingly.
> - **Question 1/2: FS and generalization connection:**  We agree that a formal FS-generalization theorem would substantially strengthen the paper.
> Our goal in the current work is more modest: to introduce FS, analyze how feature bagging affects FS theoretically, and provide empirical evidence that FS carries information about generalization beyond IS.
>     - At a technical level, the main obstacle to a Bousquet-Elisseeff-type result is that the classical leave-one-out argument exploits an independence structure created by removing one training example: the removed example is independent of the predictor trained without it.
>     Removing a feature does not create an analogous independent object in general, since the removed coordinate and the remaining coordinates may be strongly dependent.
>     Thus, FS controls sensitivity along the feature axis, whereas the usual generalization gap is defined along the sample axis.
>     For this reason, a direct analogue of the classical IS-to-generalization bound does not seem immediate.
>     More specifically,
>     The classical argument hinges on $\mathbb{E}[\ell(f^{-i,:},Z)]=\mathbb{E}[\ell(f^{-i,:},Z_i)]$, since $f^{-i,:}$ is independent of both $Z$ and $Z_i$. For feature deletion, the analogous identity $\mathbb{E}[\ell_{-j}(f^{:,-j},Z)]=\mathbb{E}[\ell_{-j}(f^{:,-j},Z_i)]$ fails in general, because $f^{:,-j}$ is still trained on $(X_i^{-j},Y_i)$.
>     - Currently, our best guess would be: a meaningful FS theorem may need a feature-wise analogue of generalization - for example, a notion of robustness under feature deletion, corruption, or exclusion - rather than only the standard sample-wise train-test gap.
>     One possible direction is to quantify a feature-wise robustness gap, namely how much performance or predictive attribution changes when a feature is removed or perturbed; this is closer in spirit to feature-attribution or knockoff-style questions than to classical uniform stability.
>     - We will revise the paper to present this interpretation more clearly and to state that establishing a formal FS-generalization theorem is an important direction for future work.
> - We thank the reviewer for the careful reading and for pointing out these minor issues:
>     - **Toy example vs (1):** We agree that the current presentation is confusing. The toy example is intended only as a heuristic motivation for why a feature-wise notion is needed: regardless of the proportion of edited features, Eq. (1) (or its feature-editing analogue, $\frac{1}{d} \sum_{j=1}^d \left( f(x)- f^{:,(j)}(x)\right)^2$) remains unchanged, even though predictive performance may differ substantially depending on the amount of feature editing. We will rewrite this part to make the logic clearer.
>     - **Either-or wording:** We apologize for the wording error. This should read ``either instance bagging or feature bagging.'' The intended point is that bagging along either axis can improve stability; in particular, feature bagging alone can sometimes improve IS, and vice versa.
>     - **Location of Thm 3.1 proof and $\Delta$ definitions:** We apologize for the lack of a clear pointer and will add one in Appendix C. Currently, Theorem 3.1 follows directly from Theorems C.1 and C.2, which in turn follow from Propositions C.3 and C.4. As for the four $\Delta$ terms, they are intermediate quantities introduced to make the decomposition interpretable: $V/B$ correspond to variance/bias contributions, while $=/\neq$ distinguish same-subsample versus different-submodel terms. We will state this explicitly and point to their formal definitions in the appendix.
>     - **Figure 2 swapped labels / Incorrect proposition reference:** We thank the reviewer for catching these issues and will correct them.
>     - **Unclear Prop. 4.6 statement:** We apologize for the inaccurate wording and will rewrite the statement to make the intended implication clear.

---

> > ### Author Rebuttal · Reviewer_fqR9 · 2026-03-31
> >
> > The authors have addressed my points satisfactorily. Given the authors incorporate the revisions they promise in this and the other rebuttals, I believe accept is an appropriate score.

---

> > > ### Author Response · Authors · 2026-04-04
> > >
> > > Thank you for your thoughtful follow-up and positive assessment. We are glad to hear that our responses adequately addressed all your concerns.
> > >
> > > If there are any remaining issues or points that still need clarification, we would be very happy to address them as clearly as possible. We appreciate your continued consideration of our work.

---

### Official Review · Reviewer_fQtK · 2026-03-07

**Soundness:** 3
**Presentation:** 3
**Significance:** 3
**Originality:** 3
**Overall Recommendation:** 4
**Confidence:** 3

**Summary:**

In contrast to instance stability, this paper introduces the notion of feature stability. It shows that feature stability plays an important role in generalization and cannot be substituted by instance stability. The paper systematically analyzes feature bagging in both parametric and nonparametric settings. Under model assumptions, the expected perturbation is characterized within a representative linear regression framework. A decomposition into variance and bias components clarifies how bagging suppresses instability, showing that sufficiently many bagging rounds remove divergence at the interpolation threshold and that smaller feature subsampling ratios lead to stronger stabilization. In model-free settings, the analysis is extended to a general Hilbert space, where stability bounds for feature-subsampled bagging are established and applied to recursive feature subsampling procedures, covering random-forest–type methods.

**Compliance With Llm Reviewing Policy:**

Affirmed.

**Final Justification:**

My concerns have been adequately addressed. And I now support a weak accept recommendation.

**Key Questions For Authors:**

* The parametric analysis relies on Gaussian linear regression with isotropic features under the proportional asymptotic regime. How sensitive are the theoretical conclusions to these assumptions? In particular, do the stability improvements predicted by the theory still hold when features are correlated or when the data distribution deviates from the Gaussian setting?
* The theoretical analysis treats features symmetrically and does not explicitly account for feature correlations or heterogeneous feature importance. How would the proposed FS framework behave in settings where features have strong correlation structures or varying levels of relevance?
* In Proposition 4.6, could the authors clarify the definition of $\delta_t$? The text refers to “proof (4.4)” for the specification of $\delta_t$, but I was unable to locate it. Is this a typographical error, and if so, could the authors indicate the correct reference and provide a clearer definition of the “concentration parameters” $\delta_t$?

**Limitations:**

Yes. The authors discuss several limitations of their work, including that the theoretical analysis focuses on specific settings such as Gaussian linear regression and recursive feature subsampling frameworks, and that some results rely on assumptions such as bounded outputs or bounded hypothesis spaces. In addition,  the discussion could be further strengthened by clarifying how the theoretical assumptions may limit applicability to more complex real-world datasets and other learning models.

**Strengths And Weaknesses:**

**Strengths**
* **Introduction of Feature Stability**  The paper introduces Feature Stability (FS) as a dual notion to Instance Stability (IS), thereby extending classical stability analysis and establishing a unified framework for algorithmic stability with respect to both instance-level and feature-level perturbations.
* **Empirical evidence highlighting the importance of Feature Stability**  The paper demonstrates empirically that Feature Stability (FS) carries substantial information about generalization performance that is not redundant with instance stability, effectively motivating FS as a meaningful concept for understanding the impact of feature bagging.
* **Theoretical analysis across both parametric and model-free settings** The paper analyzes feature bagging in both parametric (Gaussian linear regression) and model-free settings, establishing theoretical guarantees for stability improvement, and the empirical results are consistent with the theoretical predictions.

**Weaknesses**
* **Soundness:** Several results rely on relatively strong modeling assumptions that may limit the applicability. In the parametric analysis, the theory is developed under a Gaussian linear regression model with isotropic features and a proportional asymptotic regime. These assumptions cannot reflect practical settings where features are correlated and noise is heterogeneous. In the model-free analysis, the stability guarantees rely on bounded outputs or bounded-radius assumptions in a Hilbert space, which may not hold for some modern learning models such as deep neural networks. Moreover, the connection between FS and generalization performance is mainly supported by empirical observations, lacking stronger theoretical generalization guarantees.
* **Presentation:** The figures in the paper are relatively small, making some labels and annotations difficult to read. In addition, several important components, such as the experimental validation of the significance of Feature Stability and the analysis of feature stability under finite bagging, are placed in the appendix, which somewhat affects the overall readability.
* **Significance:**  The contributions of this work mainly lie in theoretical explanations, while the empirical evaluation is relatively limited and mainly illustrates theoretical trends. Consequently, the practical impact of the proposed results remains to be further validated.
* **Originality:**  The theoretical analysis largely builds on existing stability theory and bagging frameworks, extending stability analysis from the instance dimension to the feature dimension. As a result, the novelty primarily lies in extending existing theoretical tools. In addition, the analysis does not account for practical feature properties such as correlation or heterogeneity, which may limit the originality and applicability of the framework in more realistic data settings.

---

> ### Author Rebuttal · Authors · 2026-03-29
>
> We thank the reviewer for the detailed assessment and thoughtful questions. We sincerely hope that the rebuttal has addressed the concerns and would greatly appreciate it if the reviewer could consider updating the score accordingly.
> - **Weakness and Question 1/2: sensitivity of the results to (non-Gaussian/isotropic/correlated) features/varying dependence/strong assumptions in the model-free setting:** We believe these questions should be viewed from two perspectives, since Sec 3 and 4 serve different purposes and study related but distinct stability notions (not identical, though not contradictory; see our response to Reviewer ExYY for details):
>     - Section 3 (parametric analysis): Under a parametric model, we study a *distributional perturbation* notion. Under explicit distributional assumptions, this lets us characterize precisely how stability behaves under different bagging schemes. In this setting, the current analysis can in principle be extended to:
>         - **Non-gaussian distributions**: random noise and feature distributions with bounded moments, such as finite $(4+\epsilon)$ moments [3,4].
>         - **Correlated features / non-isotropic designs**: covariance matrices with eigen values bounded away from 0 and $\infty$, which can be analyzed using random matrix theory [3,4]. In these cases, however, the results generally do not admit closed-form expressions and are often less interpretable.
>             3. Edgar Dobriban, Stefan Wager, High-Dimensional Asymptotics of Prediction: Ridge Regression and Classification.
>             4. Trevor Hastie, Andrea Montanari, Saharon Rosset, Ryan J. Tibshirani, Surprises in High-Dimensional Ridgeless Least Squares Interpolation.
>         - **Varying level of dependence**: in the absence of additional information, it is natural to assume a homogeneous prior for $\beta$. However, it is technically straightforward to allow $\beta$ to be non-isotropic, which mainly changes the relevant quadratic-form calculations.
>     - Section 4 (model-free analysis): By contrast, FS here is a *conditional-on-the-dataset algorithmic* property defined with respect to the algorithm's randomness, and our goal is to prove uniform guarantees at the algorithmic level. From this perspective, feature correlation, varying dependence levels, and non-Gaussianity are not obstacles, as they are all dataset properties. We do agree, however, with the reviewer that the bounded-output/radius assumptions do not cover all modern models. That said, these are standard assumptions in model-free stability theory and are also used in related work (Soloff et al., 2024b).
>     - We will revise to make these distinctions and possible extensions more explicit.
> - **Question 3: definition of $\delta_t$:**  We apologize for the typo. In the current version, the reference should be to Eq. (40) in the proof, which describes the inflation behavior at each round of recursive feature bagging. The quantity $\delta_t$ is general and may vary with the space $\mathcal{W}$ and the estimator $w$. For example, upper bounds for $\delta_t$ are provided for RFS and RF in Thms. 4.7 and D.2, respectively. We will define $\delta_t$ explicitly and clarify its role in the main text.
> - **Weakness: Lack of generalization theory and weak empirical results:**  We agree that, in the current paper, the FS-generalization connection is empirical. Establishing such a result appears particularly challenging because a direct Bousquet-Elisseeff-type FS-generalization bound is not immediate: unlike leave-one-out over samples, removing a feature does not generally create an independent object. Developing such a generalization result is an important direction for future work.
> - **Weakness: figures are small / contents in appendix:**  We will enlarge figures and move key experiments to the main text (if space permitting).
> - **Weakness: limited empirical impact:** We agree that the empirical contribution is relatively limited compared with the theory. First, the current paper is already dense, leaving limited space for additional experiments. Also, we have substantially expanded the empirical study to test the robustness of the FS-generalization connection; please see rebuttal to Reviewer ExYY for details.
> - **Weakness: novelty beyond prior work:**  We thank the reviewer for this thoughtful comment. We agree that the paper builds on existing frameworks. However, we believe the novelty is more than direct extensions. Besides introducing FS, we go beyond generic upper bounds and explicitly compute the expected instability under distributional assumptions, yielding a clear comparison between feature and instance bagging. As noted above, the analysis can be readily extended to accommodate feature correlation and heterogeneity. Moreover, although the model-free analysis is technically related to recent work such as Soloff et al. (2024b, 2024c), the perspective is new, and the resulting guarantees for RF and RFS are, as we know, new as well.

---

> > ### Author Rebuttal · Reviewer_fQtK · 2026-04-01
> >
> > Thank you for the helpful rebuttal. The authors have addressed the concerns I raised and clarified the main points satisfactorily. Based on the response, I increased my score.

---

> > > ### Author Response · Authors · 2026-04-04
> > >
> > > Thank you for your thoughtful follow-up and positive assessment. We are glad that our responses adequately addressed all your concerns.
> > >
> > > If you find that the rebuttal has satisfactorily addressed the main concerns, we would be grateful if this could be reflected in your overall final assessment. If any issues still remain, we would be very happy to clarify them as clearly as possible.

---

### Official Review · Reviewer_i59N · 2026-03-12

**Soundness:** 3
**Presentation:** 3
**Significance:** 3
**Originality:** 3
**Overall Recommendation:** 4
**Confidence:** 3

**Summary:**

The paper is about the idea of feature stability (FS) in machine learning algorithms.  That is, the authors study how stable a learning algorithm can be when a single feature is being removed and ultimately care about the situation where different bags of features are being used to train different predictors and in sequence use such predictors in an ensemble form.  This idea is drawing a parallel from the idea of instance stability (IS) and of traditional bagging which is used in random forests.  IS is about the stability/sensitivity of a learning algorithm with respect to the removal of a single instance from the set of training examples.  The authors study the feature stability of linear regression for different sample sizes and input dimensions as well as for bagging rounds and subsampling ratios.  The authors also study assumption-free algorithms in the context of feature bagging.

**Compliance With Llm Reviewing Policy:**

Affirmed.

**Final Justification:**

I am happy with the response of the authors and the explanations that they provided to the various points raised in the reviews.  Having said that, I still have some reservations and I would like to maintain my score for weak accept (4).

**Key Questions For Authors:**

**Q1.** Regarding the joint prediction in situations where we have bagged features, do we feed instances to all individual predictors using the subset of features that they use and therefore we use all the models in the ensemble for the final joint prediction?  Or is something else happening?

**Limitations:**

Yes

**Strengths And Weaknesses:**

The paper builds on the well-known idea of instance stability.  However, this is presented as stability upon removal of a specific example, whereas what I am familiar with is the stability upon replacement of a specific training example (see, e.g., Section 13.2, in the book Understanding Machine Learning).  Regardless, I think there is merit in the work and the authors provide theoretical justification for their findings - though I have not checked the details of the proofs.  While bagging examples is a standard technique that can create an ensemble of different models that can be used for a joint prediction, I believe that the authors need to be more explicit for the mechanism used for a joint prediction in situations where we have bagged features.

---

> ### Author Rebuttal · Authors · 2026-03-29
>
> We thank the reviewer for the thoughtful and supportive review, as well as for the helpful questions. We sincerely hope that the rebuttal has addressed the concerns and would greatly appreciate it if the reviewer could consider updating the score accordingly.
> - **Question 1: joint prediction under feature bagging:** The reviewer is correct regarding the prediction scheme. At test time, the same input is passed to all base learners, and each learner uses only the coordinates in its own sampled feature subset. The final ensemble prediction is the average of these base-learner predictions. A key difference between feature-bagged and instance-bagged models arises at prediction time: in instance bagging, all submodels receive the same full test sample, whereas in feature bagging, each submodel receives only the coordinates corresponding to its sampled feature subset. We agree that this mechanism should be stated more explicitly in our description of feature bagging, and we will clarify it in the revision.
> - **Weakness: removal vs. replacement stability:** We agree that both removal- and replacement-based conventions appear in the stability literature. As discussed in Bousquet and Elisseeff and in Soloff et al. (2024b), the two are closely related, and results for one often imply results for the other. In our paper, we use removal as the primary convention, mainly because feature replacement is less natural to define and can complicate the analysis.
> For example, without distributional assumptions, one may define replacement by modifying only the $j$-th coordinate of $X$, i.e., taking $X' \in \mathcal{X}$ such that $X^{\prime k} = X^k$ for all $k \neq j$. With distributional information, a more principled definition would replace $X_i^j$ by an i.i.d. copy conditioned on $(Y_i, X_i^{-j})$. Both formulations lead to significantly more involved analysis in the linear setting.
> Nevertheless, replacement-based results are directly implied by our current bounds, since
> $$\mathrm{Stability}(\mathcal{D}, \mathcal{D}^{(j)}) \le
> \mathrm{Stability}(\mathcal{D}, \mathcal{D}^{-j}) +
> \mathrm{Stability}(\mathcal{D}^{(j)}, \mathcal{D}^{-j}).$$
> We thank the reviewer for this comment and will make this connection explicit in the revision.

---

> > ### Author Rebuttal · Reviewer_i59N · 2026-04-04
> >
> > I would like to thank the authors for their response. I would also like to maintain my score.

---

> > > ### Author Response · Authors · 2026-04-04
> > >
> > > Thank you for your thoughtful follow-up and positive assessment. We are glad that our responses adequately addressed all your concerns.
> > >
> > > If you find that the rebuttal has satisfactorily addressed the main concerns, we would be grateful if this could be reflected in your overall final assessment. If any issues still remain, we would be very happy to clarify them as clearly as possible.

---

### Decision · Program_Chairs · 2026-04-30

**Decision:**

Accept (regular)

**Comment:**

The paper introduces feature stability, which support the argument that feature bagging improves algorithmic stability. The review finds the work valuable, while also remains various concerns, including the soundness and applicability to distinct scenarios and settings, and valid experiments and evaluation.